# High-throughput fitness experiments reveal specific vulnerabilities of human-adapted *Salmonella* during stress and infection

Benjamin X. Wang [1], Dmitry Leshchiner[2], Lijuan Luo[3], Miles Tuncel [1], Karsten Hokamp [4], Jay C. D. Hinton [3] & Denise M. Monack [1]✉

*Salmonella enterica* is comprised of genetically distinct 'serovars' that together provide an intriguing model for exploring the genetic basis of pathogen evolution. Although the genomes of numerous *Salmonella* isolates with broad variations in host range and human disease manifestations have been sequenced, the functional links between genetic and phenotypic differences among these serovars remain poorly understood. Here, we conduct high-throughput functional genomics on both generalist (Typhimurium) and human-restricted (Typhi and Paratyphi A) *Salmonella* at unprecedented scale in the study of this enteric pathogen. Using a comprehensive systems biology approach, we identify gene networks with serovar-specific fitness effects across 25 host-associated stresses encountered at key stages of human infection. By experimentally perturbing these networks, we characterize previously undescribed pseudogenes in human-adapted *Salmonella*. Overall, this work highlights specific vulnerabilities encoded within human-restricted *Salmonella* that are linked to the degradation of their genomes, shedding light into the evolution of this enteric pathogen.

*Salmonella enterica*—an enteric human pathogen causing 100 million infections and 200,000 deaths annually[1–3]—is comprised of distinct strains, or 'serovars', with varied host ranges and disease presentations. Nontyphoidal serovars like *S. enterica* serovar Typhimurium (*S.* Typhimurium), are generalists with broad host range causing self-limiting gastroenteritis in humans[1,4], while typhoidal *Salmonella*, including *S.* Typhi and *S.* Paratyphi A, are human-restricted and induce enteric fever[4–6]—a severe systemic infection (Fig. 1a). Despite decades of study, the molecular mechanisms governing host range and disease differences among these serovars remain largely unclear.

Genomic analyses highlight notable genetic variation and serovar-specific genes in typhoidal *Salmonella*[4,7–12], particularly within *Salmonella* pathogenicity islands (SPIs). For example, SPI-7 in *S.* Typhi encodes the Vi capsule, which inhibits complement binding[13], dampens immune responses[14] and prevents uptake by neutrophils[15]. SPI-11 of *S.* Typhi encodes the typhoid toxin[16]—a typhoid-specific virulence factor inducing DNA damage in host cells[17,18]. Importantly, typhoidal serovars harbor hundreds of pseudogenes, some of which are involved in intestinal colonization when functional[1,19–21], indicating genomic decay in human-restricted *Salmonella* during its evolution to an extraintestinal pathogen[7].

Despite abundant bioinformatic data, our understanding of how genotypic differences among *Salmonella* isolates correlate with phenotypic variations remains limited. Bioinformatic approaches have identified hundreds of typhoid-specific genes, but many lack known physiological functions[1,4]. Moreover, although human-restricted *Salmonella* encode hundreds of pseudogenes[7,11,12], the functional consequences of pseudogene accumulation have not been

[1]Department of Microbiology and Immunology, Stanford University School of Medicine, Stanford, CA, USA. [2]Biology Department, Boston College, Chestnut Hill, MA, USA. [3]Institute of Infection, Veterinary and Ecological Sciences, University of Liverpool, Liverpool, UK. [4]Department of Genetics, School of Genetics and Microbiology, Smurfit Institute of Genetics, Trinity College Dublin, Dublin, Ireland. ✉e-mail: dmonack@stanford.edu

studied systematically. Bioinformatics also cannot easily determine whether genes conserved across *Salmonella* serovars have distinct functions in different isolates. Thus, more detailed functional studies are needed to better characterize the impact of genetic variation across *Salmonella*.

Transposon sequencing (Tn-seq) is a powerful method linking genotype to phenotype, extensively contributing to *Salmonella* biology by uncovering genes crucial for survival under various infection-related stresses[22–33]. However, most *Salmonella* Tn-seq studies have focused on generalist serovars like *S.* Typhimurium, while human-adapted strains remain severely understudied. Moreover, Tn-seq is costly and low-throughput, limiting the number of conditions assayed in each study. Recent advances in random barcoded Tn-seq (Rb-Tn-seq) overcome this limitation by enabling high-throughput assessment of microbial fitness[34–36], but Rb-Tn-seq has not been applied systematically to interrogate *Salmonella* virulence and evolution.

Here, we employ Rb-Tn-seq to explore genotypic and phenotypic differences among generalist and human-restricted *Salmonella* serovars. We capture thousands of significant fitness events across 25 host-associated stresses encountered at key stages of *Salmonella* infection within humans. We use a systems biology approach to identify serovar-specific changes in fitness within gene networks, including those involved in lipopolysaccharide (LPS) modification, amino acid metabolism and metal homeostasis. We perturb these networks experimentally to identify specific pseudogenes, including several previously undescribed pseudogenes, contributing to typhoidal-specific fitness effects. Overall, our results provide a comprehensive functional perspective on how genetic differences between generalist and host-restricted *Salmonella* have influenced the evolution of this enteric pathogen.

## Results

### Set-up of Rb-Tn-seq experiments

We constructed Rb-Tn-seq libraries to study *Salmonella* stress response[37] in four phylogenetically distinct and genetically tractable serovars. These included two generalist isolates (*S.* Typhimurium ST4/74 and D23850) and two human-restricted isolates (*S.* Typhi Ty2 and *S.* Paratyphi A 9150). On average, each library contained 166,905 unique genome-wide transposon insertion sites which integrated every 27.7 bp. The median insertions per gene ranged from 12 to 44 across these serovars, with central Tn insertions in ~90–92% of coding genes. Barcoded transposons showed even distribution across chromosomes and plasmids, ensuring high genome coverage with minimal strand or coverage bias (Fig. 1b, Supplementary Fig. 1a–e and Supplementary Table 4). The insertion index of each gene was calculated to identify 427 to 476 putative essential genes for each serovar (Supplementary Data 1 and Supplementary Fig. 2a–d), several of which were serovar-specific, including *igaA* in Typhimurium ST4/74 and D23580, *rpoE* in *S.* Typhi Ty2 and various iron homeostasis genes in Paratyphi A (Supplementary Fig. 2e and Supplementary Note 1)[38].

We conducted fitness assays on each library, evaluating their response to (1) extracellular stresses encountered in the intestinal tract and/or systemic tissues, (2) intracellular stresses within host cells, including macrophages and (3) exposure to a diverse suite of antibiotics (Fig. 1c and Supplementary Table 5). Stressor concentrations were optimized to achieve ~30–50% growth reduction[38] (Fig. 1c and Supplementary Fig. 3a–d). Each experiment included biological duplicates, which were tightly correlated and passed published quality-control metrics[36] (Supplementary Fig. 3e–h and Supplementary Data 2).

We used a moderated *t*-like statistic with $|t| > 4$ to identify significant fitness events[36], leading to the identification of hundreds of genes with significant fitness effects for each serovar across our conditions (Fig. 1d and Supplementary Data 3 and 4). We then used agglomerative clustering to generate heatmaps with all genes with $|t| > 4$ in at least 1 condition, retaining 678 to 781 genes for each serovar (Supplementary Figs. 4–7). These clustered heatmaps revealed patterns linking similar conditions, including grouping intracellular stresses (for example, InSPI2, InSPI2 Mg, $H_2O_2$, NO, bleach), extracellular stresses (for example bile, heat stress, anaerobiosis, gut microbiota media (GMM)) and another grouping linking various antibiotics (for example, ciprofloxacin, azithromycin, rifampicin).

We identified many functionally related gene clusters involved in stress response (Supplementary Data 5). For instance, *acrAB/tolC* Tn insertions displayed significant fitness defects during bile stress for typhoidal *Salmonella* (Fig. 1e)[39,40]. LPS modification *arn* operon and *pmrAB* mutations exhibited reduced fitness under polymyxin B in Typhimurium D23580 (Fig. 1e and Supplementary Fig. 8a)[41–44]. Mutations in DNA repair genes (for example *recDGNQX*) led to decreased D23580 fitness with ciprofloxacin—an antibiotic inducing DNA damage (Supplementary Fig. 8b)[45,46]. Tn insertions in iron homeostasis genes (for example, *entDEF*, *exbD*, *tonB*) caused decreased fitness under iron restriction in Paratyphi A 9150 (Supplementary Fig. 8c). Tn insertions in molybdenum (Mo) metabolism genes (for example, *moeA*, *moaA*, *mog*, *mobA*) showed fitness defects in GMM in several serovars (Supplementary Fig. 8d)[47,48]. LPS-synthesizing gene mutations (for example, *rfaL*, *rfbBD*, *wzyO4*, *waaK*) increased sensitivity to both intracellular and extracellular stresses, underscoring the broad role of LPS during bacterial stress response[49,50] (Supplementary Fig. 8e). Intriguingly, mutations in *barA* and *sirA* exhibited increased fitness under multiple stresses (Supplementary Fig. 8f), possibly explaining their frequent occurrence in chronically infected *Salmonella* patients[51].

Several SPI-encoded genes displayed significant phenotypes (Supplementary Data 6)[4]. For instance, mutations in *hilD*—an SPI-1 encoded transcription factor—increased fitness during bile stress in *S.* Typhi Ty2 and heat shock in *S.* Typhimurium D23580 (Supplementary Data 6), aligning with a study proposing a role for HilD role in enhancing membrane permeability[52]. SPI-3-encoded magnesium importers *mgtB* and *mgtC* mutations decreased *S.* Paratyphi A 9150 survival in

**Fig. 1 | Construction and validation of Rb-Tn-seq libraries in four serovars.**
**a**, Left, genome alignments of *S.* Typhimurium ST4/74, *S.* Typhi Ty2, *S.* Paratyphi A 9150 and *S.* Typhimurium D23580. GC skew is indicated by the internal black trace. Right, schematic displaying the host range of generalist and human-restricted *Salmonella*. **b**, Location of all barcoded transposon insertions in each genome, as indicated by the colored lines on the outside of the gray circle (representing the chromosome). **c**, Schematic of Rb-Tn-seq workflow. Left, general growth scheme for all Rb-Tn-seq experiments. Right, 24 plate-based in vitro conditions tested, sorted by the type of stressor. Iron limitation can be encountered both intracellularly and extracellularly and is depicted in the overlap region. **d**, Plots showing all significant fitness changes ($|t| > 4$) for each isolate, across all 24 conditions. Red, *S.* Typhimurium ST4/74; blue, *S.* Typhi Ty2; green, *S.* Paratyphi A 9150; purple, *S.* Typhimurium D23580. Data are combined from two biologically independent Rb-Tn-seq replicates. **e**, Top, correlation of

gene fitness changes between (−) bile (LB only) and (+) bile (LB + 4% ox bile) in *S.* Typhi Ty2, with the *acrAB/tol* genes highlighted in red and all other genes shown in purple. Bottom, correlation of gene fitness changes between (−) polymyxin B and (+) polymyxin B in *S.* Typhimurium D23580, with LPS modification genes in red (including *arn* operon) and *pmrABD* genes highlighted in blue; all other genes shown in purple. For both panels in **e**, data are derived from $n = 2$ biologically independent Rb-Tn-seq replicates and shown as a density plot, where colors range from dark purple (low) to yellow (high), representing the kernel density estimation from low to high density. **f**, Functional classification of genes with significant fitness changes for each serovar, based on available GO terms and manual classifications. In **a**, **b** and **f**, *S.* Typhimurium ST4/74 is in red, *S.* Typhi Ty2 is in blue, *S.* Paratyphi A 9150 is in green, and *S.* Typhimurium D23580 is in purple. TCS, two-component system. Created with Biorender.com.

macrophage-mimicking media InSPI2 Mg (Supplementary Data 6). Mutations in SPI-7-encoded Vi capsule genes in Typhi increased fitness under protamine stress—a positively charged antimicrobial peptide (Supplementary Data 6). Despite Typhi and Paratyphi A encoding unique genes not found in Typhimurium genomes, only a small proportion of these unique genes exhibited significant fitness effects (3.9% in Typhi, 3.2% in Paratyphi A) (Supplementary Data 7). In contrast, a higher proportion of shared orthologs had significant phenotypes in Typhi (18.5%) and Paratyphi A (16.5%) (Supplementary Table 6). To highlight specific processes involved in *Salmonella* stress response,

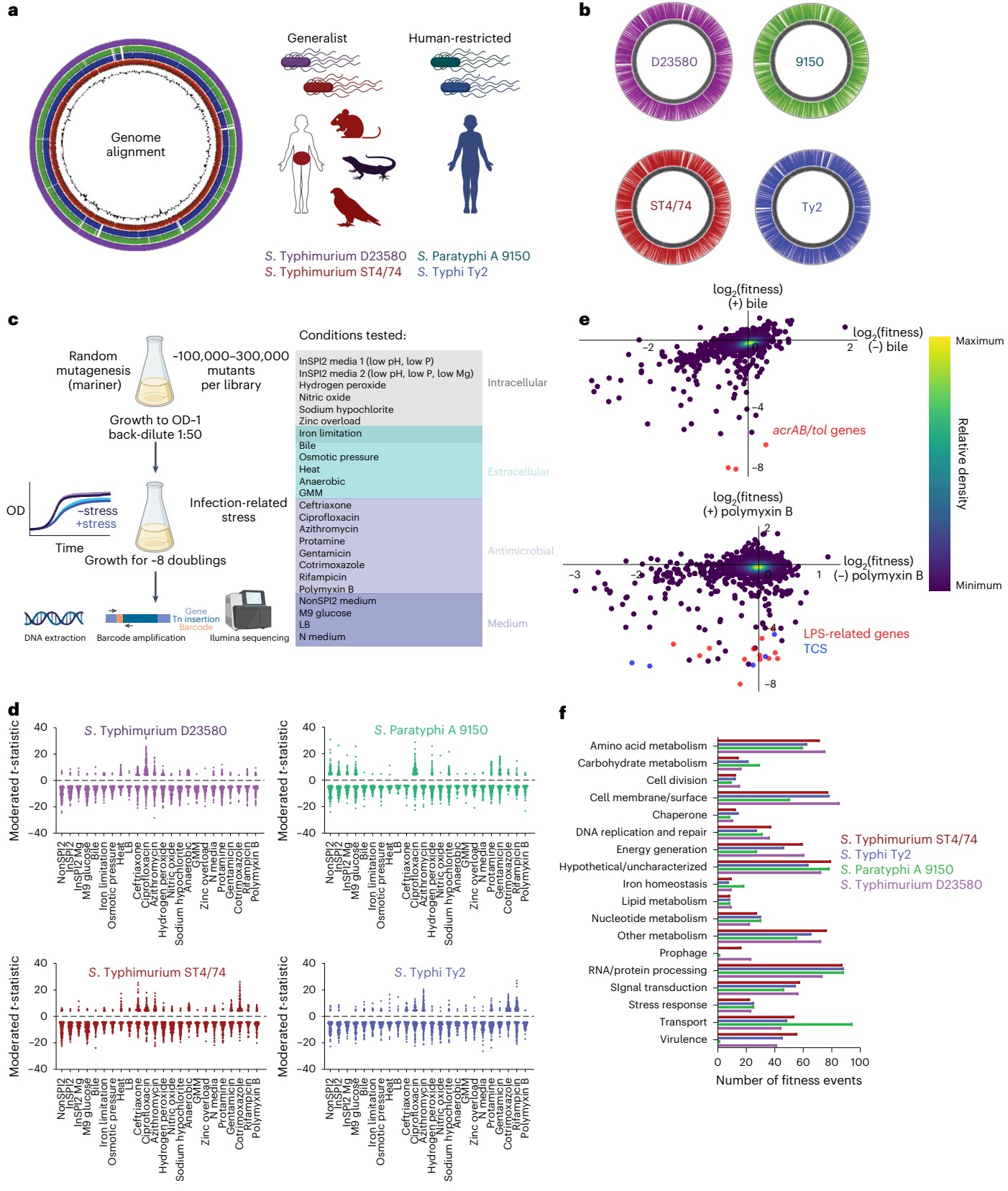

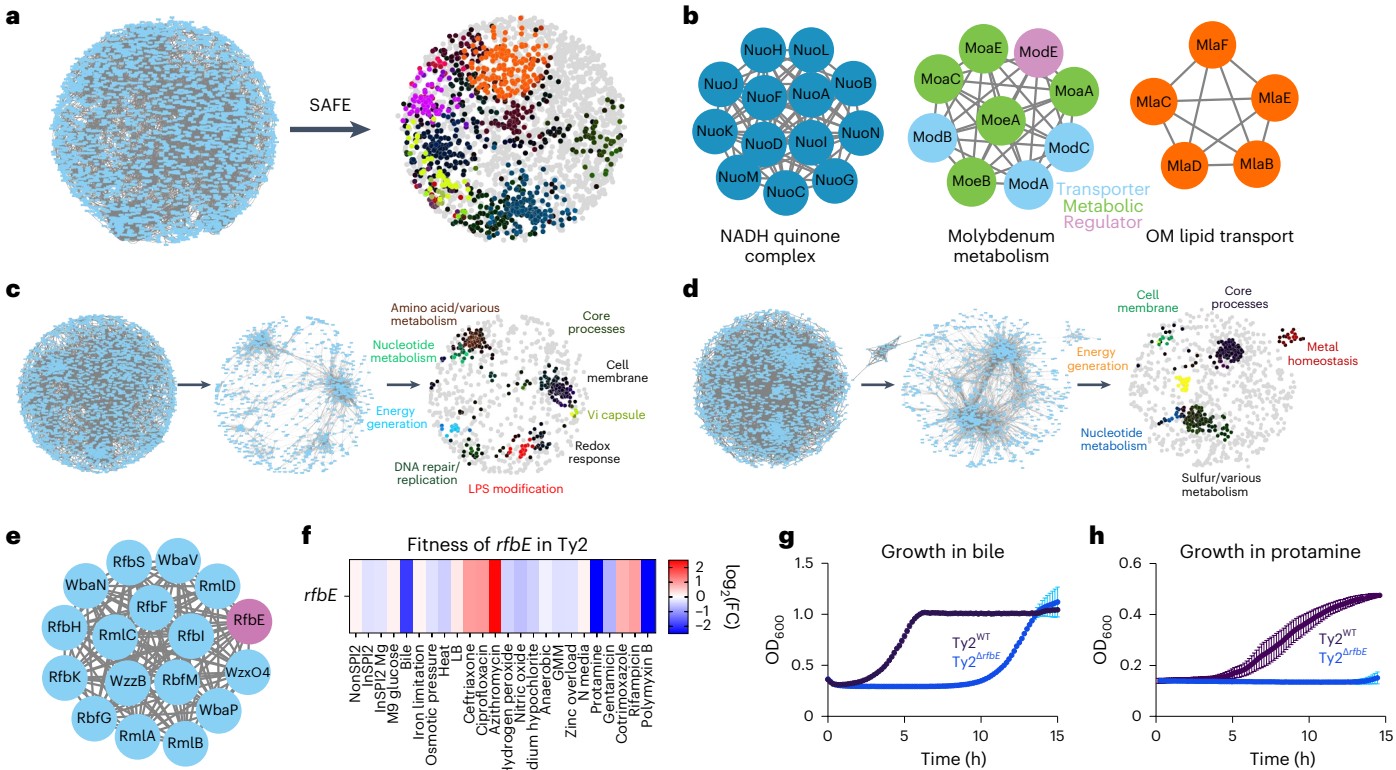

**Fig. 2 | Cofitness network analysis and SAFE identify serovar-specific gene fitness changes. a**, Left, cofitness network analysis of all genes in *S*. Typhimurium ST4/74 with *r* > 0.75; all blue nodes are genes and all gray lines connect pairs of cofit (*r* > 0.75) genes. Right, SAFE highlights regions of the network that are enriched in certain functional terms. Each colored area represents a different functionally enriched area on the network. **b**, Examples of subclusters of genes that are identified through SAFE. **c,d**, Left, cofitness network analysis of all genes in *S*. Typhi Ty2 (**c**) and *S*. Paratyphi A 9150 (**d**) with *r* > 0.75; all blue nodes are genes and all gray lines are between pairs of cofit genes. Middle, filtered network that only includes genes with fitness changes that are (1) significant in *S*. Typhi/*S*. Paratyphi A but not in *S*. Typhimurium and (2) > 2-FC in *S*. Typhi/*S*. Paratyphi A compared with S. Typhimurium. Right, SAFE highlights regions of the network

that are enriched in certain functional terms on these filtered maps. **e**, An LPS modification cluster including *rfbE* (purple) is identified through SAFE on the *S*. Typhi filtered map. **f**, Heatmap showing the fitness values of Tn insertions in *rfbE* across 24 plate-based stresses for *S*. Typhi Ty2. Color gradient is derived from the $\log_2(FC)$ from each condition in the Rb-Tn-seq experiments. **g**, Growth curves of Ty2$^{WT}$ (black) and Ty2$^{\Delta rfbE}$ (blue) when exposed to 4% bile, with reads taken at OD$_{600}$ once every 10 min. **h**, Growth curves of Ty2$^{WT}$ (black) and Ty2$^{\Delta rfbE}$ (blue) when exposed to 2.3 μg ml$^{-1}$ protamine, with reads taken at OD$_{600}$ once every 10 min. For growth curve experiments (**g,h**), each point and error bar indicates the mean ± s.e.m. of OD$_{600}$, derived from *n* = 4 (**g**) and *n* = 3 (**h**) biologically independent experiments.

we sorted significant fitness events by annotated gene ontology (GO) terms and BioCyc-derived functional annotations, grouped into various functional classes[38,55] (Fig. 1f). Dozens of fitness events involved uncharacterized genes, suggesting that our Rb-Tn-seq dataset holds rich uncharacterized biology (Fig. 1f).

**Systems biology approach to analyze fitness profiles**
To systematically analyze the thousands of fitness effects captured through Rb-Tn-seq, we employed cofitness network analysis and spatial analysis of functional enrichment (SAFE) to overlay functional data onto network maps[38], which has been performed previously in *Saccharomyces cerevisiae*[53] and *Streptococcus pneumoniae*[38]. Briefly, we constructed correlation matrices reflecting the $\log_2$ fitness changes for each gene across all conditions. These matrices were transformed into cofitness interaction networks, where nodes represented genes and edges indicated correlation values, using a Pearson's correlation of *R* > 0.75 to identify closely related fitness profiles (Fig. 2a and Supplementary Data 8). Stability testing[38] indicated high significance across our networks (Supplementary Data 8).

SAFE was then applied to annotate each node based on BioCyc classifications and GO terms (Supplementary Data 9)[54], through which we identified local network neighborhoods enriched for specific functional classes[38,55] (Fig. 2a and Extended Data Fig. 1a–d). To validate this

analysis, we searched SAFE outputs for gene networks expected to cluster together based on related functionality, finding clusters for the NADH quinone oxidoreductase complex, molybdenum metabolism genes and genes associated with lipid trafficking to the outer membrane (Fig. 2b). Intriguingly, ~15–20% submodules contain at least one hypothetical gene, including *RS16480/STM_3341* in ST4/74, which is correlated strongly with *cpxR*, and *RSO3310/tO654* in Typhi Ty2, which is correlated strongly with 24 amino acid metabolism genes (Supplementary Data 10 and Supplementary Note 2)[56].

We used the cofitness network pipeline to pinpoint typhoid-specific fitness changes. To this end, we applied an additional filtering step where, for each condition, we only retained genes that (1) had a significant fitness change (|*t*| > 4) in Typhi Ty2 or Paratyphi A 9150 but not in Typhimurium ST4/74 and (2) had a fold change (FC) that was at least twofold greater in Typhi or Paratyphi compared with Typhimurium (Fig. 2c,d and Supplementary Data 11–12). These filters removed ~75% and 65% of the nodes and connections within each cofitness network for Typhi and Paratyphi, respectively, but still retained thousands of serovar-specific correlations. We then applied SAFE to these filtered networks and identified gene clusters with serovar-specific fitness changes (Fig. 2c,d), including those involved in LPS modification, amino acid metabolism and metal homeostasis. In contrast, only ~3% of the network was retained in D23580 when doing this same analysis,

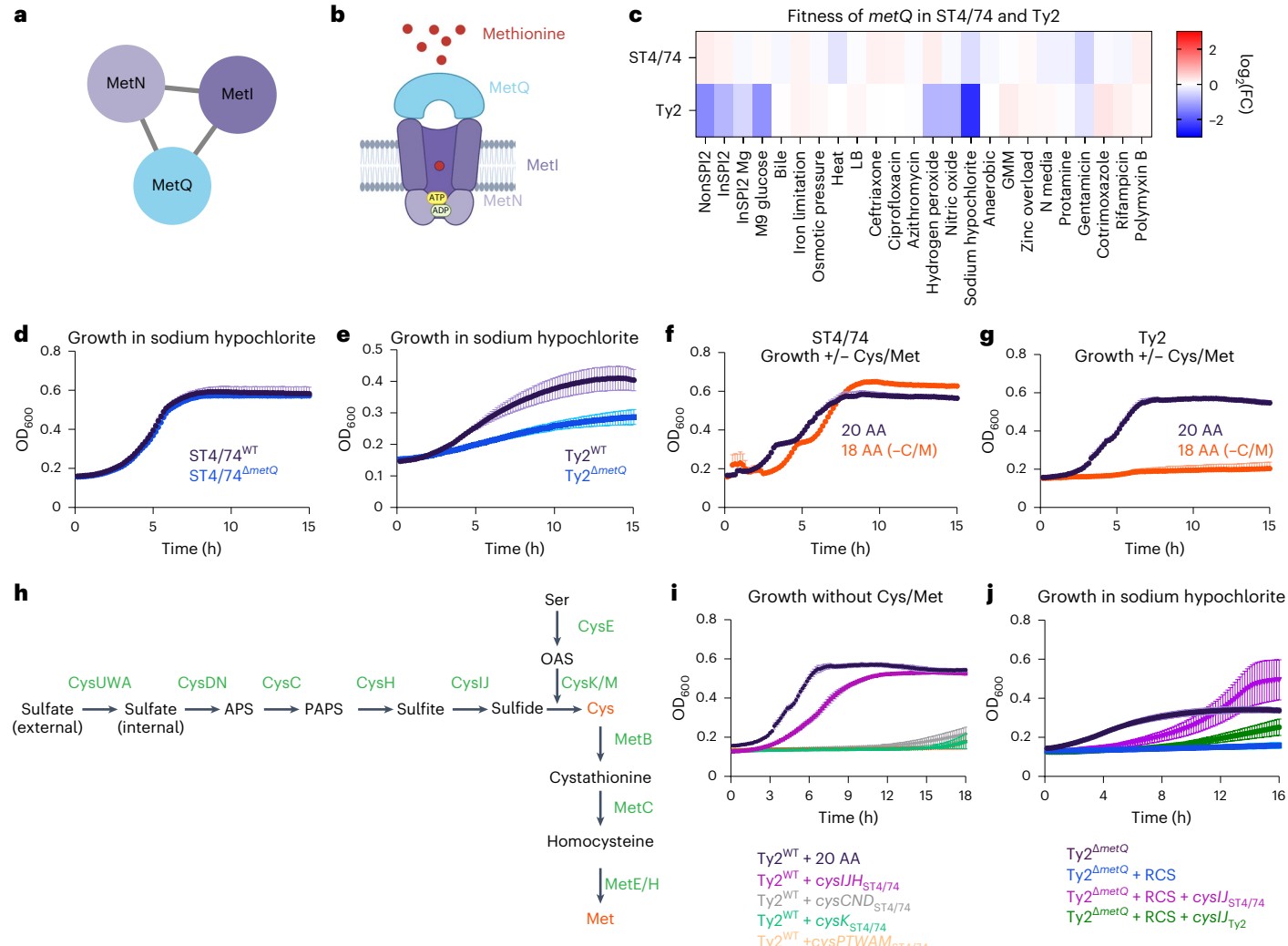

**Fig. 3 | *metIQN* has typhoid-specific fitness defects during RCS. a**, *metIQN* cluster, derived from the *S*. Typhi Ty2 filtered network. **b**, Schematic of MetIQN. **c**, Heatmap showing fitness of m*etQ* Tn insertions across 24 conditions for *S*. Typhimurium ST4/74 and *S*. Typhi Ty2. Color gradient is derived from the log$_2$(FC) from Rb-Tn-seq. **d**, Growth of ST4/74$^{WT}$ (black) and ST4/74$^{\Delta metQ}$ (blue) exposed to 12.5 µg ml$^{-1}$ sodium hypochlorite, derived from *n* = 3 biologically independent experiments. **e**, Growth of Ty2$^{WT}$ (black) and Ty2$^{\Delta metQ}$ (blue) exposed to 12.5 µg ml$^{-1}$ sodium hypochlorite, derived from *n* = 3 biologically independent experiments. **f,g**, Growth of ST4/74$^{WT}$ (**f**) and Ty2$^{WT}$ (**g**) when grown in defined minimal medium with a mix of all 20 amino acids added (20 AA; purple), or with 18 amino acids added, without Cys/Met (18AA (−C/M); orange), derived from *n* = 3 biologically independent experiments. **h**, Schematic of endogenous synthesis pathway for Cys/Met. Intermediate substrates are shown in black and enzymes are shown in green. Cys/Met are shown in red. **i**, Growth of WT and complemented strains in which different operons in the endogenous Cys/Met pathway from

*S*. Typhimurium ST4/74 are expressed in *S*. Typhi Ty2. As a control, Ty2$^{WT}$ was grown in minimal medium supplemented with 20 AA and is shown in black (*n* = 3 biologically independent experiments). All complementation growth curves were run in minimal medium supplemented with 18AA, but no Cys/Met (−C/M), and are derived from *n* = 4 biologically independent experiments; magenta, CysIJH$_{ST4/74}$; gray, CysCND$_{ST4/74}$; green, CysK$_{ST4/74}$; orange, CysAWUM$_{ST4/74}$. **j**, Growth curves of WT, mutant and complementation strains when exposed to a lethal dose of 25 µg ml$^{-1}$ sodium hypochlorite. Black, Ty2$^{\Delta metQ}$ growth in the absence of RCS; blue, Ty2$^{\Delta metQ}$ growth with RCS; purple, growth of Ty2$^{\Delta metQ}$ + cysIJ$_{ST4/74}$ in the presence of RCS; green, growth of Ty2$^{\Delta metQ}$ + cysIJ$_{Ty2}$. All curves are derived from *n* = 5 biologically independent experiments. For all growth curves (**d**–**g**,**i**,**j**), each point and error bar indicates the mean ± s.e.m. of OD$_{600}$ with reads taken at OD$_{600}$ once every 10 min. APS, adenosine 5′-phosphosulfate; OAS, O-acetyl-Ser; PAPS, 3′-phosphoadenosine 5′-phosphosulfate. Created with Biorender.com.

indicating similar fitness profiles between these Typhimurium isolates (Extended Data Fig. 1e and Supplementary Data 13).

We then examined several gene networks with serovar-specific changes in typhoidal *Salmonella*, focusing on three categories for further mechanistic investigation. The first includes clusters with genes unique to typhoidal *Salmonella*, including *rfbE* (Fig. 2). The second comprises gene clusters where all genes are shared between nontyphoidal and typhoidal *Salmonella* but exhibit different phenotypes in distinct isolates (for example, *metIQN* in Fig. 3). The third involves gene clusters containing at least one uncharacterized gene in *Salmonella* (for example, *ybdZ* in Fig. 4).

We were intrigued by an LPS modification gene cluster containing *rfbE* (Fig. 2e), a typhoid-specific gene that modifies the O-antigen terminal sugar to tyvelose[56,57]. While tyvelose is used for serotyping, its biological function in typhoidal *Salmonella* remains unclear. Tn insertions in *rfbE* sensitize typhoidal *Salmonella* to bile acids, protamine and polymyxin B (Fig. 2f and Supplementary Fig. 9). To validate these results, we deleted *rfbE* in *S*. Typhi Ty2 and found that Ty2$^{\Delta rfbE}$ exhibited decreased survival in response to bile acids (Fig. 2g) and protamine (Fig. 2h) compared with Ty2$^{WT}$, suggesting that this unique tyvelose moiety protects Typhi from membrane-insulting stresses and antibiotics.

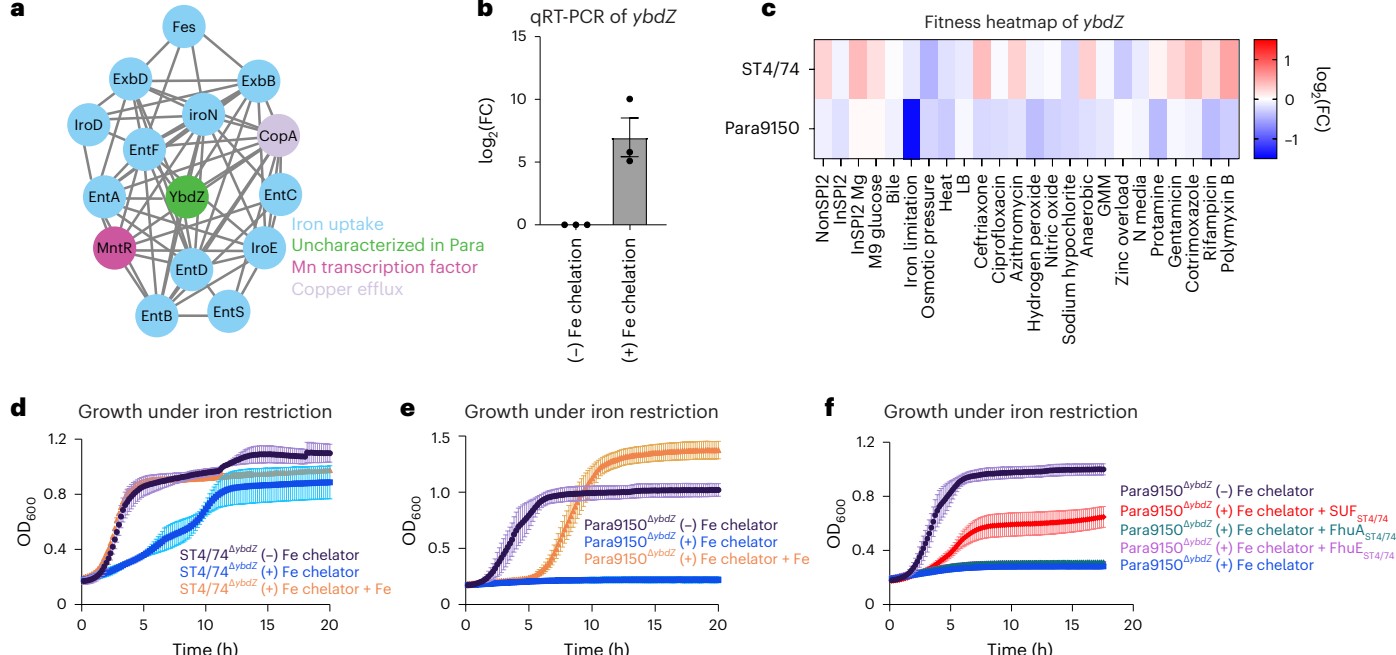

**Fig. 4 | A *ybdZ*-containing gene cluster with paratyphoid-specific fitness effect. a**, Cluster of genes involved in metal homeostasis, identified through SAFE on the *S.* Paratyphi A 9150 filtered network. **b**, Transcript levels of *ybdZ* measured by qRT-PCR and normalized to a control gene (*rpoD*). Bars indicate mean ± s.e.m., with individual measurements shown (black dots). Each bar is derived from $n = 3$ biologically independent experiments. **c**, Heatmap showing the fitness values of Tn insertions in *ybdZ* across 24 stress conditions for *S.* Typhimurium ST4/74 and *S.* Paratyphi A 9150. Color gradient is derived from the $\log_2(FC)$ from each condition in the Rb-Tn-seq experiments. **d**, Growth curves of ST4/74$^{WT}$ (black) and ST4/74$^{\Delta ybdZ}$ (blue) when exposed to 100 µM 2,2′-dipyridyl, +/− addition of exogenous 1 mM FeCl$_3$ (orange), with reads taken at OD$_{600}$ once every 10 min, derived from $n = 3$ biologically independent experiments. **e**, Growth curves of Para9150$^{WT}$ (black) and Para9150$^{\Delta ybdZ}$ (blue) when exposed to

100 µM 2,2′-dipyridyl, +/− addition of exogenous 1 mM FeCl$_3$ (orange), with reads taken at OD$_{600}$ once every 10 min, derived from $n = 3$ biologically independent experiments. **f**, Growth curves of Para9150$^{\Delta ybdZ}$ and complementation strains in which functional versions of different iron-related pseudogenes from *S.* Typhimurium ST4/74 are expressed in Para9150$^{\Delta ybdZ}$ under iron restriction, with reads taken at OD$_{600}$ once every 10 min; SUF$_{ST4/74}$ is shown in red, FhuA$_{ST4/74}$ is shown in green, and FhuE$_{ST4/74}$ is shown in purple. For controls, the growth of Para9150$^{\Delta ybdZ}$ without iron restriction (−100 µM 2,2′-dipyridyl) is shown in black, while the growth of Para9150$^{\Delta ybdZ}$ under iron restriction (+100 µM 2,2′-dipyridyl) is shown in blue. Each point and error bar indicates the mean ± s.e.m. of OD$_{600}$, where $n = 3$ biologically independent experiments for all curves except SUF$_{ST4/74}$ in red, which was derived from $n = 7$ biologically independent experiments.

## A typhoid-specific phenotype for *metQ* during reactive chlorine stress

*metIQN* encodes the main methionine transporter in *Salmonella* and was identified as a gene cluster with typhoid-specific fitness effects by SAFE[58] (Fig. 3a,b). Tn insertions in *metIQN* rendered both *S.* Typhi Ty2 and *S.* Paratyphi A highly susceptible (~sevenfold) to reactive chlorine stress (RCS) but did not sensitize *S.* Typhimurium ST4/74 (Fig. 3c and Extended Data Fig. 2a,b). To confirm these Rb-Tn-seq results, we deleted *metQ* in both *S.* Typhi Ty2 and *S.* Typhimurium ST4/74 and found that Ty2$^{\Delta metQ}$ had a more severe growth defect under RCS than ST4/74$^{\Delta metQ}$ (Fig. 3d,e). Deleting *metQ* in six *S.* Typhi clinical isolates, including CT18 and multidrug resistance strains belonging to the H58 lineage (ISP-04-06979, ISP-03-07467, E03-9804, ISO(98S) and E03-4983)[59], also increased RCS sensitivity (Extended Data Fig. 2c–h).

RCS kills bacteria by oxidizing cysteine (Cys) and methionine (Met)[60]. *Salmonella* replenishes these sulfated amino acids by either importing them via dedicated transporters like MetIQN or through de novo synthesis[58]. Given the increased susceptibility of Ty2$^{\Delta metQ}$ to RCS, we reasoned that endogenous Cys/Met in Typhi may be impaired. To test this hypothesis, we cultured both *S.* Typhimurium ST4/74 and *S.* Typhi Ty2 in defined minimal medium. Both strains grew when all 20 amino acids (AA) were added (Fig. 3f,g). However, in medium lacking Cys/Met (18AA −Cys −Met), ST4/74 continued to grow, while Ty2 failed to grow (Fig. 3f,g), suggesting that *S.* Typhi cannot synthesize Cys/Met endogenously.

To pinpoint the compromised part of the Cys/Met synthesis pathway in *S.* Typhi (Fig. 3h), we expressed plasmid-borne functional

versions of each operon in this pathway (*cysIJH*, *cysCND*, *cysK*, *cysPT-WAM*) from *S.* Typhimurium in *S.* Typhi Ty2. Only *cysIJH* from ST4/74 restored Ty2 growth in the 18AA medium lacking Cys/Met (Fig. 3i). Furthermore, deleting *cysIJ* in ST4/74 ablated growth of this isolate in medium without Cys/Met (Extended Data Fig. 2i), emphasizing the importance of CysI/J in both nontyphoidal and typhoidal *Salmonella* under Cys/Met limitation. Importantly, *cysIJ*$_{ST4/74}$ expression rescued Ty2$^{\Delta metQ}$ growth under a lethal dose of RCS (Fig. 3j). In contrast, expressing *cysIJ* from Ty2 only partially rescued *S.* Typhi growth in Cys/Met-deficient media (Extended Data Fig. 2j), and weakly in Ty2$^{\Delta metQ}$ under lethal RCS exposure (Fig. 3j), further suggesting that CysIJ$_{Ty2}$ function is impaired. Collectively, these findings indicate the increased sensitivity of Ty2$^{\Delta metQ}$ to RCS is driven by defects in endogenous Cys/Met synthesis.

## A serovar-specific phenotype for *ybdZ* under iron restriction

We identified a paratyphoid-specific gene network displaying fitness defects, primarily consisting of iron-related genes and featuring an unannotated gene *RS10805*, sharing ~60% identity with *ybdZ* in *Escherichia coli* (Fig. 4a and Extended Data Fig. 3a–c), which enhances enterobactin production[61]. Quantitative real-time (qRT)-PCR showed an increase in *ybdZ* expression of ~200-fold during iron restriction in *S.* Paratyphi A (Fig. 4b), consistent with a role during iron limitation in *Salmonella*. Tn insertions in *ybdZ* caused a more pronounced fitness defect under iron limitation in *S.* Paratyphi A than in *S.* Typhimurium (Fig. 4c). To confirm this Rb-Tn-seq result, we deleted *ybdZ* in both S. Paratyphi A 9150 and S. Typhimurium ST4/74 and observed

that Para9150$^{\Delta ybdZ}$ had a stronger growth defect under iron limitation than ST4/74$^{\Delta ybdZ}$ (Fig. 4d,e). Deleting *ybdZ* in other Paratyphi A strains (Para11511 and Para12176) also led to heightened sensitivity to iron restriction, and exogenous iron rescued these phenotypes (Extended Data Fig. 3d,e).

To understand why Para9150$^{\Delta ybdZ}$ exhibits a more severe growth defect under iron limitation than ST4/74$^{\Delta ybdZ}$, we investigated the impact of pseudogenes in typhoidal *Salmonella* involved in iron acquisition, including *fhuA*, *fhuE*, *sufD* and *sufS*[7]. SufS/D are components of the SUF complex, one of two multiprotein complexes that synthesizes iron-sulfur clusters. In *E. coli*, SUF contributes to survival during iron restriction[62–64]. Strikingly, expressing functional versions of these genes from *S*. Typhimurium in Para9150$^{\Delta ybdZ}$ revealed that only SUF$_{ST4/74}$ expression restored growth during iron restriction (Fig. 4f), suggesting that functional SUF is sufficient to rescue this growth defect. To determine whether SUF is also necessary for survival under iron limitation in a $\Delta ybdZ$ mutant background, we constructed a double deletion mutant of *ybdZ* and *sufSD* ($\Delta ybdZ\Delta sufSD$) in Typhimurium ST4/74 and found that it was highly sensitive to iron restriction (Extended Data Fig. 3f), confirming that SUF contributes to survival under iron restriction in both nontyphoidal and typhoidal *Salmonella*. Notably, *sufS* and/or *sufD* are pseudogenes in all deposited Paratyphi A sequences on BioCyc, indicating the increased sensitivity of Paratyphi A to iron restriction is likely a general feature of Paratyphi A.

## Serovar-specific fitness during macrophage infection

We aimed to identify serovar-specific phenotypes in a host-associated setting. *Salmonella* replication within mammalian macrophages is a key feature of this pathogen[65–67]. Despite previous Tn-seq studies examining *Salmonella* fitness in macrophages[27–29], a systematic comparison of intracellular fitness profiles between generalist and human-restricted *Salmonella* within human macrophages is lacking. To address this gap, we performed Rb-Tn-seq on human THP-1 macrophages infected with our four barcoded libraries, using a multiplicity of infection and infection duration favoring *Salmonella* replication while minimizing host cell death (Fig. 5a and Supplementary Fig. 10a–g). Two macrophage passages were conducted to enrich subtle phenotypes (Fig. 5a). Consistent with published work, the absence of Vi capsule increased *S*. Typhi uptake into macrophages[68] (Supplementary Fig. 10h); thus, we performed the Typhi Rb-Tn-seq in a $\Delta$Vi capsule background to increase intracellular bacteria.

We used agglomerative clustering to generate a heatmap from these macrophage experiments, retaining all genes with significant fitness effects (P < 0.05) in at least one serovar (Supplementary Fig. 11). This heatmap revealed that Typhi and Paratyphi A cluster together while Typhimurium ST4/74 and D23580 cluster together (Supplementary Fig. 11), suggesting that the genes involved in macrophage infection exhibit greater similarity within typhoidal strains and

nontyphoidal isolates, respectively. This heatmap captured several SPI-related gene clusters (Supplementary Table 7). Tn insertions in SPI-2 genes displayed decreased fitness in *S*. Typhimurium ST4/74, D23580 and *S*. Typhi Ty2 (Supplementary Table 7 and Supplementary Fig. 12a), consistent with previous findings highlighting the critical role of SPI-2 in intracellular *S*. Typhimurium[28,29,65,68] and *S*. Typhi proliferation[69]. Intriguingly, no significant fitness defects in SPI-2 genes were observed in *S*. Paratyphi A 9150. This might suggest that Paratyphi A does not rely solely on SPI-2 for intracellular survival[69], or could reflect the low intracellular biosynthetic capacity of *S*. Paratyphi A[70]. Mutations in the Typhimurium-specific effectors *sseK1* and *sseK3* led to decreased fitness in ST4/74 and D23580 within macrophages (Supplementary Data 14). Mutations in the SPI-1 encoded *sitABCD* Fe/Mn import system decreased Typhi and Paratyphi fitness, suggesting high sensitivity of typhoidal *Salmonella* to intracellular perturbations in Fe/Mn pools (Supplementary Data 14). Similarly, mutations in the SPI-3 encoded Mg$^{2+}$ importers *mgtB* and *mgtC* decreased *S*. Paratyphi A 9150 fitness during macrophage infection (Supplementary Data 14).

Beyond SPI-genes, Tn insertions in several two-component signaling genes (for example, *phoPQ*, *envZ/ompR*, *arcAB*) and redox-related genes (for example, *trxA*, *trxB*, *sodA*, *oxyR*) decreased fitness during macrophage infection (Supplementary Data 14). In contrast, increased fitness was observed with Tn insertions in *fepE* in *S*. Paratyphi A 9150 (Supplementary Data 14). The regulation of very long O-antigen chains by FepE in Paratyphi[71,72] indicates a potential effect of this modified LPS structure during macrophage uptake, akin to the role of Vi capsule in reducing Typhi phagocytosis[68]. Tn insertions in many LPS genes increased fitness in *S*. Typhimurium ST4/74 and D23580, consistent with previous studies highlighting enhanced invasiveness of O-antigen deficient *S*. Typhimurium (Supplementary Fig. 12b and Supplementary Table 7)[29]. Conversely, mutations in these O-antigen genes decreased Typhi fitness, emphasizing the importance of the O-antigen layer in shielding typhoidal isolates from membrane-disrupting stresses. For example, Ty2$^{\Delta rfbE}$, lacking its O-antigen tyvelose moiety (Fig. 2f), exhibited reduced fitness within macrophages (Supplementary Fig. 13a). Intriguingly, Tn insertions in chemotaxis genes (for example, *cheARWY*, *tar*) increased fitness in all four isolates (Supplementary Fig. 12c and Supplementary Table 7).

To identify serovar-specific fitness changes during macrophage infection, we correlated the fitness profiles between *S*. Typhimurium ST4/74 and the other isolates; the highest correlation was observed between the two Typhimurium isolates (R = 0.6; Supplementary Fig. 14), followed by Typhi (R = 0.5; Fig. 5b) and Paratyphi A (R = 0.4; Fig. 5c). We identified dozens of genes with typhoidal-specific changes in fitness, defined as genes that had FC greater than fourfold higher in *S*. Typhi Ty2 (Supplementary Table 8) or *S*. Paratyphi A 9150 (Supplementary Table 9) compared with *S*. Typhimurium ST4/74. For example, Tn insertions in *yfeX*, encoding a putative iron-dependent peroxidase,

**Fig. 5 | Serovar-specific fitness changes during macrophage infection.**
**a**, Schematic of macrophage infections, showing opsonization, cell infection, lysis and passaging. **b**,**c**, Pearson's correlation (R) of gene fitness changes between the *S*. Typhimurium ST4/74 and *S*. Typhi Ty2 (**b**) or *S*. Paratyphi A 9150 (**c**) macrophages Rb-Tn-seq experiments, with serovar-specific changes in fitness colored according to functional class. Blue, metabolic genes; orange, transporters; green, core process genes (for example, replication, transcription, translation); burgundy, regulators; pink, hypothetical genes; gold, virulence genes; light blue, metal/redox homeostasis genes; red, LPS-related genes; dark purple, all other genes. Data are shown as a density scatterplot, where colors range from dark purple (low) to yellow (high), representing the kernel density estimation from low to high density. **d**, Heatmap showing the fitness values of Tn insertions in *yfeX* across 24 stress conditions for *S*. Typhimurium ST4/74 and *S*. Typhi Ty2. Color gradient is derived from the log$_2$(FC) from each condition in the Rb-Tn-seq experiments. **e**, Growth curves of Ty2$^{WT}$ (black) and Ty2$^{\Delta yfeX}$ (blue) when exposed to 250 µM hydrogen peroxide, with reads taken at OD$_{600}$

once every 10 min. Each point and error bar indicates the mean ± s.e.m. of OD$_{600}$, derived from n = 3 biologically independent experiments. **f**, Normalized intracellular Ty2$^{WT}$ and Ty2$^{\Delta yfeX}$ bacterial counts recovered after 5 h of infection of LPS-activated THP-1 cells (P = 4.03 × 10$^{-5}$). **g**, Heatmap showing the fitness values of Tn insertions in metal homeostasis genes during THP-1 infection by *S*. Typhimurium ST4/74 and *S*. Typhi Ty2. Color gradient is derived from the log$_2$(FC) from each condition in the Rb-Tn-seq experiments. **h**, Schematic of Mn and Zn homeostasis systems in *Salmonella*. **i**, Normalized intracellular Ty2$^{WT}$ and Ty2$^{\Delta mntR}$ bacterial counts recovered after 5 h of infection in LPS-activated THP-1 cells (P = 2.05 × 10$^{-7}$). **j**, Normalized intracellular Ty2$^{WT}$ and Ty2$^{\Delta zntR}$ bacterial counts after 5 h of infection in LPS-activated THP-1 cells (P = 0.00259). For all macrophage CFU plots (**f**, **i** and **j**), bars indicate the mean ± s.e.m. of the normalized bacterial count recovered from macrophages, derived from n = 6 biologically independent experiments (**f**), n = 7 biologically independent experiments (**i**) and n = 4 biologically independent experiments (**j**). Significance was calculated using a two-tailed t-test; *P < 0.01. Created with Biorender.com.

strongly reduced fitness within THP-1 cells for *S.* Typhi Ty2 and *S.* Paratyphi A 9150 (~120× and ~30× lower, respectively) and moderately for *S.* Typhimurium ST4/74 (~4×) (Fig. 5d). Accordingly, deleting this putative peroxidase in *S.* Typhi (Ty2$^{\Delta yfeX}$) increased sensitivity to 250 µM H$_2$O$_2$ (Fig. 5e), and impaired survival in LPS-stimulated THP-1 macrophages compared with Ty2$^{WT}$ (Fig. 5f and Supplementary Fig. 13b).

Tn insertions in genes involved in manganese (Mn) and zinc (Zn) homeostasis induced stronger fitness defects in *S.* Typhi during macrophage infection than in the other serovars (Fig. 5g–h). To confirm these results, we deleted *mntR* and *zntR* in *S.* Typhi Ty2 and observed that these mutants survived worse within LPS-activated THP-1 macrophages compared with the wild-type (Fig. 5i–j and Supplementary

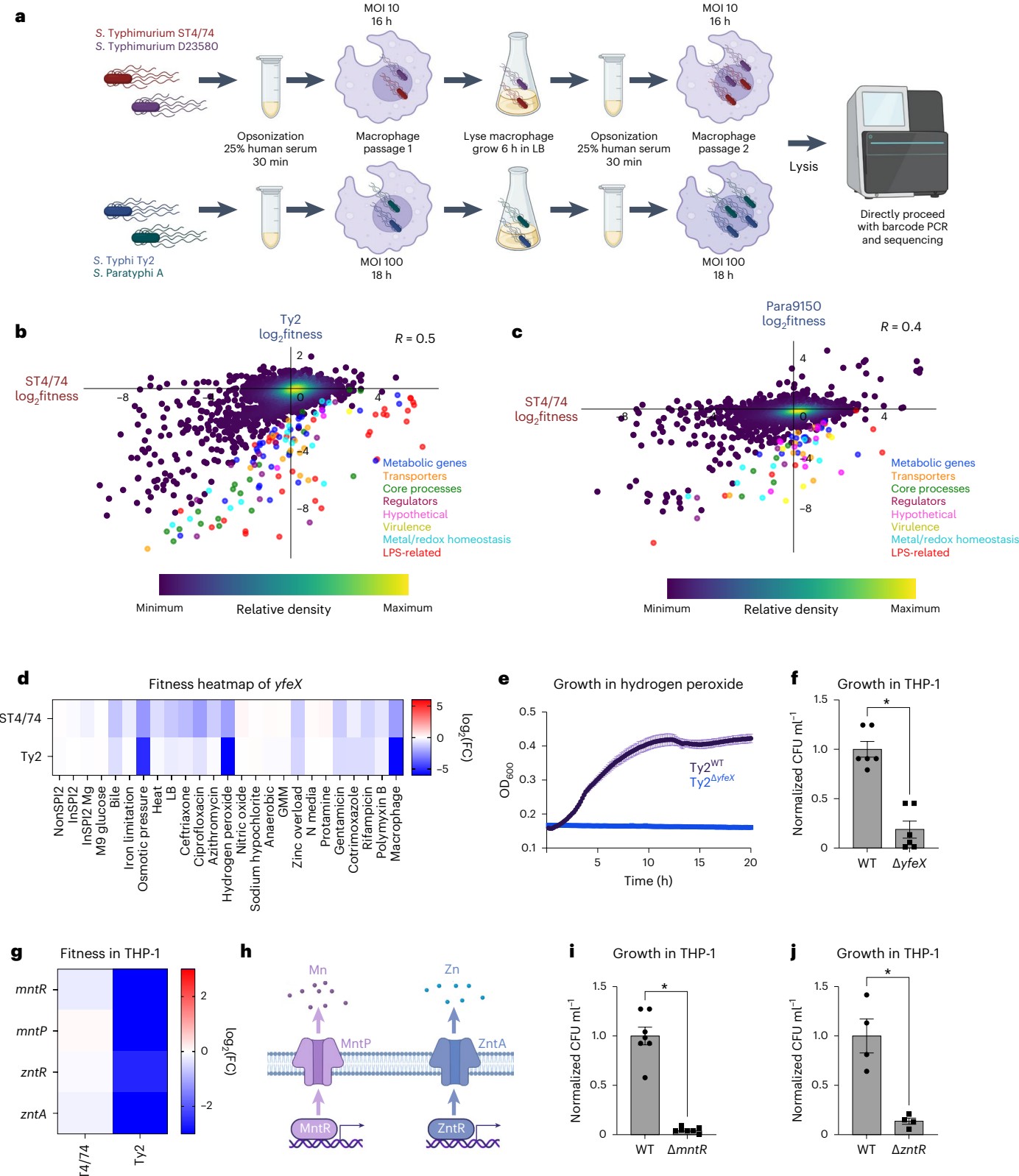

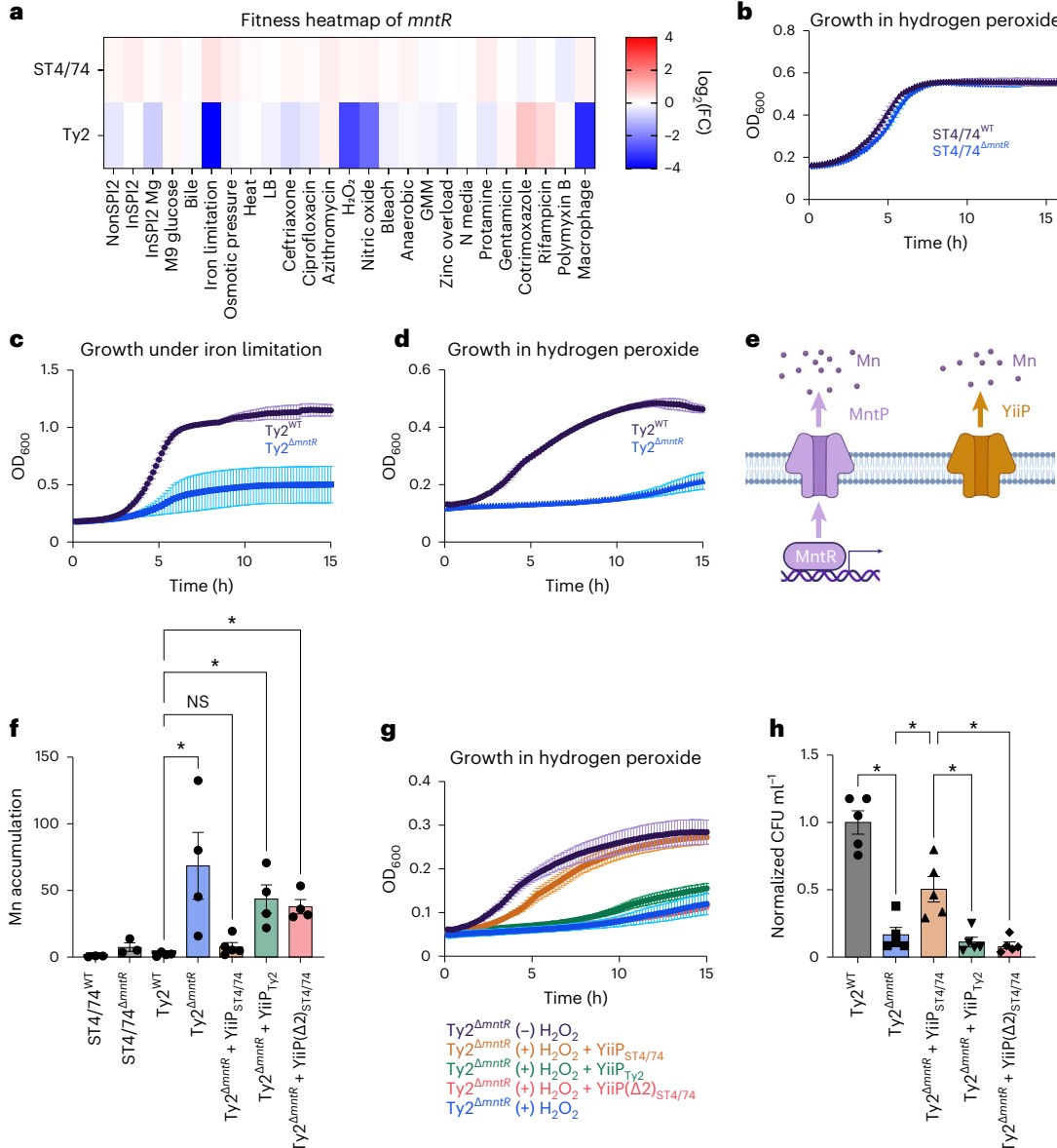

**Fig. 6 | YiiP pseudogenization sensitizes *S*. Typhi to stress. a**, Fitness heatmap of *mntR* Tn insertions for *S*. Typhimurium ST4/74 and *S*. Typhi Ty2. Color gradient is derived from the log$_2$(FC) for each condition from Rb-Tn-seq. **b**, Growth of ST4/74$^{WT}$ (black) and ST4/74$^{\Delta mntR}$ (blue) when exposed to 250 μM H$_2$O$_2$, derived from *n* = 4 biologically independent experiments. **c**, Growth of Ty2$^{WT}$ (black) and Ty2$^{\Delta mntR}$ (blue) when exposed to 100 μM 2,2′-dipyridyl, derived from *n* = 4 biologically independent experiments. **d**, Growth of Ty2$^{WT}$ (black) and Ty2$^{\Delta mntR}$ (blue) when exposed to 250 μM H$_2$O$_2$, derived from *n* = 3 biologically independent experiments. **e**, Schematic of Mn efflux systems in *Salmonella*. The MntR/MntP system is shown in purple, while YiiP is in orange. **f**, ICP-OES of Mn accumulation in WT and mutant strains of *S*. Typhimurium ST4/74 and *S*. Typhi Ty2 under Mn challenge, shown as the ratio of accumulated Mn in the (+) versus (−) Mn challenged samples, normalized to viable cell count. Each point and error bar indicates the mean ± s.e.m. of normalized Mn and is derived from *n* = 3 (ST4/74 strains), *n* = 4 (Ty2$^{WT}$, Ty2$^{\Delta mntR}$, Ty2$^{\Delta mntR}$+YiiP$_{Ty2}$, Ty2$^{\Delta mntR}$+YiiP(Δ2)$_{ST4/74}$) and *n* = 5 (Ty2$^{\Delta mntR}$+YiiP$_{ST4/74}$) biologically independent experiments. **g**, Growth

curves of Ty2$^{\Delta mntR}$ and complementation strains of Ty2$^{\Delta mntR}$ under 250 μM H$_2$O$_2$. For controls, Ty2$^{\Delta mntR}$ grown without H$_2$O$_2$ is in black, while Ty2$^{\Delta mntR}$ growth with H$_2$O$_2$ is in blue, both derived from *n* = 3 biologically independent replicates. For complementation curves, YiiP(Δ2)$_{ST4/74}$ is in red (*n* = 5 biologically independent replicates), while YiiP$_{ST4/74}$ is in orange and YiiP$_{Ty2}$ is in green, both derived from *n* = 7 biologically independent replicates. **h**, Normalized intracellular bacterial counts recovered from LPS-activated macrophages, with Ty2$^{WT}$ in gray and Ty2$^{\Delta mntR}$ in blue. For complementation strains, YiiP$_{ST4/74}$ is in orange, YiiP$_{Ty2}$ is in green and YiiP(Δ2)$_{ST4/74}$ is in red. Bars indicate the mean ± s.e.m. of the normalized bacterial count recovered from macrophages with individual values shown, derived from *n* = 5 biologically independent experiments. For all growth curves (**b–d**, **g**), each point and error bar indicates the mean ± s.e.m. of OD$_{600}$, with reads taken at OD$_{600}$ once every 10 min. For **f** and **h**, significance was calculated using a one-way ANOVA; *\*P* < 0.05, with multiple comparisons corrected by the Benjamini, Krieger and Yekutieli method (Supplementary Table 10).

Fig. 13c,d), indicating heightened sensitivity to changes in intracellular Mn$^{2+}$ and Zn$^{2+}$ levels in human-restricted *Salmonella*.

## A typhoid-specific phenotype for *mntR* under stress

We sought to understand why *S*. Typhi Ty2$^{\Delta mntR}$ has decreased intra-macrophage survival. Tn insertions in *mntR* rendered *S*. Typhi sensitive

to iron limitation, H$_2$O$_2$ and nitric oxide (NO) (Fig. 6a)—all of which all encountered within macrophages[73–75]. In contrast, Tn insertions in *mntR* had no impact on the fitness of other serovars (Fig. 6a and Extended Data Fig. 4a). Deleting *mntR* in both *S*. Typhimurium ST4/74 and *S*. Typhi Ty2 validated these results; ST4/74$^{\Delta mntR}$ did not have a growth defect with H$_2$O$_2$ (Fig. 6b), whereas Ty2$^{\Delta mntR}$ exhibited marked

growth defects under iron restriction (Fig. 6c) and $H_2O_2$ (Fig. 6d), confirming its serovar-specific phenotype. Deleting *mntR* in six additional *S.* Typhi strains sensitized each strain to $H_2O_2$, indicating conservation of this phenotype across clinical Typhi isolates (Extended Data Fig. 4b–g).

We next explored why Ty2$^{\Delta mntR}$ is sensitive to infection-related stress. MntR—a Mn-responsive transcription factor—activates MntP, a Mn efflux pump, to restore homeostasis[76,77]. Nontyphoidal and typhoidal *Salmonella* also encode YiiP—a constitutively active Mn pump[78,79] (Fig. 6e). While MntP sequences are identical between *S.* Typhimurium and *S.* Typhi, Ty2 YiiP has a two amino acid deletion ($\Delta$L95–F96) found in all Typhi isolates on BioCyc (Extended Data Fig. 4h), suggesting that YiiP$_{Ty2}$ may not be fully functional. Accordingly, we challenged WT and $\Delta mntR$ mutants of *S.* Typhimurium ST4/74 and *S.* Typhi Ty2 with 200 µM manganese and quantified intracellular Mn accumulation. Neither WT ST4/74 nor WT Ty2 accumulated Mn, indicating functional Mn efflux (Fig. 6f). The ST4/74$^{\Delta mntR}$ modestly accumulated Mn (around fourfold), suggesting that YiiP removes most intracellular Mn even without MntR in this isolate (Fig. 6f). In contrast, Ty2$^{\Delta mntR}$ accumulated high Mn (~70-fold), indicating severely impaired Mn efflux without MntR, likely due to a nonfunctional YiiP efflux pump (Fig. 6f).

To investigate whether mutated YiiP underlies the fitness defects of Ty2$^{\Delta mntR}$, we expressed functional YiiP from *S.* Typhimurium in Ty2$^{\Delta mntR}$; this complemented strain no longer accumulated Mn during Mn challenge (Fig. 6f) or exhibited strong sensitivity to $H_2O_2$ (Fig. 6g). In contrast, expressing YiiP from Ty2 in the Ty2$^{\Delta mntR}$ strain still resulted in high Mn accumulation during Mn challenge (Fig. 6f) and sustained sensitivity to $H_2O_2$ (Fig. 6g), further indicating that YiiP in *S.* Typhi is likely nonfunctional. Similarly, introducing the mutated YiiP$_{ST4/74}$ pump (YiiP($\Delta$2)$_{ST4/74}$), with an in-frame deletion of L95–F96, into Ty2$^{\Delta mntR}$ failed to correct growth defects under Mn or $H_2O_2$ stress (Fig. 6f–g). Moreover, deleting either full-length *yiiP* or just the L95–F96 sequence of this pump in ST4/74$^{\Delta mntR}$ sensitized ST4/74 to both Mn and $H_2O_2$ stress (Extended Data Fig. 4i–j), indicating that YiiP is both necessary and sufficient for *Salmonella* survival to macrophage-associated stresses. Importantly, expressing YiiP$_{ST4/74}$ rescued Ty2$^{\Delta mntR}$ survival in LPS-activated THP-1 macrophages to a significantly higher extent than expressing *yiiP*$_{Ty2}$ or mutated *yiiP*$_{(\Delta2)ST4/74}$ (Fig. 6h and Extended Data Fig. 4k). Collectively, our results indicate that *yiiP* pseudogenization in human-adapted *Salmonella* leads to a vulnerability during macrophage infection.

## Discussion

Decades of genomics research have revealed extensive genetic diversity in *Salmonella*, yet a functional understanding of its genetic evolution remains unclear. Here, we performed hundreds of high-throughput fitness assays with representative nontyphoidal and typhoidal *Salmonella* serovars, capturing thousands of fitness events across 25 infection-relevant conditions. We characterized serovar-specific fitness profiles and gene networks using a comprehensive systems biology approach and experimentally perturbed gene networks to pinpoint specific pseudogenes in typhoidal *Salmonella* contributing to serovar-specific fitness effects. Overall, these findings advance our functional understanding of how genetic differences between generalist and host-restricted *Salmonella* correlate with serovar behavior.

Our functional genomics approach has experimentally identified typhoid-specific phenotypes not detected in previous bioinformatic-based genomic comparisons[7,9–11]. Unlike traditional pseudogene identification relying on early stop codons and frameshift mutations[7], our approach identified putative pseudogenes with missense mutations or small, internal inframe deletions that likely disrupt gene function. Examples include *cysIJ* and *yiiP*, which contribute to the fitness defects of Ty2$^{\Delta metQ}$ and Ty2$^{\Delta mntR}$ under RCS and macrophage-associated stresses, respectively. Our systematic identification of pseudogenes hints at a potentially higher number of nonfunctional genes in host-adapted *Salmonella* than currently estimated.

Our results further indicate that pseudogenes arise in redundant pathways. For instance, *Salmonella* acquires Met/Cys through high-affinity transporters or de novo synthesis, processes redundant in *S.* Typhimurium[58]. However, *S.* Typhi cannot synthesis Cys/Met endogenously, heightening its sensitivity to RCS. Similarly, *Salmonella* encodes ISC and SUF complexes for iron-sulfur cluster synthesis[80,81], but SUF pseudogenization in Paratyphi increases sensitivity to iron restriction. Additionally, *Salmonella* encodes two Mn efflux pumps, MntP and YiiP, that mediate Mn homeostasis[78]. However, YiiP is likely nonfunctional in Typhi, sensitizing Typhi to macrophage-associated stresses. Overall, typhoidal *Salmonella* tolerates pseudogenization in redundant pathways, but accumulation of these mutations can sensitize these isolates to infection-related stress.

In summary, our Rb-Tn-seq screens create a genome-wide atlas for both generalist and human-restricted *Salmonella*, providing a critical public resource for further mechanistic understanding of how diverse serovars withstand human infection stresses. Although our study focuses on four representative isolates, we acknowledge the vast genetic landscape of this pathogen. Expanding these screens to encompass more genetically diverse *Salmonella* will lead to additional insights into how isolates respond to infection-related stress or sublethal doses of antibiotics. Furthermore, uncovering additional pseudogenes in typhoidal *Salmonella* will highlight vulnerabilities within the genomes of these human-restricted pathogens. Ultimately, these efforts will pinpoint genes and pathways that could serve as targets for rational drug design against *Salmonella*-related illnesses.

## Online content

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

## Methods

### Ethics declaration

The authors have complied with all ethics guidelines and have no competing interests to declare. All study protocols were approved by Stanford University.

### Experimental model and subject details

*Salmonella* Typhimurium ST4/74 and D23580, *Salmonella* Typhi Ty2 and *Salmonella* Paratyphi A 9150 were utilized for Rb-Tn-seq experiments. Both *S.* Paratyphi A 9150 and *S.* Typhi Ty2 are laboratory-adapted isolates, and *S.* Typhi Ty2 is an RpoS(−) strain that is attenuated in virulence[82]. For all cloning experiments, the conjugative *E. coli* strain Jke201 was used to move plasmids into *Salmonella*[83], and *E. coli* strain DH5α was used for storage and sequencing of plasmids. For additional experiments in clinical isolates, the published H58 isolates ISP-04-06979, ISP-03-07467, E03-9804, ISO (98 S) and E03.4983 were used[59], which display chloramphenicol, ampicillin and tetracycline resistance. For all experiments and cloning, *Salmonella* and *E. coli* strains were grown overnight in Luria-Bertani (LB) medium at 37 °C under shaking conditions before subculturing into specific stress conditions (see below). When applicable, antibiotics were added at the following concentrations for all strains: 50 µg ml⁻¹ kanamycin, 20 µg ml⁻¹ gentamicin and 100 µg ml⁻¹ carbenicillin. Strains, plasmids and primers used are listed in Supplementary Tables 1–3. For macrophage experiments, THP-1 human macrophages (ATCC TIB202) were grown routinely in RPMI medium supplemented with 10% FBS and 2 mM Glutamax and incubated at 37 °C and 5% $CO_2$. Cell cultures were passaged every ~3–5 days and passages between 4 and 9 were used for Rb-Tn-seq. Cells were genotyped at ATCC with STR profiling. Mycoplasma testing was done routinely by PCR every 3–6 months.

### General cloning procedures

Chromosomal-based modifications in *Salmonella* were done with allelic exchange, as previously described[83]. Briefly, the pFOK and pFOG vectors were utilized, which confer kanamycin and gentamicin sensitivity, respectively. These vectors also encode sucrose and anhydrotetracycline (AHT) sensitivity for counterselection. For all deletions, fragments ~500–1,000 bp upstream and downstream of the target gene were amplified from *Salmonella* genomic DNA (gDNA) using a KAPA Hifi PCR kit (Roche). Each fragment contained ~20–30-bp overlap regions. In turn, all vectors were assembled by HiFi Gibson Assembly (NEB) and transformed into DH5α *E. coli* for storage. Plasmids with the correct sequences were miniprepped and transformed into JKe201 *E. coli* on LB agar plates supplemented with 300 µM 2,6-diaminopimelic acid (DAP) and appropriate antibiotics.

For conjugation into *Salmonella*, JKe201 *E. coli* and relevant *Salmonella* strains were grown overnight for 16 h. The next morning, 500 µl of *E. coli* and *Salmonella* were mixed, centrifuged and resuspended in 50 µl LB + DAP. This mixture was then pipetted onto the center of a LB + 300 µM DAP plate and grown at 30 °C for mating to proceed. Mating times were 16 h for *S.* Typhimurium ST4/74 and D23580, 3 h for *S.* Typhi Ty2 and 6 h for *S.* Paratyphi A 9150. The mixtures were then scraped and resuspended in 1 ml PBS. Then, 100 µl of these mixtures were plated on LB supplemented with either Kan (for pFOK) or gent (for pFOG). Single colonies were streaked onto plain LB plates supplemented with 15% sucrose and 0.5 µg ml⁻¹ AHT for counterselection. Colonies were screened for chromosomal modifications using colony PCR and sanger sequencing; clones with the correct constructs were collected and stored at −80 °C.

The low-copy pWSK29 plasmid was used to generate complementation constructs[84]. Briefly, inserts of interest were PCR amplified from *Salmonella* gDNA using the KAPA Hifi PCR kit (Roche), and designed such that they contained ~20–30 bp of overlap with the multicloning site (MCS) of the pWSK29 plasmid. A ~500-bp upstream region was included for each gene to capture the native promoter. For plasmid

assembly, the pWSK29 was digested using the EcoRV-HF restriction enzyme (NEB), according to manufacturer protocols. Digested pWSK29 and PCR-amplified sequences were then assembled using the HiFi Gibson Assembly kit (NEB) and transformed into DH5α *E. coli* for plasmid storage and whole plasmid sequencing, as described above. Sequence verified plasmids were then electroporated into electrocompetent *Salmonella* strains. A 1 µl sample of assembled vector was then electroporated into 100 µl of competent *Salmonella*. Cells were allowed to recover for 1 h at 37 °C and then plated on LB agar plates supplemented with carbenicillin for overnight growth.

### Set-up of growth experiments

All growth experiments and Rb-Tn-seq screens were done in 24-well plates (Falcon), in which each well contained 1 ml of cells. Briefly, cultures were first grown overnight in LB at 37 °C with shaking for 16 h. The next morning, the overnight cultures were backdiluted 1:100 into fresh LB and grown for ~3 h at 37 °C with shaking until an OD of ~1 (input sample). Then, 1 ml of these 'time-zero' input samples were centrifuged at 8,000*g* for 3 min and frozen at −80 °C until further processing. Cultures were then backdiluted 1:50 into the 24-well plate format (20 µl into 1 ml of medium per well), and growth was monitored on a BioTek Synergy plate reader overnight with reads at $OD_{600}$ every 15 min, or until around six to eight doublings had been achieved (output sample). For these output samples, 1-ml cultures from the 24-well plates were collected by centrifugation at 8,000*g* for 3 min and then frozen at −80 °C until further processing. Log-transformed optical density plots of all growth curves in the main figures are shown in Supplementary Fig. 15 to show growth rates.

The NonSPI2 medium was composed of 80 mM MOPS (pH 7.4), 4 mM tricine, 376 µM $K_2SO_4$, 50 mM NaCl, 25 mM $K_2HPO_4$, 0.4% glucose, 15 mM $NH_4Cl$, 1 mM $MgSO_4$, 10 µM $CaCl_2$, 10 nM $Na_2MoO_4$, 10 nM $Na_2SeO_3$, 4 nM $H_3BO_3$, 300 nM $CoCl_2$, 100 nM $CuSO_4$, 800 nM $MnCl_2$, 1 nM $ZnSO_4$ and 100 µM $FeCl_3$ (ref. [37]). InSPI2 medium was the same as NonSPI2, except it contained 80 mM MES instead of 80 mM MOPS, was at pH 5.8 instead of 7.4 and contained 0.4 mM $K_2HPO_4$ instead of 25 mM $K_2HPO_4$. InSPI2 Mg was the same as InSPI2 except it contained 10 µM $MgSO_4$ instead of 1 mM $MgSO_4$. All redox stress experiments (for example, hydrogen peroxide, nitric oxide, sodium hypochlorite) were run in InSPI2 medium to mimic intracellular conditions. Protamine, which is insoluble in LB, was added to N medium[85], which contains 100 mM Tris-HCl (pH 7.4), 1 mM $MgCl_2$, 0.2% casamino acids and 0.2% glycerol. GMM[86] comprised 0.2% tryptone, 0.1% yeast extract, 2.2 mM glucose, 3.2 mM cysteine, 2.9 mM cellobiose, 2.8 mM maltose, 2.2 mM fructose, 0.5% meat extract, 100 mM $KH_2PO_4$, 0.008 mM $MgSO_4$, 4.8 mM $NaHCO_3$, 1.37 mM NaCl, 0.8% $CaCl_2$, 5.8 mM vitamin K, 1.4 mM $FeSO_4$, 0.1% histidine hematin, 0.05% tween 80, 1% ATCC vitamin mix, 1% ATCC trace mineral mix, 30 mM acetic acid, 1 mM isovaleric acid, 8 mM propionic acid and 4 mM butyric acid. All other stressors were run in LB. All stressor concentrations used for each serovar are listed in Supplementary Table 5.

### Preparation of barcoded Tn libraries

Overnight cultures (25 ml) of each *Salmonella* strain were grown in LB at 37 °C overnight with shaking for 14 h. Following the 14-h growth for Typhimurium strains, these cultures were incubated at 47 °C for an additional 2 h to increase conjugation efficiency, as has been done in other bacteria[87,88]. In parallel, 1 ml aliquots of previously constructed conjugative *E. coli* strains harboring barcoded transposon plasmids[36] were thawed and grown in 50 ml of LB + 300 µM DAP + 50 µg ml⁻¹ kanamycin for ~3 h, until the OD reached ~1. The donor *E. coli* and recipient *Salmonella* strains were then mixed at a ratio of two donor:one recipient in a total volume of 2 ml and centrifuged at 8,000*g* for 3 min. The pellet was resuspended in 100 µl of LB and placed on a 0.45 µm nitrocellulose filter on a LB + 300 µM DAP agar plate for conjugation. *S.* Typhimurium conjugations were allowed to proceed overnight, while *S.* Typhi Ty2 and

*S.* Paratyphi A conjugations proceeded for 3 and 5 h, respectively. The filters were then resuspended in LB and vortexed for 30 s to dislodge bacteria. Dilutions of this solution were plated onto 245 mm (Fluotics) square Petri plates with LB agar + 50 µg ml⁻¹ kanamycin and incubated overnight. Colonies were then scraped into 25 ml LB, grown for 1 h at 37 °C, and frozen as 1 ml glycerol stocks at −80 °C. A total of ~100–300 K colonies were collected for each library.

### Construction and sequencing of Tn-seq libraries

Tn libraries were prepared according to previously published protocols[36]. Briefly, gDNA from each library was extracted using the QIAamp DNA Mini Kit (Qiagen). gDNA was diluted to ~15 ng µl⁻¹ in 130 µl of nuclease-free water (Ambion) and sheared to an average size of 300 bp using the Covaris S2 ultrasonicator. AMPure XP beads (Beckman Coulter) were then used to size-select for 300-bp-sized fragments. Fragments were then subjected to end-repair and A-tailing using the KAPA Hyper Prep kit (Roche). Splinkerette adapters were ligated to these fragments to allow for the selection of barcode-containing fragments, and contaminants were removed using SPRI beads (Beckman Coulter). These fragments were then amplified for 15 cycles (98 °C for 15 s, 60 °C for 30 s, 72 °C for 30 s) and final extension at 72 °C for 60 s using the KAPA HiFi HotStart ReadyMix kit (Roche). In these reactions, one primer binds to the splinkerette and the other one binds to the transposon, thereby allowing for the selective amplification of sequences containing both a barcode and a transposon[36]. These amplified fragments were cleaned up with SPRI beads (Beckman Coulter) and then subjected to another round of PCR using the KAPA HiFi HotStart ReadyMix kit (Roche) for eight cycles (98 °C for 15 s, 60 °C for 30 s, 72 °C for 30 s and final extension at 72 °C for 60 s), using a universal P5 Illumina primer and a different indexed P7 Ilumina primer for each reaction[36]. SPRI-cleaned samples were sequenced on an Illumina HiSeq4000 instrument using paired-end sequencing (PE150) at Novogene. Typically, up to eight different Tn libraries were sequenced on one lane. A published perl script (MapTnSeq.pl) was used to analyze Tn-seq reads and identify the associated barcode for each Tn insertion[36]. Essential genes were mapped as previously described[89,90]. Briefly, we calculated the insertion index of each gene by dividing the number of unique Tn insertions for each gene by its length. We then plotted a histogram of this data, which showed a clear bimodal distribution (Supplementary Fig. 2). We fit these two peaks to gamma distributions using the scipy. stats function in python 3.8. Log-likelihood scores were then calculated for each gene; a gene that was 16× more likely to belong to the left-peak than the right-peak was considered essential.

### Preparation and sequencing of Rb-Tn-seq samples

Rb-Tn-seq fitness experiments were done in biological duplicate. gDNA from frozen pellets was extracted using the QIAamp DNA Mini Kit (Qiagen). The Q5 HiFi polymerase (NEB) was used to amplify the barcode regions associated with each Tn insertion; each barcode is flanked by universal primer binding sites. One primer (P5) is universal and binds to the upstream common priming site for Rb-Tn-seq, while the other primer (P7) has a unique index sequence that is different for each sample and binds to the downstream common priming site. The PCR was amplified for 25 cycles (98 °C for 15 s, 55 °C for 30 s, 72 °C for 30 s, final extension of 72 °C for 5 min). PCR samples were run on 1.2% agarose gels to verify the presence of bands (~200 bp) for each sample. Equal volumes of all samples (10 µl) were pooled and cleaned up using a PCR purification kit (Qiagen). Samples were then sequenced on an Illumina HiSeq4000 instrument using paired-end sequencing (PE150) at Novogene. Typically, up to 48 different Rb-Tn-seq experiments were sequenced on one lane. Mutant fitness was calculated using several published perl and R scripts (for example, Multicodes. pl and FEBA.R)[36]. The fitness of each strain is approximately equal to the normalized $\log_2$ ratio of counts for each Tn-associated barcoded between the initial 'time-zero' sample and the final sample collected after six to eight doublings in the presence of a stressor. The fitness of each gene is the weighted average of all strains within the central portion (10–90%) of that gene[36]. A previously described moderated *t*-like statistic ($|t| > 4$) was used to identify statistically significant genes; this statistic considers the consistency of fitness effects for all Tn insertions within a given gene[36]. To verify that this moderated *t*-statistic fits well to the standard normal distribution, we performed control comparisons between replicate time-zero samples for each of the four serovars used in this paper, as previously described[36]. Using a quartile–quartile (QQ) plot, we observed that the moderated *t*-statistic between these replicate time-zero samples is indeed distributed normally for each isolate (Supplementary Fig. 16).

For gene fitness, only insertions with the 10–50% and 50–90% of the gene region are used. In addition, genes without at least 15 time-zero reference reads are filtered out. The quality of each experiment was evaluated using a series of published metrics[36], including gMed (the median reads per gene in the sample), cor12 (measurement of the consistency of fitness data for each gene, taken by comparing the fitness of a gene using Tn insertions within the first half of the gene versus the second half of the gene using a Spearman rank correlation), mad12 (measurement of the mean absolute difference between fitness values from Tn insertions in the first versus second half of gene) and opcor (measurement of the consistency of fitness data for all genes within an operon).

### Cofitness analysis and SAFE

Cofitness analysis and SAFE were performed according to published pipelines[38]. Briefly, a gene × condition matrix was first assembled for each *Salmonella* serovar, consisting of all genes with captured $\log_2$ fitness effects in the Rb-Tn-seq experiments, and all 24 plate-based conditions tested. These matrices were then used to build a cofitness network, which calculates the Pearson's correlation coefficient for all gene pairs in the network[38]. We used a correlation cutoff of $R > 0.75$ for two reasons. First, this value was used in previous publications performing cofitness analysis and SAFE[38]. Second, we empirically tried several different cut-offs, ranging from 0.6 to 0.9, and conducted a manual analysis of the generated SAFE clusters, aiming to identify an optimal cutoff point that preserves genes expected to cluster together due to their related functionality while minimizing the clustering of genes with no functional association. Using $R > 0.75$ led to the retention of nodes and edges in the network between pairs of genes with similar fitness profiles. Statistical significance of each correlation value was determined using $t = r\sqrt{\frac{n-2}{1-r^2}}$, where *r* is the correlation and *n* is the sample size of 24 conditions. *P* values were calculated from the *t*-statistics in Python, and false discovery rate (FDR) was calculated using the Benjamini–Yekutieli method. We then performed an additional stability test according to published methods[38]. Briefly, we built a correlation matrix using partial data by repeatedly hiding random conditions and performed our correlation analysis with 20 out of the 24 possible conditions 100 times. This resulted in 100 binary matrices that we then summed up, with each possible correlation receiving a score between 0 and 100; we considered scores >75 to be 'stable,' in line with published work[38].

These networks were then visualized in Cytoscape using the edge-weighted spring layout, with the absolute correlation value used as the edge weight. Next, SAFE was used to identify functionally enriched clusters of genes on these networks[55]. SAFE attributes were assigned using available GO term-based functional annotations from the BioCyc database for each *Salmonella* serovar (Supplementary Data 9). The distance threshold was set to 2% of the map-weighed distance, and the Jaccard similarity index was set to 0.5.

### RNA extraction and qRT-PCR

RNA was purified using previously published methods[91]. Briefly, 200 µl of cells were pelleted at 12,000*g* for 2 min and resuspended in 300 µl

of Cell Lysis solution (Epicentre) with 2 µl of 50 µg µl⁻¹ proteinase K. This lysis solution was incubated for 30 min at 65 °C, chilled on ice for 5 min, and then mixed with 175 µl of MPC Protein Precipitation Reagent (Epicentre) to remove all proteins. Samples were then spun for 15 min at 15,000$g$ at 4 °C to remove insoluble precipitated proteins. The supernatant was collected and then mixed with 500 µl of isopropanol to precipitate nucleic acids. This mixture was centrifuged for another 15 min at 15,000$g$ at 4 °C, and the resulting pellet was washed with 70% ethanol twice to remove contaminants. The pellet was then air-dried for 5 min and resuspended in 50 µl of nuclease-free water (Ambion). Then, 1.5 µl of Turbo DNAse (Invitrogen) was then added for 30 min at 37 °C to remove contaminating DNA from the preparation. DNAse was removed using DNAse inactivation reagent (Invitrogen), and the resulting supernatant containing purified RNA was collected and placed at −80 °C for long-term storage.

cDNA for qRT-PCR experiments was prepared using the Proto-Script II reaction mix (NEB), as described previously[92]. Briefly, 500 ng of purified RNA was mixed with 2 µl of random hexamers, 2 µl of Protoscript enzyme and 10 µl of ProtoScript II reaction mix. Samples were then incubated at 25 °C for 5 min and then 42 °C for 1 h The Sybr Fast qPCR kit (Roche) was used to perform all qRT-PCR experiments. Briefly, 10 ng of cDNA was mixed with 300 nM of each primer, nuclease-free water, and 2× Sybr master mix (Roche), in a final volume of 10 µl. All qRT-PCR experiments were run in 384-well plate format on a LightCycler 480 system (Roche). A program of 95 °C for 10 min, followed by 40 cycles of 95 °C for 15 s, 60 °C for 30 s and 72 °C for 30 s was used. FC was calculated using the ddCT method, using *rpoD* as the reference gene.

### Rb-Tn-seq experiments in THP-1 macrophages

THP-1 infections were carried out according to published methods[69]. Briefly, for Rb-Tn-seq experiments, THP-1 cells were seeded at a density of 5 × 10⁷ cells on a large 150 × 15 mm Petri dish (Falcon). A final phorbol myristate acetate concentration of 5 ng ml⁻¹ was used for differentiation. Cells were allowed to rest for 48 h in fresh medium before *Salmonella* infection experiments. For infections, *S*. Typhimurium ST4/74 and D23580 were used at a multiplicity of infection (MOI) of 10, while *S*. Typhi Ty2 and *S*. Paratyphi A 9150 were used at an MOI of 100. Frozen 1 ml aliquots of each library were thawed in 50 ml of LB + kanamycin and were allowed to grow until OD ~0.5. Based on the desired MOI, appropriate volumes of these libraries were then centrifuged and resuspended in 200 µl of 25% human serum (MP Biomedicals) and incubated at 22 °C for 30 min. Opsonized solutions of bacteria were passed three times through a 25-gauge needle to separate clumps of cells and then allowed to infect differentiated THP-1 macrophages (passage 1). Following incubation at 37 °C for 1 h, cells were washed once with PBS and then incubated with THP-1 medium supplemented with 100 µg ml⁻¹ gentamicin for 1 h to kill extracellular bacteria. The plates were then washed again with PBS and incubated with THP-1 medium supplemented with 10 µg ml⁻¹ gentamicin for 16 h (*S*. Typhimurium strains) or 18 h (*S*. Typhi and *S*. Paratyphi A). The THP-1 macrophages were then lysed for 1 h at 37 °C with 2% saponin, and the resulting lysate was added to 5 ml of LB + kanamycin for passaging and allowed to grow at 37 °C for 6 h to increase the number of bacteria. These passaged libraries were then subjected to the same treatment as above, starting with opsonization of these libraries in 25% human serum, and allowed to infect a fresh plate of differentiated THP-1 macrophages (passage 2), using the same parameters as above.

At the end of the second passage, macrophages were again lysed with 2% saponin to release intracellular bacteria. Bacterial counts were measured routinely by plating serial dilutions of this lysate on LB + kanamycin plates; typical recovery from Rb-Tn-seq experiments ranged from ~3 × 10⁷ to 1 × 10⁸ colony forming units (CFU) per milliliter. Macrophage lysates were centrifuged to concentrate intracellular *Salmonella*, and these samples were processed directly using the MasterPure DNA/RNA extraction kit (Epicentre), which has been used

previously to extract nucleic acids from small numbers of bacteria[91]. Briefly, the cell pellets were resuspended in 300 µl of Cell Lysis solution (Epicentre) with 2 µl of 50 µg µl⁻¹ proteinase K and incubated for 1 h at 65 °C. These samples were then chilled on ice for 5 min and mixed with 175 µl of MPC Protein Precipitation Reagent (Epicentre) to remove all proteins. Samples were then spun for 15 min at 12,000$g$ at 4 °C, and the supernatant was recovered and mixed with 500 µl isopropanol to precipitate DNA. These samples were then spun down at 12,000$g$ for 15 min at 4 °C to pellet the DNA, which was then washed with 70% ethanol twice and resuspended in nuclease-free water at a final volume at 30 µl. Rb-Tn-seq using the Q5 polymerase was then used to amplify all barcodes captured from these samples, as described above. Changes in gene fitness and significance within macrophages were calculated using the DEseq2 method, as has previously been done for *Salmonella* macrophage Tn-seq experiments[29].

*Salmonella* mutant strain validation in THP-1 cells was performed in a similar manner as described above. Briefly, THP-1 cells were seeded on six-well plates, at a density of 1 × 10⁶ THP-1 cells per well. The same MOIs as above were used for these infection assays. To capture stronger phenotypes in these experiments, THP-1 cells were stimulated with 100 ng ml⁻¹ of LPS for 24 h, as previously described[93]. *Salmonella* strains were allowed to infect these activated THP-1 cells for 5 h using the same gentamicin protection assay described above. Macrophages were then lysed with 2% saponin, and serial dilutions of the lysate were plated on LB agar plates to calculate CFUs from WT and mutant *Salmonella* strains.

Sytox green and replication assays using the pFCcGI plasmid were performed on the Incucyte imaging platform (Satorius). For these assays, 96-well black plates with clear bottoms (Falcon) were used for seeding THP-1 cells, which were seeded at a density of 5 × 10⁴ cells per well using the same general seeding protocol as above. For cell death assays, Sytox Green was added at a final concentration of 20 nM to each well during the low gentamicin incubation step; cell death was then monitored by the green signal on the Incucyte with measurements once every hour. A final spike-in of 10% Triton-X 100 was used to kill all cells for a full lysis control. For pFCcGI-based experiments, bacteria were grown to OD ~0.5 in the presence of 0.2% arabinose to induce GFP production. Infections were then allowed to proceed in the same way as described above, with imaging for both GFP and mCherry signals on the Incucyte once an hour.

### Inductively coupled plasma-optical emission spectroscopy

Overnight cultures were diluted 1:100 into fresh LB and grown until an OD ~0.5. Cells were then challenged with either 0 or 200 µM MnCl₂ for 2 h. After Mn challenge, cells were spun down at 8,000$g$ for 5 min and washed once in 10 mM HEPES + 2 mM EDTA (pH 7.5), followed by two washes in 10 mM HEPES (pH 7.5). Cell pellets were then dried for 3 h in a speed vac and resuspended in 600 µl of 30% v/v nitric acid (Sigma). These solutions were incubated at 95 °C for 1 h to release intracellular Mn, with vortexing every 15 min; 500 µl of these solutions were then diluted into 4.5 ml of 3% v/v nitric acid. All samples were filtered sterilized through a 0.22 µM filter before running on the inductively coupled plasma-optical emission spectroscopy (ICP-OES) instrument. Each ICP-OES run also included blank controls (3% v/v nitric acid) and serially diluted commercially available metal standards (Accustandard). All samples were run on a ICAP 6300 Duo View Spectrometer (Thermo Scientific), at the Stanford Environmental Measurements Facility. Intracellular Mn concentrations were normalized to viable bacterial counts, as previously described[78].

### Statistics and reproducibility

Two independent biological replicates were used for all Rb-Tn-seq experiments and exhibited a high degree of correlation (Supplementary Data 2). All other assays (for example, growth curves, ICP-OES, and so on) were done with at least three biologically independent

experiments, with exact *n* values indicated in each figure legend. Sample sizes were not predetermined, but all experiments were highly reproducible and our sample sizes were similar to previous related publications[36,38]. The experiments were not randomized. Data collection and analysis was not blinded. Pairwise statistical comparisons were done with two-sided *t*-tests, while multiple comparisons were done with one-way analysis of variance (ANOVA) with FDR correction, as described in each figure legend. No data were excluded from this study.

### Reporting summary

Further information on research design is available in the Nature Portfolio Reporting Summary linked to this article.

## Data availability

All strains from this study are available by request to the corresponding author. All raw sequencing data and processed fitness data in this study can be found on NCBI GEO at accession numbers GSE261860, GSE261867, GSE261873, GSE262768, GSE262769, GSE262848, GSE261757, GSE261749 and GSE261214. All raw FC and statistics from the dataset are in Supplementary Data 3, 4 and 14. All raw correlations for network analysis are in Supplementary Data 8. All source data for SAFE annotations are in Supplementary Data 9. All fitness changes from this work can be searched on an interactive website (https://bioinf.gen.tcd.ie/cgi-bin/salcomfit.pl).

## Code availability

Code used in this study is derived from previous publications[36,38], which we have deposited to Zenodo at https://doi.org/10.5281/zenodo.10963611 (ref. 94).

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

## Acknowledgements

We thank the members of the Monack laboratory for valuable discussions, especially D. Arve-Butler, B. Di Luccia and T. Pham. We thank A. Bhatt for insights into our paper. We thank A. Deutschbauer (University of California, Berkeley), D. Bumann (University of Basel) and D. Pickard (University of Cambridge) for generously providing strains and protocols. We thank T. Van Opijnen for helpful discussions about Tn-seq computational analysis. We thank Y. Le Guen at the Stanford Quantitative Sciences unit for help with coding. This work was supported by the grants R01-AI116059 and R01-AI095396 from the National Institute of Allergy and Infectious Diseases (to D.M.M.), the Paul Allen Stanford Discovery Center on Systems Modeling of Infection (to D.M.M.), the Maternal and Child Health Research Institute (to B.X.W.) and the A.P. Giannini Postdoctoral Fellowship (to B.X.W.). L.L. is funded by an EMBO fellowship (ALTF 95-2023). This work was supported in part by a Wellcome Trust Investigator Award (grant no. 222528/Z/21/Z) to J.C.D.H. For the purpose of open access, the authors have applied a CC BY public copyright license to any Author Accepted Manuscript version arising from this submission.

## Author contributions

B.X.W. and M.T. performed the experiments. B.X.W., L.L. and D.L. performed computational analysis. D.L. provided scripts used in this work. J.C.D.H. contributed to writing the paper. J.C.D.H. and K.H. created the interactive SalComFit browser associated with this work. B.X.W. and D.M.M. wrote the paper and conceived the ideas in this work.

## Competing interests

The authors declare no competing interests.

## Additional information

**Extended data** Extended data is available for this paper at https://doi.org/10.1038/s41588-024-01779-7.

**Correspondence and requests for materials** should be addressed to Denise M. Monack.

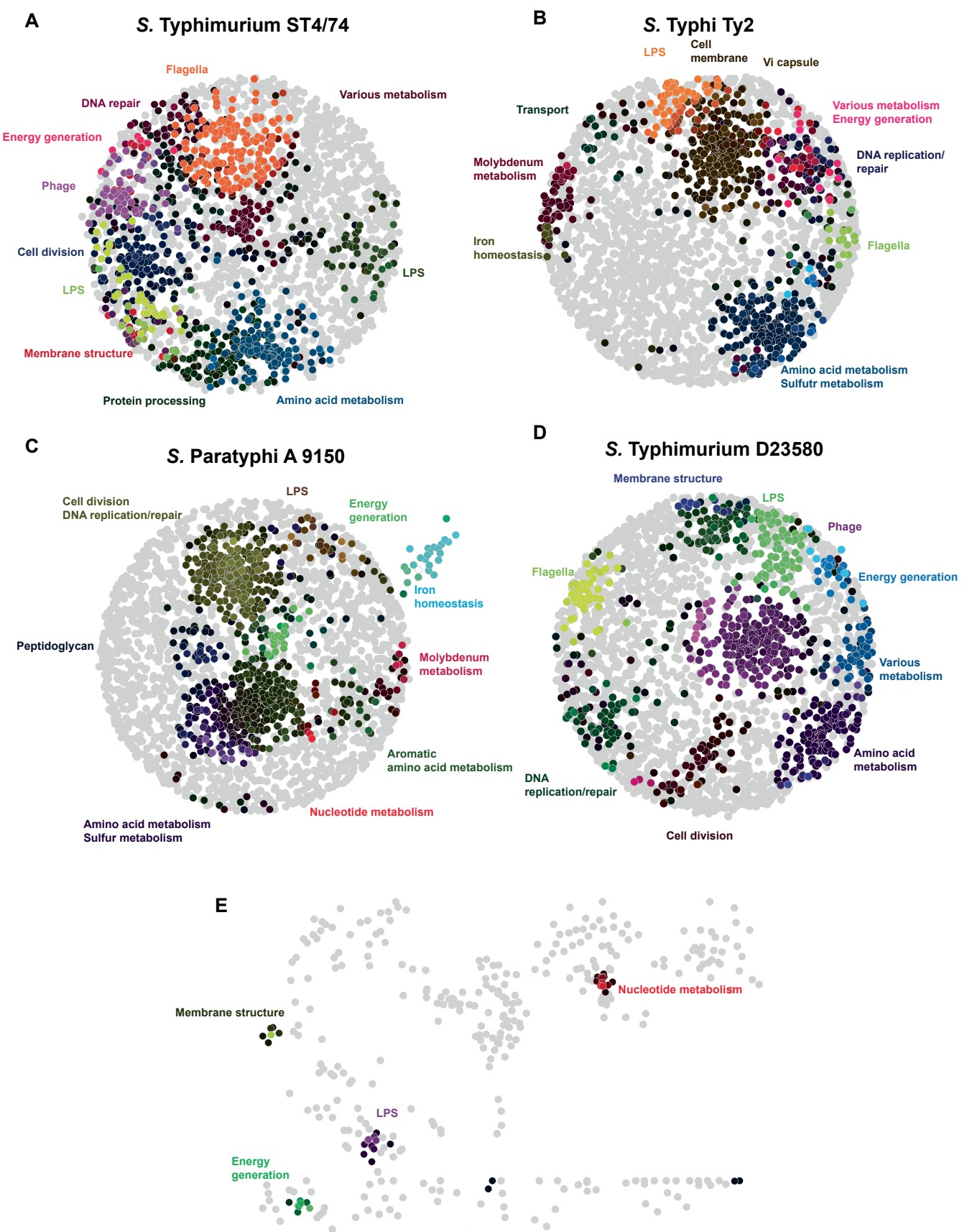

**Extended Data Fig. 1 | See next page for caption.**

**Extended Data Fig. 1 | Co-fitness and SAFE network analysis across four *Salmonella* isolates. a–d)** Co-fitness network analysis using r > |0.75| and SAFE for *S.* Typhimurium ST4/74 (**A**), *S.* Typhi Ty2 (**B**), *S.* Paratyphi A 9150 (**C**) and *S.* Typhimurium D23580 (**D**), with regions enriched in certain functional attributes highlighted in different colors, as indicated on the maps. **E)** Filtered network for *S.* Typhimurium D23580, which includes genes with fitness changes that are 1) significant (|t| > 4) in *S.* Typhimurium D23580 but not in *S.* Typhimurium ST4/74, and 2) >|2-fold| higher fitness change in *S.* Typhimurium D23580 compared to *S.* Typhimurium ST4/74. Colors indicate areas on these filtered maps that are enriched in certain functional terms through SAFE.

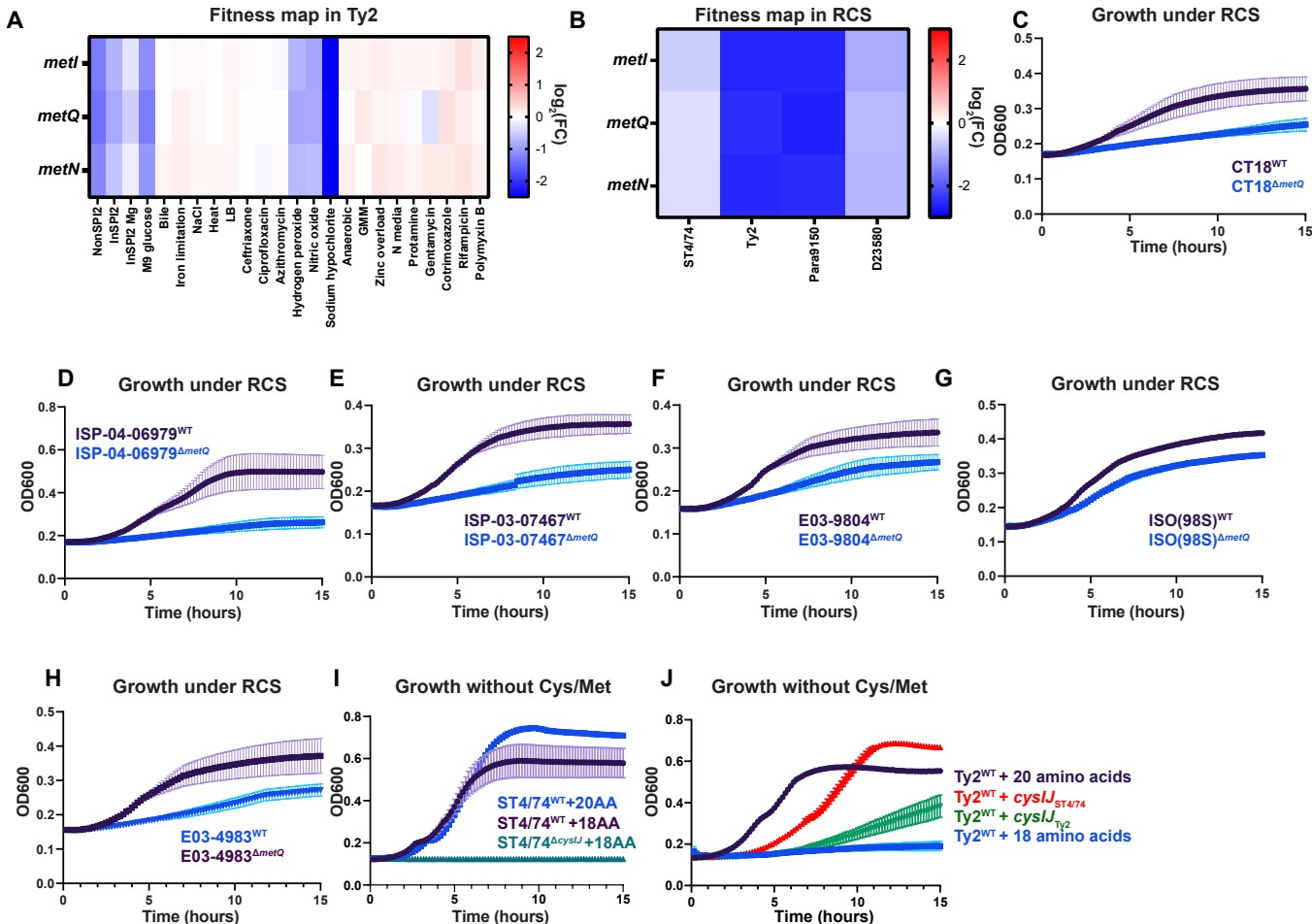

**Extended Data Fig. 2 | Tn insertions in *metIQN* lead to a typhoid-specific fitness effect during RCS. a)** Heatmap showing the fitness values of Tn insertions in *metIQN* across 24 stress conditions for *S.* Typhi Ty2. Color gradient is derived from the log$_2$(fitness) from each condition in the Rb-Tn-seq experiments. **b)** Heatmap showing the fitness values of Tn insertions in *metIQN* during sodium hypochlorite stress (RCS) for all isolates. Color gradient is derived from the log$_2$(fitness) from each condition in the Rb-Tn-seq experiments. **c–h):** Representative growth curves of CT18 **(C)**, ISP-04-06979 **(D)**, ISP-03-07467 **(E)**, E03-9804 **(F)**, ISO(98 S) **(G)**, E03-4983 **(H)**, with WT (black) and Δ*metQ* mutants (blue) in the presence of 12.5 μg/mL sodium hypochlorite, with reads taken at OD$_{600}$ once every 10 minutes, each derived from n = 3 biologically independent experiments. **i)** Growth curves of WT and mutant strains of Typhimurium ST4/74, with reads taken at OD$_{600}$ once every 10 minutes, derived from n = 3 biologically independent experiments. ST4/74$^{WT}$ growth in minimal media supplemented with all 20 amino acids is shown in blue, and ST4/74$^{WT}$ growth in minimal media supplemented with 18 amino acids lacking Cys/Met is shown in black. ST4/74$^{ΔcysIJ}$ growth in minimal media supplemented with 18 amino acids lacking Cys/Met is shown in green. **j)** Growth curves of WT and complementation strains in which the *cysIJ* locus from both *S.* Typhimurium ST4/74 (red) and *S.* Typhi Ty2 (green) is expressed in the WT *S.* Typhi Ty2 background, with reads taken at OD$_{600}$ once every 10 minutes, derived from n = 4 biologically independent experiments. Each complementation growth experiment was run in minimal media supplemented with 18 amino acids, but no Cys/Met. For controls, Ty2$^{WT}$ grown in minimal media supplemented with all 20 amino acids is shown in black, while Ty2$^{WT}$ grown in minimal media supplemented with 18 amino acids without Cys/Met is shown in blue. For these curves, reads were taken at OD$_{600}$ once every 10 minutes, and each point and error bar indicates the mean ± SEM of OD$_{600}$.

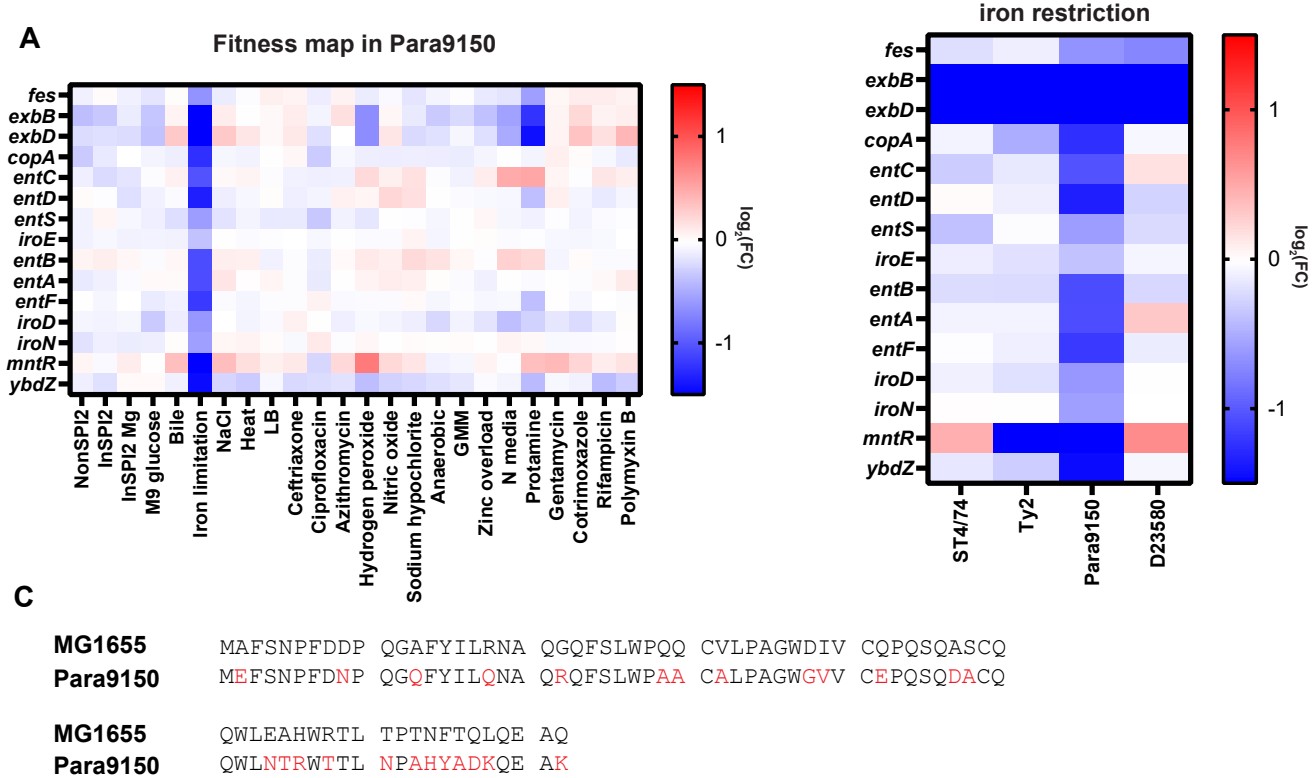

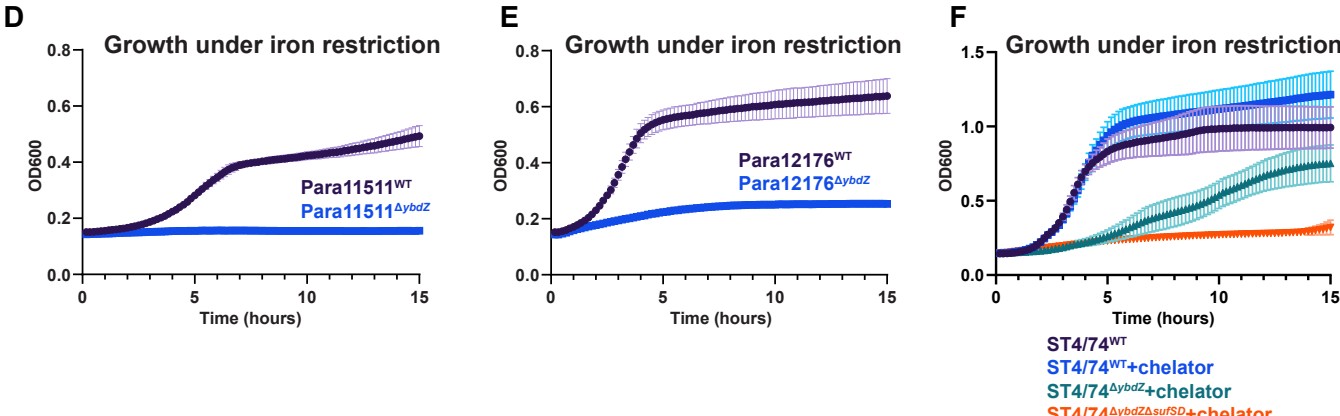

**Extended Data Fig. 3 | Tn insertions in *ybdZ* lead to a paratyphoid-specific fitness defect during iron limitation. a**) Heatmap of genes that comprise the sub-network of LPS-synthesis genes in Paratyphi A 9150 shown in Fig. 4a. Color gradient is derived from the log₂(fitness) from each condition in the Rb-Tn-seq experiments. All fitness values on these heatmaps were generated from the average of two biologically independent biological replicates. **b**) Heatmap showing the fitness values of Tn insertions in iron homeostasis genes during iron restriction for all isolates. Color gradient is derived from the log₂(fitness) from each condition in the Rb-Tn-seq experiments. **c**) Alignment of *ybdZ* from *E. coli* MG1655 and *S*. Paratyphi A 9150. Residues that are different in *S*. Paratyphi A are highlighted in red. **d-e**) Growth curves of *S*. Paratyphi A 11511 (**D**) and *S*. Paratyphi A 12176 (**E**) with WT (black) and Δ*ybdZ* mutants (blue) in the presence of 100 μM

2,2'-dipyridyl, with reads taken at OD₆₀₀ once every 10 minutes, derived from n = 3 biologically independent experiments. (**f**) Growth curves of WT and mutant strains in Typhimurium ST4/74. ST4/74^WT growth in LB without iron restriction is shown in black (n = 3 biologically independent experiments), and ST4/74^WT growth under iron restriction (100 μM 2,2'-dipyridyl) is shown in blue (n = 4 biologically independent experiments). ST4/74^Δ*ybdZ* growth in LB supplemented with 100 μM 2,2'-dipyridyl is shown in green (n = 6 biologically independent experiments). ST4/74^Δ*ybdZsufSD* growth in LB supplemented with 100 μM 2,2'-dipyridyl is shown in orange (n = 4 biologically independent experiments). For these curves, reads were taken at OD₆₀₀ once every 10 minutes, and each point and error bar indicates the mean ± SEM of OD₆₀₀.

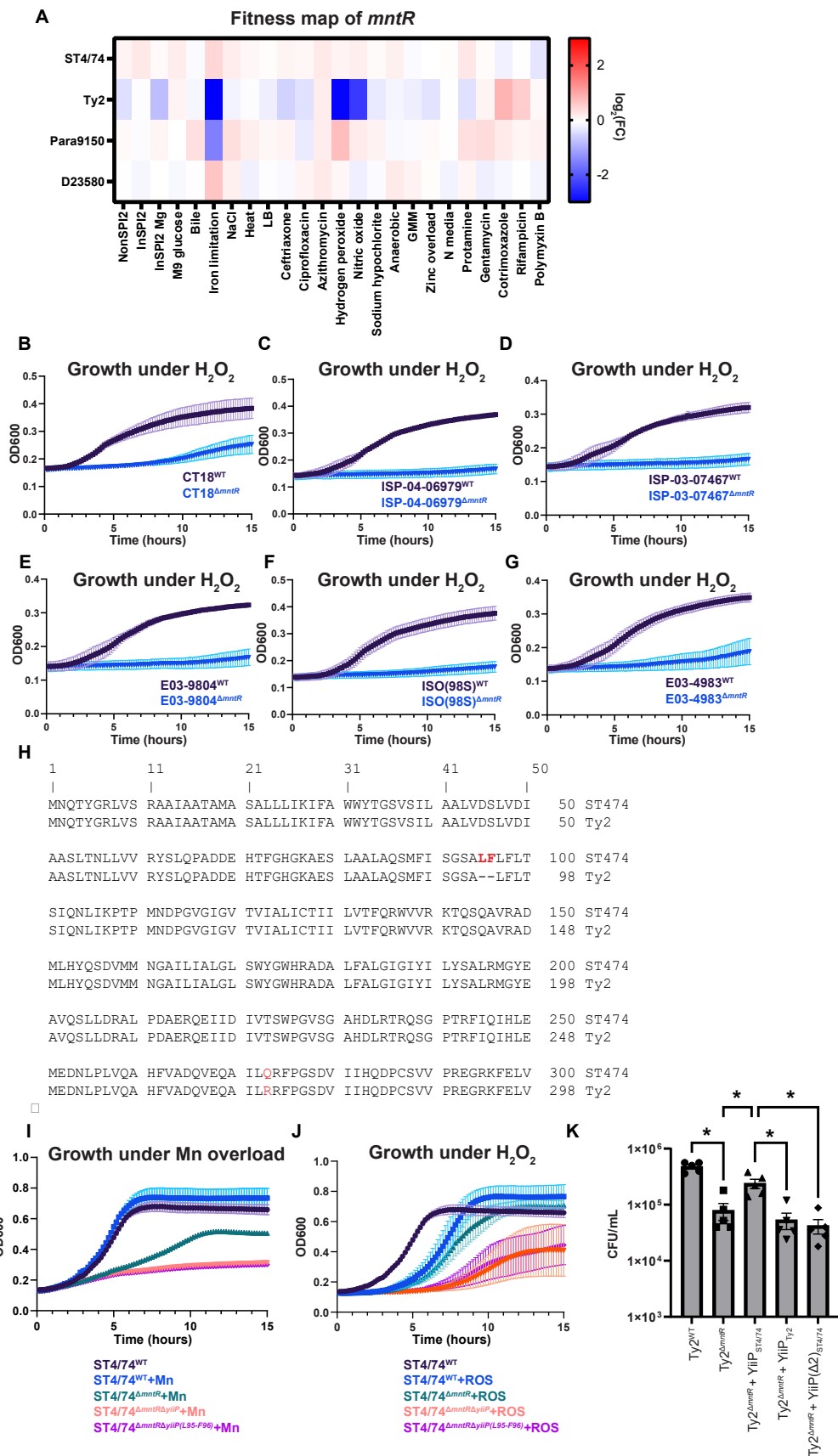

**Extended Data Fig. 4 | See next page for caption.**

**Extended Data Fig. 4 | Tn insertions in *mntR* lead to a typhoid-specific fitness defect during macrophage-associated stress. a**) Heatmap of *mntR* across 24 stresses. Color gradient is derived from the $\log_2$(fitness) from each condition in the Rb-Tn-seq experiments. All fitness values on these heatmaps were generated from the average of two biologically independent biological replicates. **b**–**g**): Growth of CT18 (**B**), ISP-04-06979 (**C**), ISP-03-07467 (**D**), E03-9804 (**E**), ISO(98 S) (**F**), E03.4983 (**G**), with WT (black) and Δ*mntR* mutants (blue) with 250 μM hydrogen peroxide, derived from n = 3 biologically independent experiments. **h**) Alignment of *yiiP* from *S.* Typhimurium ST4/74 and *S.* Typhi Ty2. Residues that are different in *S.* Typhi Ty2 are highlighted in red. **i**) Growth of WT and mutant strains in Typhimurium ST4/74 under Mn challenge. ST4/74^WT growth in LB without Mn challenge is shown in black and ST4/74^WT growth under Mn challenge (200 μM Mn) is shown in blue (both derived from n = 3 biologically independent experiments). ST4/74^Δ*mntR* growth in LB supplemented with 200 μM Mn is shown in green, and ST4/74^Δ*mntRyiiP* growth in LB supplemented with 200 μM Mn is shown in orange (both derived from n = 5 biologically independent experiments). ST4/74^Δ*mntRyiiP(L95-F96)* growth in LB supplemented with 200 μM

Mn is shown in purple (n = 6 biologically independent experiments). **j**) Growth curves of WT and mutant strains in Typhimurium ST4/74 under hydrogen peroxide challenge (ROS). ST4/74^WT growth in InSPI2 media without hydrogen peroxide is shown in black, and ST4/74^WT growth under hydrogen peroxide stress (600 μM) is shown in blue, both derived from n = 3 biologically independent experiments. ST4/74^Δ*mntR* growth under 600 μM hydrogen peroxide is shown in green (n = 4 biologically independent experiments). ST4/74^Δ*mntRyiiP* growth with 600 μM hydrogen peroxide is shown in orange and ST4/74^Δ*mntRyiiP(L95-F96)* growth under 600 μM hydrogen peroxide is shown in purple, both derived from n = 5 biologically independent experiments. **k**) Intracellular bacterial counts for Ty2^WT and Ty2^Δ*mntR* complementation strains recovered after 5 hours in LPS-activated THP-1 macrophages, derived from n = 5 independent biological experiments. Significance was calculated using a one-way ANOVA, with * indicating p < 0.05 and multiple comparisons corrected by the Benjamini, Krieger, and Yekutieli method; a list of exact *P* values is found in Supplemental Table 10. For all growth curves (**B-G, I-J**), reads were taken at $OD_{600}$ once every 10 minutes, and each point and error bar indicates the mean ± SEM of $OD_{600}$.

# Reporting Summary

## Statistics

For all statistical analyses, confirm that the following items are present in the figure legend, table legend, main text, or Methods section.

| n/a | Confirmed | |
|---|---|---|
| ☐ | ☒ | The exact sample size (*n*) for each experimental group/condition, given as a discrete number and unit of measurement |
| ☐ | ☒ | A statement on whether measurements were taken from distinct samples or whether the same sample was measured repeatedly |
| ☐ | ☒ | The statistical test(s) used AND whether they are one- or two-sided *Only common tests should be described solely by name; describe more complex techniques in the Methods section.* |
| ☒ | ☐ | A description of all covariates tested |
| ☐ | ☒ | A description of any assumptions or corrections, such as tests of normality and adjustment for multiple comparisons |
| ☐ | ☒ | A full description of the statistical parameters including central tendency (e.g. means) or other basic estimates (e.g. regression coefficient) AND variation (e.g. standard deviation) or associated estimates of uncertainty (e.g. confidence intervals) |
| ☐ | ☒ | For null hypothesis testing, the test statistic (e.g. *F*, *t*, *r*) with confidence intervals, effect sizes, degrees of freedom and *P* value noted *Give P values as exact values whenever suitable.* |
| ☒ | ☐ | For Bayesian analysis, information on the choice of priors and Markov chain Monte Carlo settings |
| ☒ | ☐ | For hierarchical and complex designs, identification of the appropriate level for tests and full reporting of outcomes |
| ☐ | ☒ | Estimates of effect sizes (e.g. Cohen's *d*, Pearson's *r*), indicating how they were calculated |

*Our web collection on statistics for biologists contains articles on many of the points above.*

## Software and code

Policy information about availability of computer code

| | |
|---|---|
| Data collection | No special software was used in this study |
| Data analysis | All data was analyzed using published methods; for Rb-Tn-seq fitness experiments, we used the pipeline described in Wetmore et al., 2015 in mbio. For SAFE analysis, we used the pipeline described in Leshchiner et al., 2022 in Nature Communications. For these analyses, we used perl (Ubuntu for windows 20.04.3 LTS), Cytoscape (3.9.1), and jupyter notebook (6.4.12). |

For manuscripts utilizing custom algorithms or software that are central to the research but not yet described in published literature, software must be made available to editors and reviewers. We strongly encourage code deposition in a community repository (e.g. GitHub). See the Nature Portfolio guidelines for submitting code & software for further information.

## Data

Policy information about availability of data

All manuscripts must include a data availability statement. This statement should provide the following information, where applicable:

- Accession codes, unique identifiers, or web links for publicly available datasets
- A description of any restrictions on data availability
- For clinical datasets or third party data, please ensure that the statement adheres to our policy

All sequencing data is publicly accessible at NCBI GEO with accession numbers (GSE261860, GSE261867, GSE261873, GSE261757, GSE261749, GSE261214). All data from Rb-Tn-seq is also searchable with our user-interactive website, SalcomFit, at https://bioinf.gen.tcd.ie/cgi-bin/salcomfit.pl.

# Research involving human participants, their data, or biological material

Policy information about studies with <u>human participants or human data</u>. See also policy information about <u>sex, gender (identity/presentation), and sexual orientation</u> and <u>race, ethnicity and racism</u>.

| | |
|---|---|
| Reporting on sex and gender | This is not relevant to our study. |
| Reporting on race, ethnicity, or other socially relevant groupings | This is not relevant to our study. |
| Population characteristics | This is not relevant to our study. |
| Recruitment | This is not relevant to our study. |
| Ethics oversight | This is not relevant to our study. |

Note that full information on the approval of the study protocol must also be provided in the manuscript.

# Field-specific reporting

Please select the one below that is the best fit for your research. If you are not sure, read the appropriate sections before making your selection.

☒ Life sciences ☐ Behavioural & social sciences ☐ Ecological, evolutionary & environmental sciences

For a reference copy of the document with all sections, see <u>nature.com/documents/nr-reporting-summary-flat.pdf</u>

# Life sciences study design

All studies must disclose on these points even when the disclosure is negative.

| | |
|---|---|
| Sample size | All Rb-Tn-seq experiments were done in biological duplicate, and correlation values were very high (generally R>0.8). All other experiments were done with at least 3 independent biological replicates. Exact sample sizes are reported below each figure, in the caption. Sample sizes were not predetermined. However, our replicate numbers were in line with the field (n at least 3 for all experiments). In addition, we note that both our Rb-Tn-seq results and individual growth curves/mutant validations were all highly reproducible, suggesting that our sample sizes were sufficient. |
| Data exclusions | No data was excluded. |
| Replication | We quantified the correlation value (R) for all Rb-Tn-seq experiments and found that they are very high (R~0.85 across all experiments on average), indicating that the data is reproducible. All other experiments were repeated at least 3 independent times, with exact numbers reported below each figure, in the caption. |
| Randomization | This is not applicable to our study, because all bacterial strains/samples were subjected to the same set of conditions. |
| Blinding | This is not applicable to our study, because all bacterial strains/samples were treated in the same way, with the same set of conditions. |

# Reporting for specific materials, systems and methods

We require information from authors about some types of materials, experimental systems and methods used in many studies. Here, indicate whether each material, system or method listed is relevant to your study. If you are not sure if a list item applies to your research, read the appropriate section before selecting a response.

## Materials & experimental systems

| n/a | Involved in the study |
|---|---|
| ☒ | Antibodies |
| ☐ | ☒ Eukaryotic cell lines |
| ☒ | Palaeontology and archaeology |
| ☒ | Animals and other organisms |
| ☒ | Clinical data |
| ☒ | Dual use research of concern |
| ☒ | Plants |

## Methods

| n/a | Involved in the study |
|---|---|
| ☒ | ChIP-seq |
| ☒ | Flow cytometry |
| ☒ | MRI-based neuroimaging |

# Eukaryotic cell lines

Policy information about cell lines and Sex and Gender in Research

| | |
|---|---|
| Cell line source(s) | ATCC THP-1 |
| Authentication | ATCC authenticated these cells using STR profiling. |
| Mycoplasma contamination | We confirmed that the cells did not contain mycoplasma contamination using PCR. |
| Commonly misidentified lines (See ICLAC register) | No commonly misidentified cell lines were used in this study. |

# Plants

| | |
|---|---|
| Seed stocks | *Report on the source of all seed stocks or other plant material used. If applicable, state the seed stock centre and catalogue number. If plant specimens were collected from the field, describe the collection location, date and sampling procedures.* |
| Novel plant genotypes | *Describe the methods by which all novel plant genotypes were produced. This includes those generated by transgenic approaches, gene editing, chemical/radiation-based mutagenesis and hybridization. For transgenic lines, describe the transformation method, the number of independent lines analyzed and the generation upon which experiments were performed. For gene-edited lines, describe the editor used, the endogenous sequence targeted for editing, the targeting guide RNA sequence (if applicable) and how the editor was applied.* |
| Authentication | *Describe any authentication procedures for each seed stock used or novel genotype generated. Describe any experiments used to assess the effect of a mutation and, where applicable, how potential secondary effects (e.g. second site T-DNA insertions, mosaicism, off-target gene editing) were examined.* |

