## [Peer Review File · Nature Genetics]

Peer Review Information

Manuscript Title: High-throughput Fitness Experiments Reveal Specific Vulnerabilities of Human-Adapted Salmonella Isolates During Stress and Infection

Corresponding author name(s): Professor Denise Monack

Reviewer Comments & Decisions:

Decision Letter, initial version:

7th Nov 2023

Dear Professor Monack,

Your Article, "High-throughput Fitness Experiments Reveal Specific Vulnerabilities of Human-Adapted Salmonella Isolates During Stress and Infection" has now been seen by 3 referees. You will see from their comments below that while they find your work of interest, some important points are raised. We are interested in the possibility of publishing your study in Nature Genetics, but would like to consider your response to these concerns in the form of a revised manuscript before we make a final decision on publication.

In brief, the three reviewers all sound enthusiastic for your work, but offer a range of opinions on how much revision is required before your study would be publishable.

Reviewer #1, while saying your work is a "landmark", has one major concern, regarding generalisability of your results: they suggest that you will need to assay a substantially broader range of strains and serovars to support claims of such.

Referee #2 is "very positive", but makes a detailed list of comments regarding improving the screen analysis and presentation of results.

Reviewer #3 is the most positive of all; their specific requests are primarily presentational.

In our reading of these results, we think there is a clear path to publication and that most of these requests appear eminently doable without substantial further work (e.g. Referee #2's specific guidance). While we think that Reviewer #1's comments on generalisability are well-made (and were also noted by Reviewer #2), we leave it up to you and your co-authors to decide how best to respond to this concern; given the positivity overall voiced in these reports, we will not absolutely require the major expansion of the screen that Referee #1 suggests.

To guide the scope of the revisions, the editors discuss the referee reports in detail within the team,

including with the chief editor, with a view to identifying key priorities that should be addressed in revision and sometimes overruling referee requests that are deemed beyond the scope of the current study. We hope that you will find the prioritized set of referee points to be useful when revising your study. Please do not hesitate to get in touch if you would like to discuss these issues further.

We therefore invite you to revise your manuscript taking into account all reviewer and editor comments. Please highlight all changes in the manuscript text file. At this stage we will need you to upload a copy of the manuscript in MS Word .docx or similar editable format.

*2) If you have not done so already please begin to revise your manuscript so that it conforms to our Article format instructions, available here. Refer also to any guidelines provided in this letter.

Please be aware of our guidelines on digital image standards.

[redacted]

Nature Genetics is committed to improving transparency in authorship. As part of our efforts in this direction, we are now requesting that all authors identified as 'corresponding author' on published papers create and link their Open Researcher and Contributor Identifier (ORCID) with their account on the Manuscript Tracking System (MTS), prior to acceptance. ORCID helps the scientific community achieve unambiguous attribution of all scholarly contributions. You can create and link your ORCID

from the home page of the MTS by clicking on 'Modify my Springer Nature account'. For more information please visit www.springernature.com/orcid.

Sincerely,

Michael Fletcher, PhD
Senior Editor, Nature Genetics

ORCID: 0000-0003-1589-7087

Referee expertise: infectious pathogen functional genomics, including Salmonella.

Reviewers' Comments:

Reviewer #1:
Remarks to the Author:

The manuscript presented by Wang et al is a landmark work that addresses the crucial question of functional links between genetic and phenotypic differences among *Salmonella enterica* strains that exhibit wide variation in terms of host range and clinical manifestations.

The authors implemented random barcoded Tn-seq technology for a set of 4 strains, namely two generalist *S. Typhimurium* strains with a broad host range (D23580 and ST4/74) and two *Salmonella enterica* Typhi and Paratyphi A serovars (*S. Typhi* Ty2 and *S. Paratyphi A* 9150) restricted to humans and causing enteric fever.

Fitness assays were carried out under 25 different conditions reproducing different stages of infection and "ecological niches". This is undoubtedly a huge workload, and the analytical tools are state-of-the-art, so I have very few criticisms to make in this respect.

In terms of results, the authors detected hundreds of genes that play a role in strain fitness, and a large majority of them match the usual suspects. Not surprisingly, genes involved in metal homeostasis, LPS modification, macrophage infection and SPI-encoded appeared first. In addition, the team was able to identify a few privileged metabolic networks that appear to be specific to human infections. In a similar vein, the authors show that restricting the host panel also leads to accelerated pseudogenization and genome degradation, mainly due to genes involved in redundant pathways. The authors also tested their hypotheses using mutant and target-deleted strains to confirm the effects observed via Rb-Tn-seq.

However, while this is an important piece of work, I see serious limitations to this approach in its current state. *Salmonella enterica* is composed of strains that undergo extensive horizontal gene transfer, resulting in genetically admixed strains with a non-negligible pangenome. Consequently, focusing on just 4 strains to characterize the specificity and uniqueness of a *Salmonella Typhimurium* versus an enteric fever-inducing *Salmonella enterica* seems rather difficult and far from reflecting the situation as a whole. In other words, the genes and network detected here may be too narrow and

specific for the strains selected. This sounds more like precision medicine than a universal message.

The adaptive landscape of these two entities is quite vast. Therefore, to successfully highlight specific pathways that could serve as therapeutic targets, I would definitely recommend comparing at least 10 members of each class considered here. Common denominator genes and networks from a more substantial sample will be much more likely to lead to the main objective of the manuscript, and will definitely be more universal in their therapeutic application.

I would also recommend achieving maximum genetic and clade diversity for the Typhimurium and Typhi and Paratyphi A serovars. In addition, it might be useful to present the position of selected strains in a genome-based phylogeny. The emerging clone H58 would definitely be a good candidate to add to the sampling scheme I've suggested, due to its considerable importance in public health and increasing frequency over the last two decades.

Reviewer #2:

Remarks to the Author:

Wang et al. present a systematic profiling of 4 saturated genome-wide libraries in different *Salmonella enterica* serovars (2x -Typhimurium, Typhi and Paratyphi) over 24 in vitro conditions and during macrophage infection. This allows them to identify fitness phenotypes for a number of genes across conditions and serovars. They focus on phenotypes that are specific to the two human-restricted serovars (Typhi and Paratyphi), which are known to undergo genomic decay and have accumulated pseudogenes, which makes them less functional redundant in their responses to stress. Using these phenotypes, they identify the responsible pseudogene that makes these strains more sensitive, and often rescue the phenotype by complementing with the functional gene. Overall, this provides a rich resource for the community, at a level that does not exist for any microbe at this moment. Hence I am very positive about this study. That being said, there are some points that need to be fixed so the utility of those unique resources and data fulfill their great potential.

Main points

1. The authors construct highly saturated transposon libraries for 3 serovars/4 strains of same species – but the characterization of these libraries is at best rudimentary (Fig 1 and Table S4 contain minimal information). The analysis should be done properly both per insertion site (the mariner Tn inserts almost exclusively in TA sites) and at gene level (taking into account gene size, probability of Tn hitting gene or not, and specifying what central part of gene means). Importantly with highly saturated Tn libraries, it is imperative to make calls on gene essentiality and perform a comparative analysis of essential genes across the serovars. This is a unique opportunity to make a stronger claim about genomic decay/loss of redundancy in the human-restricted serovars.

2. The statistical analysis, calling of hits and general quality control of the chemical genomics screens needs to be explained better in some parts and/or done more rigorously in others:

- i) it is unclear what is the distribution of reads per Tn insertion is in their experiments, if the authors sum up reads per gene (entire part of gene?) to calculate log₂ ratios (and are there lower boundaries for reads per gene?), whether genes/conditions are filtered out in some cases, and what is biological replicate correlation in all conditions (not in one as shown in Fig S1E-H).

ii) the log₂ gene ratios on the different experiments need to be shown somewhere where variance and normal distributions of data can be assessed (as Fig 1D but violin plots with log₂ and all data). The t-test statistic authors use for significance of scores assumes normally distributed data.

iii) Authors use a co-fitness correlation cut-off of 0.75 for their data based on arbitrary cutoffs used in previous studies. First, it is completely unclear if this co-fitness correlation is a Pearson correlation (or a non-parametric one) and whether based on log₂ values or t-statistic metric. Please note that if the variance of the metric in different conditions is not normalized, Pearson correlations will be heavily driven by data with more variance (i.e. grown under more pressure or for longer, so extreme fitness effects are more pronounced). Second, there needs to be an FDR calculation for those co-fitness correlations to assign a statistically significant cut-off for the R, based on the data, rather than an arbitrary chosen one.

iv) Tables 1 and 2 are uninformative. There is no description whatsoever on the rationale for genes shown (vs not shown), their effect sizes, whether they have positive or negative contribution to fitness and for which of the 4 strains are those hits! Comparisons between conditions are impossible and functional grouping of genes is not apparent (although authors try to put them next to each other); it's just a long list of genes. I would find this much more useful if authors show a clustered heatmap of genes above a certain statistical significance threshold in 1 or more conditions, across the different conditions, and allow clustering to get functional groups. 4 strains can be put together (each gene 4 entries) OR shown separately.

v) I would think that some benchmarking of data will be useful – for example do co-fitness correlations capture known biology in a systematic way (pathways associations, physical complexes).

3. The comparisons of gene responses across conditions should be done in a more systematic way, so reader has a better overview of common or strain/serovar specific responses. Authors follow up MANY examples and talk about even more, but is quite unclear how representative those examples are of the overall picture. This runs the danger that a systematic study ends up featuring anecdotal examples, without an unclear thread line of what they represent as part of the entire dataset. This justification of how examples are picked needs to be strengthened, by a better presentation of results across serovars/strains that allows reader to see the role these examples serve.

First it's unclear how many genes are common in all strains or strain/serovar specific. Based on this, one can assess if conserved or strain-specific genes have more phenotypes on average, and if specific strain/serovar-specific genes that are important in certain (or across) conditions exist.

Second, it's unclear the degree that common functional units (protein complexes, pathways, or even SAFE clusters) respond the same (or differently) across serovars in the different in vitro and in vivo conditions. Are different responses the norm or the exception? Some visualization of that (heat map, network) that allows to highlight the examples chosen to be followed-up is important. Fig 2, but also subsequent ones fail to do this in a manner that one can see major units across organisms and what examples represent.

4. The overall presentation of the screens needs to be done in a better way so others can use it more broadly. I don't think these days long supplementary tables are the way to make this amount of data available to the community. Some interactive interface (e.g. shiny app) with all data should be there so reader can actually navigate through data easily. This will ensure data are used more broadly in the

community.

5. There is no effort to assess how many phenotypes or associations with known genes are discovered for unknown genes. There is a statement about dozens of them in lines 182-187, but no specific analysis on unknown genes across serovars. I find this a lost opportunity.

6. Infection screens.

The authors go for an MOI of 10 - it would be good to access if this means that on average more than one cell is phagocytized by macrophages and if this translates to some extracellular phenotypes (e.g. that of effector mutants) may be trans-complemented this way.

The authors also focus exclusively on genes that are indispensable in the human-restricted serovars to make the point about pseudogenes. However, looking into Figs 5B-C, there are as many genes needed in Typhimurium specifically. What are those genes and why this happens warrants at least some discussion.

The authors almost ignore serovar-specific genes (i.e. genes present only in one/some serovars), which seems a pity. I would assume several effectors and infection related genes will be among them.

Minor points

1. Abstract – can benefit of being more specific on biological findings and processes investigated
2. lines 147-150: rephrase to be more specific – myriads of genes is not that quantitative.
2. lines 150-170: full of examples, which are impossible to see what they represent and if phenotypes are across strains/serovars. For example, *acrAB-tolC* the bile response is exemplified (is this only? as a major pump one would expect more) and responses is cited as universal, but unclear if one serovar or more plotted in scatter plot of Fig 1E.
3. Fig 1B – hard to see much in this resolution. Some representation of gene conservation across serovars (genomes can be shown as lines and synteny between them) + insertion coverage would be more helpful.
4. Fig 1D – n is needed in these plots (for data above/below cutoff)
5. Fig 1E – is this for all strains/one strain. If all for strains how calculated? N needs to be provided. Why are so few LPS-modification genes shown (in contrast numerous dots for *AcrAB-TolC*)?
6. Fig S1E-H – axes are not labelled, n is not provided.
7. lines 212-215 & Fig. 2C/D: where are the unknown genes and how SAFE deals with those will be informative for reader.
8. lines 224-227: it's unclear how these criteria are implemented – what is the SAFE score used? what

is the way infection-related genes are defined? which genes are deemed as not studies in typhoidal Salmonella.

9. Fig 2A /B– what do the colors in SAFE network in 2A represent – how is link from that to Fig 2B? Are all genes shown in Fig. 2B?

10. Fig 2C/D – using different colors for same processes is confusing.

11. Fig 2E – why is rbfE deemed unknown, whereas all others as known? It would be more informative to show all rfb genes log2 scores in heatmap.

12. All heatmaps in Figs – please use the same scale.

13. All OD plots in Figs – please plot OD logarithmically so reader gets a better overview of growth rates (now all focus is on yields).

14. *cysJI* – Typhi Ty2: why is the function impaired? Is expression impaired or do they have a specific mutations that affects protein activity? Latter can be easier deduced these days with structural models.

15. complementation experiments in Fig 3, 4 and 6 need better description –are they always done with plasmids? what expression/copy number?

16. *YbdZ* is not an Fe-S cluster protein, so the connection to the *SUF* complex remains enigmatic – can the authors provide more explanation on this?

17. If the *SUF* complex is defective in Paratyphi, than the *ISC* system should become essential in these strains. Is this something the authors see?

18. line 315 – provide the MOI here.

19. line 322 – how were significant phenotypes identified here?

20. lines 324 – *sitABCD* importance in the the human-restricted serovars- is this linked to the *mntR* mutant followed up later (or everything explained by *YiiP*?)

21. lines 332-334 – don't the authors want to separate between the 2 scenarios? To me it sounds quite important for putting overall results in context

22. lines 359-367 – why *YfeX* plays no role in Typhimurium? What is the redundant peroxidase?

23. line 426 – actually they are just a hundred fitness assays (4 x 25).

24. Fig 4D&E – black trace is not visible.

25. Fig 5B/C – scatter plots need *n* and *r*. Data on other Typhimurim strain are not shown. I would think all pairwise comparisons would be informative. Also if functional groups are shown beyond the only Typhi specific phenotypes (to all phenotypes), this would be more informative.

26. Fig 5 E-J – stating the condition tested in figure panel would be easier for reader to navigate through these panels.

Reviewer #3:

Remarks to the Author:

This remarkable piece of work represents an important step forward and is a tour de force. The authors have used an exciting new technology to ask fundamental questions about the host specificity of *Salmonella enterica*. Over the past 3 decades, *Salmonella* researchers have used a variety of approaches to address this key question that the results were largely disappointing.

This is the first study to investigate the function of ALL coding genes in 3 important *Salmonella* serovars (*S. Typhimurium*, *S. Typhi* and *S. Paratyphi*). Not only did the team investigate gene function in 24 in vitro conditions, they also identified the genes of each serovar involved in the intracellular infection of human macrophages.

As well as thoroughly describing the complex set of transposon mutagenesis experiments and the comprehensive analysis strategies, the authors built on the results of the global screening approach by adding experimental evidence for a number of profound discoveries. For example, mutation of the *yiiP* gene of *S. Typhi* causes serovar-specific effects related to manganese transport, and the functional role of the mutation of the *ybdZ* gene of *S. Paratyphi*. The discovery of the importance of the synthesis of sulphated amino acids in *S. Typhi* could have interesting implications.

Specific comments:

1) This paper is it is likely to become extremely highly cited and be a valuable resource to the community... However, there is a trivial but important issue related to the supplementary data that needs to be corrected. In the supplementary tables, genes are identified using LocusID/gene identifiers such as STM474_RS01505 or STMMW_RS14660. The inclusion of the “_RS” identifier originates from the NCBI “Prokaryotic RefSeq Genome Re-annotation Project” of 2019, but has not been widely adopted by the *Salmonella* community – These “_RS” identifiers could change with future NCBI-generated genome annotations.

Instead of using identifiers with the “_RS” suffix, the authors should add a column to all Supp tables with more widely accepted gene identifiers, such as those specified in the UNIPROT database. For examples, the identifier of the *sifA* gene is STM474_1221 (<https://www.uniprot.org/uniprotkb/E8XFF1/entry>)

The authors should also add a column to all Supp tables that lists the “common gene name” for each gene (for example, *sifA* for the STM474_1221 gene), to assist those readers interested in gene function.

2) A particular highlight of the study of the excellent multi-panel figures which accurately summarise key findings. However, as many figures can have between 8 and 10 panels, they can be a challenge to understand because of the need to repeatedly consult the figure legend. This problem can be solved by adding strain names and gene names to specific panels. For example, in supplementary figure 1 panels ABCDEFGH, the name of each of the strains should be added. Perhaps to the top right of each

panel? Other examples that need the strains to be specified on certain panels are Figure 3F, 3G, 5F, 5I & 5J

Similarly, the clarity of the extremely informative heat maps that show the fitness values of Tn insertions for particular genes in different serovars (such as supplementary figure 1, Panel I) should be improved. I suggest that the individual gene name should be added to the top-right of each of the heatmap panels in ALL Figures & Supplementary Figures, including Figures 3C, 4C, 5D, 6A & supplementary figure 3A.

3) A key strength of the study is the functional comparison of gene function between typhimurium and typhi. For example, Figures 3D and 3E identify an important physiological difference that was recognised from the transposon mutant data shown in panel 3C.

Line 30 (Abstract): The names of the 3 serovars (*S. Typhimurium*, *S. Typhi* and *S. Paratyphi*) should be incorporated into the abstract.

Line 112: the phrase "systemic tissues". Little is known about the precise stresses experienced by *Salmonella* in human tissue.

Line 163: Explain more about the link between Molybdenum and replication in the gut. Ref 47 is a review article, the primary research papers should be cited. Ref 47 only describes a link between Molybdenum and virulence for *S. Typhi*, not for other serovars.

Line 171 onwards. Previously, Barquist and colleagues (PMID: 23470992) showed that Tn insertions in certain SPI2 genes caused reduced fitness for *S. Typhimurium* & *S. Typhi*. Add some sentences commenting on this finding, and state whether similar fitness effects were seen in the Rb-Tn-seq screens for all or certain serovars.

Line 190 onwards: The use of SAFE analysis is clearly justified because it underpinned a number of important mechanistic insights. However, reference 38 describes the use of SAFE analysis for a eukaryote (*Strep pneumoniae*). Perhaps this is the first use of SAFE analysis to investigate functional genomic data for a Gram-negative pathogen? Please add a couple of sentences to put the SAFE analytical approach into the context of other network analysis approaches that have been used for co-fitness analysis - and explain why SAFE analysis was the most appropriate choice here.

Line 194 and elsewhere in the manuscript. Rephrase "based off the". Perhaps change to "that reflects" ?

Line 209: Please explain the fitness cutoff ($|t| > 4$) in some part of the paper, with enough details to explain its basis, without needing the Wetmore et al 2015 paper to be consulted.

Line 241: To help the reader locate the metI gene in the ST4/74 genome, state that the gene is not currently annotated, and specify that it is STM474_0255, or yaeE.

Line 282: change "RS10815" gene identifier to the UNIPROT equivalent.

Line 453: add a citation for the genome decay in host-adapted *Salmonella*.

Line 708: Table 1 should be improved as it could be more informative. Rather than simply listing gene names, it should be explained whether these genes are important in all serovars or not. Perhaps use colours to indicate whether the fitness changes were positive or negative? Perhaps some type of heat map would be useful here?

Line 724: Table 2 should be improved as it could be more informative. Make this into a larger table that describes the role of particular genes during macrophage infection by the 4 *Salmonella* strains. Perhaps use colours to indicate whether the fitness changes were positive or negative?

Line 750. Change "Dh" to "DH" (i.e. the name of Douglas Hanahan).

Lines 818-823. To improve the reproducibility of experiments in other laboratories, add a supplementary table that describes both the supplier and the product code of the individual chemicals used to prepare the GMM medium.

Line 970 & 273: change "THP-1s" to "THP-1 cells".

Figure 1B is useful – please add to this information by adding a Supplementary Figure that shows the Tn insertions in all Plasmids carried by the 4 strains.

Figure 1 legend. Explain precisely what the "-bile" and "-Polymyxin B" conditions are.

Supplementary Figure 3 legend. Specify which serovar the strains used for Panels CDEFG belong to.

Supplementary Table 5 is an excellent idea. To improve the reproducibility of experiments in other laboratories, add columns to the table that describe both the supplier and the product code of the individual chemicals.

Author Rebuttal to Initial comments

We thank all three reviewers for their positive reviews and their many helpful suggestions! We have incorporated these suggestions and respond point-by-point to each individual query below. We believe that overall, these changes have improved our manuscript. Note that the line numbers referenced here belong to the track changed version of our paper.

Reviewer #1:

Remarks to the Author:

The manuscript presented by Wang et al is a landmark work that addresses the crucial question of functional links between genetic and phenotypic differences among *Salmonella enterica* strains that exhibit wide variation in terms of host range and clinical manifestations. The authors implemented random barcoded Tn-seq technology for a set of 4 strains, namely two generalist *S. Typhimurium* strains with a broad host range (D23580 and ST4/74) and two *Salmonella enterica* Typhi and Paratyphi A serovars (*S. Typhi* Ty2 and *S. Paratyphi A* 9150) restricted to humans and causing enteric fever.

Fitness assays were carried out under 25 different conditions reproducing different stages of infection and "ecological niches". This is undoubtedly a huge workload, and the analytical tools are state-of-the-art, so I have very few criticisms to make in this respect.

In terms of results, the authors detected hundreds of genes that play a role in strain fitness, and a large majority of them match the usual suspects. Not surprisingly, genes involved in metal homeostasis, LPS modification, macrophage infection and SPI-encoded appeared first. In addition, the team was able to identify a few privileged metabolic networks that appear to be specific to human infections. In a similar vein, the authors show that restricting the host panel also leads to accelerated pseudogenization and genome degradation, mainly due to genes involved in redundant pathways.

The authors also tested their hypotheses using mutant and target-deleted strains to confirm the effects observed via Rb-Tn-seq.

However, while this is an important piece of work, I see serious limitations to this approach in its current state. *Salmonella enterica* is composed of strains that undergo extensive horizontal gene transfer, resulting in genetically admixed strains with a non-negligible pangenome. Consequently, focusing on just 4 strains to characterize the specificity and uniqueness of a *Salmonella Typhymurium* versus an enteric fever-inducing *Salmonella enterica* seems rather difficult and far from reflecting the situation as a whole. In other words, the genes and network detected here may be too narrow and specific for the strains selected. This sounds more like precision medicine than a universal message.

The adaptive landscape of these two entities is quite vast. Therefore, to successfully highlight specific pathways that could serve as therapeutic targets, I would definitely recommend comparing at least 10 members of each class considered here. Common denominator genes and networks from a more substantial sample will be much more likely to lead to the main objective of the manuscript and will definitely be more universal in their therapeutic application. I would also recommend achieving maximum genetic and clade diversity for the *Typhymurium* and *Typhi* and *Paratyphi A* serovars. In addition, it might be useful to present the position of selected strains in a genome-based phylogeny. The emerging clone H58 would definitely be a good candidate to add to the sampling scheme I've suggested, due to its considerable importance in public health and increasing frequency over the last two decades.

We thank Reviewer 1 for their comments. We note that the editor has not required the expansion of the screen suggested in this review, but we acknowledge the points regarding generalizability that the Reviewer mentions. Thus, we address the generalizability of our results in the following ways, using a mixture of bioinformatic and experimental approaches:

1) For hits that we follow-up on in detail, we note that the associated 'pseudogenes' influencing serovar-specific fitness effects are notably conserved across *Paratyphi* and *Typhi* isolates. For instance, for *ybdZ*, we find that pseudogenization of *sufS* and/or *sufD* in *Paratyphi A* 9150 renders a $\Delta ybdZ$ mutant more sensitive to iron restriction. This pseudogenization pattern is consistently observed in all *Paratyphi A* isolates listed in the NCBI database (taxa 54388) and in all deposited sequences on the BioCyc database. Similarly, for *mntR*, we identified a 2 amino acid deletion in YiiP (L95-F96) that sensitizes *S. Typhi* Ty2 to macrophage-associated stresses. Importantly, this in-frame deletion mutation in YiiP is conserved across all 107 deposited *Typhi* sequences on BioCyc. Consequently, we believe that the serovar-specific fitness effects we studied more mechanistically in this work are likely representative of broader trends in *Typhi* and *Paratyphi A* isolates. We have added this information to the text in **lines 409-413** and **lines 532-534**.

2) To experimentally validate the generalizability of our findings, we constructed clean deletion mutants of "hits" (ie, *metQ*, *ybdZ*, *mntR*) in multiple *Typhi* and *Paratyphi A* isolates beyond the *Typhi* Ty2 and *Paratyphi A* 9150 reference strains. Importantly, these mutants across different typhoidal genetic backgrounds consistently exhibited heightened sensitivity to stress (see figures **S12C-H** for *metQ*, **S13D-E** for *ybdZ*, **S19B-G** for *mntR*). In reference to the

Reviewer's interest in H58 isolates of Typhi, we note that several of the Typhi isolates we constructed mutants in (ISP- 04-06979, ISP-03-07467, E03-9804, ISO(98S), E03-4983) are H58 isolates. Just like the reference Ty2 strain, mutants in these H58 isolates also displayed increased sensitivity to stress, further highlighting the potential generalizability of our findings.

3) To further demonstrate the generalizability of our results, we reconstructed pseudogenes that are present in typhoidal *Salmonella* in the non-typhoidal *Salmonella* Typhimurium ST4/74 background. More specifically, we generated mutants of *cysJ* (which contributes to the inability of *S. Typhi* to grow in minimal media lacking Cys/Met), *sufSD* (which contributes to the increased sensitivity of the $\Delta ybdZ$ mutant in Paratyphi A to iron limitation) and *yiiP* (which contributes to the increased sensitivity of the $\Delta mntR$ mutant in Typhi to several macrophage-associated stress conditions) in the ST4/74 background. Importantly, each of these Typhimurium mutants also showed heightened stress sensitivity, suggesting that pseudogenization of these genes sensitizes *Salmonella* to infection-related stress, regardless of the genetic background. We have added this data in **Sup. Fig 12I for *cysJ* (lines 363-367), Sup. Fig 13F for *ybdZ* (lines 406-410), and Sup. Fig. 19I-J for *mntR* (lines 554-557).**

4) While we note that specific genes in our study likely have widespread, generalizable effects (ie, for *metQ*, *ybdZ*, *mntR*), we acknowledge that there may be many other "hits" that cannot be identified by screening these four representative *Salmonella* isolates. Thus, we have expanded on this point in the discussion and suggest that the further screening of many more genetically diverse *Salmonella* isolates will likely lead to the identification of many more genes with serovar- specific fitness effects with therapeutic potential (see **lines 620-624**).

Reviewer #2:

Remarks to the Author:

Wang et al. present a systematic profiling of 4 saturated genome-wide libraries in different *Salmonella enterica* serovars (2x -Typhimurium, Typhi and Paratyphi) over 24 in vitro conditions and during macrophage infection. This allows them to identify fitness phenotypes for a number of genes across conditions and serovars. They focus on phenotypes that are specific to the two human-restricted serovars (Typhi and Paratyphi), which are known to undergo genomic decay and have accumulated pseudogenes, which makes them less functional redundant in their responses to stress. Using these phenotypes, they identify the responsible pseudogene that makes these strains more sensitive, and often rescue the phenotype by complementing with the functional gene. Overall, this provides a rich resource for the community, at a level that does not exist for any microbe at this moment. Hence I am very positive about this study. That being said, there are some points that need to be fixed so the utility of those unique resources and data fulfill their great potential.

Main points

1. The authors construct highly saturated transposon libraries for 3 serovars/4 strains of same species – but the characterization of these libraries is at best rudimentary (Fig 1 and

Table S4 contain minimal information). The analysis should be done properly both per insertion site (the mariner Tn inserts almost exclusively in TA sites) and at gene level (taking into account gene size, probability of Tn hitting gene or not, and specifying what central part of gene means). Importantly with highly saturated Tn libraries, it is imperative to make calls on gene essentiality and perform a comparative analysis of essential genes across the serovars. This is a unique opportunity to make a stronger claim about genomic decay/loss of redundancy in the human- restricted serovars.

We thank the reviewer for these comments. We have added more detail into **Supplemental Table 4**, which now includes all metrics used by the original Barseq paper (Wetmore et al., 2015 in mBio), as well as other metrics used in recent Tradis papers (see Gray et al., 2023 in BioRxiv). These new details include the number of sequencing reads that were used to map each library, the average number of bp between each Tn insertion in each library, the % of genes with central insertions (defined as in the 10-90% portion of a gene, as described in Wetmore et al., 2015), the mean and median # of insertions within the central part of each gene for each library, and the median number of reads per hit gene from the sequencing of each library. We have also expanded upon this information in the main text (**see lines 134-145**).

In addition, the reviewer correctly points out the high saturation of our Tn libraries allows us to make calls on gene essentiality. To this end, we followed the analysis done previously in Langridge et al., 2009 (Genome Research) & Gray et al., 2023 (BioRxiv), in which we first normalized our Tn-mapping data by dividing the number of unique insertion sites for each gene by its gene length to calculate the "insertion index" of each gene.

When plotting these distributions, we found that they were bimodal, agreeing with the distributions found in both Langridge et al and Gray et al. The left peak, around 0, includes Tn insertions that led to significant growth inhibition during library construction (essential genes), whereas the right peak contained Tn insertions that did not significantly decrease cell growth during library construction (see **Sup. Fig. 2A-D**). In accordance with Langridge et al., we calculated the likelihood ratios (LRs) for each gene's probability of belonging to either the essential gene group (left peak) or the non-essential gene group (right peak). Setting a log₂-LR threshold of 4 (indicating a gene is 16 times more likely to be essential than non-essential), we identified 427 to 476 genes as likely essential in each serovar (**see Sup. Table 5**).

When comparing the essential gene pools across serovars, ~65% of these essential genes are common to all 4 *Salmonella* isolates used in this paper (**Sup. Fig. 2E**). However, we also identified several serovar-specific essential genes. For instance, the inner membrane protein *igaA* is essential for survival in *S. Typhimurium* ST4/74 and D23580, as has been previously reported (Cano et al., 2002, Genetics). Intriguingly, this gene is not vital in either Typhi or Paratyphi A, though the reasons behind this disparity remain unclear at this stage.

We also highlight the essentiality of the alternative sigma factor RpoE and its associated factors DegS & RseP in Typhi, but not in other serovars. While RpoE is thought to be important under conditions of envelope stress in *S. Typhimurium*, our finding that *rpoE* is absolutely essential in Typhi is intriguing and suggests that this sigma factor may be playing different roles in *S. Typhi* (ie, responding to cues beyond envelope stress).

Finally, we note that multiple genes related to iron homeostasis and iron-sulfur cluster assembly (*iscU*, *hcsA*, *hcsB*, *fdx*) are essential in Paratyphi A but not in Typhimurium ST4/74. This suggests

that Paratyphi A may be more sensitive to iron restriction than Typhimurium ST4/74- a finding that we later confirm in the paper (see Fig. 4). In addition, the chaperone SurA, which is involved in the biogenesis of outer membrane proteins, is essential in Paratyphi A but not in the other isolates. These examples are expanded upon in the text from **lines 146-163**.

2. The statistical analysis, calling of hits and general quality control of the chemical genomics screens needs to be explained better in some parts and/or done more rigorously in others:

i) it is unclear what is the distribution of reads per Tn insertion is in their experiments, if the authors sum up reads per gene (entire part of gene?) to calculate log₂ ratios (and are there lower boundaries for reads per gene?), whether genes/conditions are filtered out in some cases, and what is biological replicate correlation in all conditions (not in one as shown in Fig S1E-H).

Gene fitness was calculated in the same way as described in Wetmore et al., 2015 MBio, in which the Barseq technique was developed. Briefly, the fitness of each strain (ie, Tn insertion) is the normalized log₂ ratio of counts between the stress condition and the reference time-zero sample. The fitness of each gene is the weighted average of the fitness of all strains (ie, Tn insertions) within that given gene. For gene fitness, only insertions within 10 to 50% and 50 to 90% of the gene region are used; the 10-90% section of each gene is defined as the “central” portion of that gene. In addition, genes without at least 15 time-zero reference reads are automatically filtered out. We have added this detail to the text at **lines 172-179 & lines 990-1010** and reference Wetmore et al 2015., in which these same parameters were developed.

In addition, the Barseq scripts output a variety of quality scores for each dataset; all of our conditions passed each quality score, so we did not need to exclude any conditions. We have added these quality scores into **Supplemental Table 7**, which also includes the median number of reads for each gene in every fitness experiment.

These metrics include:

- 1) n-used (total number of reads in central portions of genes)
- 2) gMed (median reads per gene in the sample, we want this to be >50 according to Wetmore et al)
- 3) cor12- a measure of how consistent the fitness data is for each gene. Essentially, we take the fitness of a gene using only Tn insertions within the first half of each gene (10-50%) and compared this to the fitness of Tn insertions only within the second half of each gene (50-90%). The cor12 value is the Spearman rank correlation of these two sets of values, and should be >0.2 according to Wetmore et al.
- 4) mad12- measures the median absolute difference (m.a.d) between the fitness according to the first half vs. the second half of each gene. Values should be <0.5 in successful experiments, according to Wetmore et al.
- 5) opcor- measures the consistency of fitness data for each operon, as genes within the same operon should have similar fitness values. In this metric, a Spearman rank correlation on the fitness value for each upstream and downstream gene within an operon is

calculated. Scores >0.2 are considered successful, according to Wetmore et al.

This information was added as **supplemental table 7** and at **lines 1000-1015**. More detail on each of these individual metrics can be found in Wetmore et al., 2015 Mbio.

In addition, we have added a table of biological replicate correlation values for ALL 100 Rb-Tn-seq experiments in **Supplemental Table 7** (see sheet 5). We note that the average R value across these experiments is R~0.85, indicating that biological replicates for all experiments are generally very well correlated.

ii) the log2 gene ratios on the different experiments need to be shown somewhere where variance and normal distributions of data can be assessed (as Fig 1D but violin plots with log2 and all data). The t-test statistic authors use for significance of scores assumes normally distributed data.

We have added in violin plots of the log2 fitness changes for every serovar, across all conditions, in **Supplemental Figure 4A-D**.

In regards to the t-like statistic, we clarify that this is a moderated t-like statistic developed by Wetmore et al., 2015 in the original Barseq paper, where

$$t = \frac{f}{\sqrt{0.1^2 + \max(Ve, Vn)}}$$

In which f is gene fitness, and Ve & Vn are two ways in which the variance can be measured (see Wetmore et al., 2015 for more details).

Essentially, this moderated t-statistic estimates the reliability of the fitness measurement of each gene, in part by estimating the variance in gene fitness; the more reads there are for a gene, the less variance there will be in its fitness measurements. To verify that this moderated t-statistic fits well to the standard normal distribution, we followed the example done in the Wetmore et al 2015 Mbio paper, in which they performed control comparisons between replicate time-zero samples; under these reference conditions, there should be no genuine fitness differences so the modified t-statistic should fit well to a normal distribution. Using the same type of quartile-quartile plot as in Wetmore et al., 2015 (see Figure S5 in their paper), we see that the moderated t-statistic for control comparisons between replicate time-zero samples is indeed normally distributed for each serovar (**Supplemental Figure 21A-D**). We have also clarified these points in the paper text in the methods at **lines 1000-1005**.

iii) Authors use a co-fitness correlation cut-off of 0.75 for their data based on arbitrary cutoffs used in previous studies. First, it is completely unclear if this co-fitness correlation is a Pearson correlation (or a non-parametric one) and whether based on log2 values or t-statistic metric. Please note that if the variance of the metric in different conditions is not normalized, Pearson correlations will be heavily driven by data with more variance (i.e. grown under more pressure or for longer, so extreme fitness effects are more pronounced). Second, there needs

be an FDR

calculation for those co-fitness correlations to assign a statistically significant cut-off for the R, based on the data, rather than an arbitrary chosen one.

We thank the reviewer for these comments and address them with the following points:

1) As was done in Leshchiner et al., 2022 Nature Comms, the co-fitness correlation analysis was performed using log₂ fitness values and using a Pearson's correlation. In addition, we note that the variance of fitness values is normalized, as described in Wetmore et al., 2015 in mBio. Briefly, the Barseq scripts calculate the smoothed median of gene fitness values, using a window of 251 genes so that genuine biological effects are unlikely to be removed. This local median (the estimated bias) is then subtracted from each gene fitness value. In addition, the mode is also subtracted, reflecting an assumption that most Tn insertions have no fitness effects. This analysis is described more in detail in Wetmore et al., 2015. We have added this detail to the paper text at **lines 990-1000**. In regards to possible variance among conditions, we note that the concentration of each stressor was controlled so that the libraries would grow at roughly the same rate in response to each condition (~30-50% reduction in growth in the presence of each stressor), and each experiment was performed for the same amount of time (18 hours).

2) We note that the cut-off of $r=0.75$ was determined empirically, in which we tried a series of different cut-offs (0.6,0.65,0.7,0.75,0.8,0.85,0.9), and then manually analyzed the SAFE clusters to identify a cutoff that would retain genes that we would expect to cluster together based off related functionality, while minimizing clustering among genes that have no functional connection. As the reviewer noted, this 0.75 has also been used by past studies, including Leshchiner et al., 2022. We have added this detail in the **lines 1020-1030**.

3) As suggested by the reviewer, we have now calculated p-values and FDR for all correlations above $r>0.75$ that we captured in our analysis, which can be found in **Supplemental Table 14**. We note that all p-values and FDR values are statistically significant, suggesting that this cut-off of $r=0.75$ is fairly stringent. We have added these details to the methods at **lines 1020-1025**.

4) To further demonstrate the robustness of our analysis, we have performed an additional "stability testing" measurement of all correlations in our networks, as described in Leshchiner et al., 2022. Briefly, to determine the quality of each edge in the network, we built a correlation matrix using partial data by repeatedly hiding random conditions and performed our correlation analysis with 20 out of the 24 possible conditions 100 times. This resulted in 100 binary matrices that we then summed up, with each possible correlation receiving a score between 0 and 100. We find that most of the captured correlations (~75-80%) for each serovar received a stability score >75 , suggesting that most of the connections in our networks are "stable". Importantly, we note that all clusters that we follow-up on in the paper (containing *rbfE*, *metQ*, *ybdZ*, *mntR*) have stability scores >80 on average. We have added stability scores for all edges in our networks in **Supplemental Table 14** (together with the p-values and FDR from the point above), and we expanded on this analysis in the text from **lines 265-275 and 1025-1035**.

iv) Tables 1 and 2 are uninformative. There is no description whatsoever on the rationale for genes shown (vs not shown), their effect sizes, whether they have positive or negative contribution to fitness and for which of the 4 strains are those hits! Comparisons between conditions are impossible and functional grouping of genes is not apparent (although authors try to put them next to each other); it's just a long list of genes. I would find this much more useful if authors show a clustered heatmap of genes above a certain statistical significance threshold in 1 or more conditions, across the different conditions, and allow clustering to get functional groups. 4 strains can be put together (each gene 4 entries) OR shown separately.

We appreciate the reviewer's suggestions and have replaced Tables 1 and 2 with heatmaps. For Heatmap 1, which covers 24 *in vitro* conditions, we included genes showing significant fitness changes ($|t| > 4$) in at least one condition. We generated separate heatmaps for each serovar, each containing between 678 to 781 genes, as shown in **Supplemental Figure 5-8**. We note a few important observations. First, intracellular stress conditions (e.g., bleach, InSPI2-Mg, H₂O₂, NO) tend to cluster together, as do extracellular stresses (e.g., bile, heat, anaerobic conditions, osmotic pressure) and antibiotics (e.g., ciprofloxacin, azithromycin, rifampicin, etc), respectively. This suggests that *Salmonella* employs overlapping gene sets to respond to similar stresses. Second, we observed many clusters of functionally related genes. Given the extensive number of genes on these heatmaps, we have extracted and displayed key clusters in **Supplemental Figure 9** for clarity. We also generated a new **Supplemental Table 10**, listing all gene clusters from the heatmaps across the four serovars. We have re-written the text around expected hits from our Tn-seq screen in lines **194-227**, using these heatmaps as our reference point instead of Table 1. Some examples of these sub-heatmaps include:

- A) The *arnABCT* and *pmrAB* genes cluster together in Typhimurium (see cluster #31 in D23850). Tn insertions in these genes lead to strong fitness defects under polymyxin B treatment.
- B) Genes involved in DNA repair (ie, *recD*, *recG*, *recN*, *recQ*, *recX*)- see cluster #10 in D23580, #27 in ST474). Similar cluster of DNA repair genes (*recJ*, *recN*, *uvrD*, *xthA*) in Para9150- see cluster #20). Tn insertions in these genes lead to strong fitness defects under H₂O₂ and/or ciprofloxacin stress.
- C) Genes involved in iron uptake (ie, *entA*, *entC*, *entD*, *entE*, *entF*, *exbB*, *exbD*)- see cluster #33 in Paratyphi A. Tn insertions in these genes lead to strong fitness defects under iron restriction.
- D) Genes involved in molybdenum metabolism (ie, *moeA*, *moeB*, *moaE*, *moaC*, *moaA*, *modABC*)- see cluster 16 in Typhi, cluster 28 in ST474, cluster 33 in D23580. Tn insertions in these genes lead to strong fitness defects under GMM growth and azithromycin exposure.
- E) Genes involved in LPS synthesis/modification (ie, *rfaL*, *rfaB*, *rfaD*, *wzyO4*, *waaK*)- see cluster #31 in ST4/74, #9 in Ty2, #24 in Paratyphi A, #30 in D23580)- Tn insertions in these genes lead to decreased fitness effects across many conditions, including InSPI2- Mg, protamine, H₂O₂, NO, and polymyxin B treatment.

These examples are highlighted in the text at **lines 194-227**.

For heatmap 2 (replacing Table 2), we incorporated genes demonstrating significant fitness

effects ($p < 0.05$) in the macrophage dataset in at least one serovar. This consolidated heatmap, shown in **Supplemental Figure 15**, includes data from all four serovars and features 1,521 genes. Notably, the heatmap reveals a distinct clustering pattern: typhoidal isolates (Ty2 & Paratyphi A) form one group, while non-typhoidal isolates (ST4/74 and D23580) form another, suggesting that the genes involved in macrophage infection are more similar between the two typhoidal isolates and the two non-typhoidal isolates, respectively. Furthermore, we identify several gene clusters that group together based off related functionality; as it is impossible to see all 1,521 genes on the heatmap, we instead point out several examples of sub-clusters of genes. Some examples include:

- A) SPI-2 related genes cluster together (see cluster 20). Tn insertions in the genes lead to fitness defects in ST4/74, Ty2, and D23580, as mentioned in the text at lines **436-450** and shown in **Sup. Fig. 16 and Sup. Table 21**.
- B) Some flagella genes (*fliD*, *L*, *N*, *P*, *R*, and *motAB*, *fliS*, *fliE*) group together (see cluster 23, 34). Tn insertions in these genes lead to stronger fitness defects in Typhimurium compared to Ty2 or Para9150 (see **Sup. Fig. 16 and Sup. Table 21**).
- C) Some chemotaxis genes (*cheY*, *cheR*, *tar*, *cheW*, *cheA*) cluster together (see cluster 25). Tn insertions in these genes lead to increased fitness in all 4 serovars (see **Sup. Fig. 16 and Sup. Table 21**).
- D) Some LPS-related genes (ie, *wbaP*, *rfbC*, *rfbD*, *rfaL*) group together (see cluster 29, 37- 38). Tn insertions in these genes lead to increased fitness in ST4/74, Para9150, and D23580, and decreased fitness in Ty2 (see **Sup. Fig. 16 and Sup. Table 21**).

We expand on some of these examples in the text as **lines 440-480**.

v) I would think that some benchmarking of data will be useful – for example do co-fitness correlations capture known biology in a systematic way (pathways associations, physical complexes).

Yes, we note in **Figure 2B** that genes with known pathway associations (ie, those involved in molybdenum metabolism, transport, and regulation) cluster together, as well as genes in the *mfa* operon that encode machinery involved in OM lipid transport. In addition, we note that genes in the *nuo* operon encode different components of the same physical complex- namely, the NADH quinone oxidoreductase complex.

3. The comparisons of gene responses across conditions should be done in a more systematic way, so reader has a better overview of common or strain/serovar specific responses. Authors follow up MANY examples and talk about even more, but is quite unclear how representative those examples are of the overall picture. This runs the danger that a systematic study ends up featuring anecdotal examples, without an unclear thread line of what they represent as part of the entire dataset. This justification of how examples are picked needs to be strengthened, by a better presentation of results across serovars/strains that allows reader to see the role these examples serve.

First it's unclear how many genes are common in all strains or strain/serovar specific. Based on this, one can assess if conserved or strain-specific genes have more phenotypes on average, and if specific strain/serovar-specific genes that are important in certain (or across) conditions exist.

We thank the reviewer for this comment. We have now added a new supplemental table (**Sup. Table 12**) listing all the genes uniquely encoded by *S. Typhi* and *S. Paratyphi A*, compared to *S. Typhimurium*, along with their moderated t-statistics across every stress condition. We note that comparatively few of these strain-specific genes have significant phenotypes; 15/380 (3.9%) have significant phenotypes in Typhi, while 8/246 (3.2%) of strain unique genes in Paratyphi A had a significant fitness effect. Typhi-specific genes with phenotypes include *rbfE*, which adds on a unique tyvelose sugar into the LPS layer, as well as multiple Vi-capsule encoding genes (ie, *vexABCDE*). Intriguingly, we find that Tn insertions in these Vi capsule genes lead to increased fitness under several conditions including protamine stress. In contrast, we note that a higher number of shared genes had significant fitness effects in Typhi (668/3606, or 18.5%) and Paratyphi A (571/3464, or 16.5%). Together, these observations may suggest that shared genes are more likely to have fitness effects under the panel of more general stress conditions that we screened in this paper, while genes uniquely encoded by certain serovars may have phenotypes under more specific conditions that were not tested in our 24 conditions. We have expanded on these points in **lines 235-245** in the text.

Second, it's unclear the degree that common functional units (protein complexes, pathways, or even SAFE clusters) respond the same (or differently) across serovars in the different *in vitro* and *in vivo* conditions. Are different responses the norm or the exception? Some visualization of that (heat map, network) that allows to highlight the examples chosen to be followed-up is important. Fig 2, but also subsequent ones fail to do this in a manner that one can see major units across organisms and what examples represent.

As the reviewer suggested, we have now added in more visualization of the example clusters we follow-up on in the paper in the form of heatmaps. For the LPS-related cluster with *rbfE* from Fig. 2, we have now added in a heatmap of all genes in the SAFE cluster, and their fitness values across all *in vitro* conditions for *S. Typhi* Ty2 (see **Sup. Fig. 11**). For the *metlQN* SAFE cluster (Fig. 3), we have now added in a heatmap of all genes in this cluster and their fitness values across all conditions for Typhi Ty2 (**Sup. Fig. 12A**). We have also added an additional heatmap showing the fitness changes of *metlQN* across all 4 serovars in response to sodium hypochlorite (see **Sup. Fig. 12B**). For the metal homeostasis SAFE cluster (Fig. 4), we have now added a heatmap showing the fitness values of all genes in this cluster in *S. Paratyphi A* 9150, in response to all conditions (see **Sup. Fig. 13A**). We have also added an additional heatmap showing the fitness changes of these metal-related genes across all 4 serovars in response to iron restriction (see **Sup. Fig. 13B**). Finally, we show an additional heatmap of the fitness changes of *mntR* across all 4 serovars, in response to all 24 *in vitro* stress conditions (see **Sup. Fig. 19A**).

Through these added heatmaps, we demonstrate that 1) genes within a given SAFE cluster usually have very similar effects in response to stress; indeed, this is why these genes cluster together. 2) For the SAFE clusters harboring the serovar-specific examples we followed up on (*metlQN*, *ybdZ*, *mntR*), these clusters of genes respond differently across the different serovars in response to stress.

4. The overall presentation of the screens needs to be done in a better way so others can use it more broadly. I don't think these days long supplementary tables are the way to make this amount of data available to the community. Some interactive interface (e.g. shiny app) with all data should be there so reader can actually navigate through data easily. This will ensure data are used more broadly in the community.

We are working with our new co-authors Dr. Karsten Hokamp and Dr. Jay Hinton, who created the interactive Salcomm database (<http://bioinf.gen.tcd.ie/cgi-bin/salcom.pl? HL>) to release an interactive website with all of our data to accompany the eventual publication of our manuscript. We have so far created a draft website of our database, called SalcommFitness, which can be accessed here (http://bioinf.gen.tcd.ie/cgi-bin/jay/salcom_v2b.pl?db=barseq_HL), in which readers can type in the names of any genes of interest to see their fitness effects across all conditions. This website will be polished and fully ready for release at the time of publication.

5. There is no effort to assess how many phenotypes or associations with known genes are discovered for unknown genes. There is a statement about dozens of them in lines 182-187, but no specific analysis on unknown genes across serovars. I find this a lost opportunity.

We thank the reviewer for raising this important point. We have now added an additional analysis in which we searched for correlations on our network maps that contain at least one hypothetical gene, in each of the 4 serovar maps (see **Supplemental Table 16**). We find that ~15-20% of the connections found on each network map contain at least 1 hypothetical gene. Intriguingly, we find that some of these very strong connections ($R > 0.9$) are between one unknown and one known gene, offering valuable insights for hypothesizing the functions of these unidentified genes. This supplementary analysis should serve as a beneficial resource for researchers seeking to elucidate the roles of these unknown genes. We have incorporated several examples of these associations in the text, including:

- A) RS16480 (ST4/74) and RS16305 (D23580) & *cpxR* in Typhimurium. CpxR is a response regulator that responds to envelope stress, while RS16480/RS16305 is not characterized in *Salmonella* and is very poorly characterized in *E. coli*, with no known biological role. The fact that the fitness effects of these two genes are highly correlated (0.96 in ST4/74, 0.92 in D23580) suggests that RS16480/RS16305 may be involved in the CpxR-mediated envelope stress response pathway in Typhimurium.
- B) RS03310 clusters with many genes involved in amino acid metabolism in *S. Typhi* Ty2 (ie, *cysQ*, *hisH*, etc.). RS03310 is not characterized in *Salmonella* and is poorly studied in *E. coli*, in which it has been shown to be a transcription factor with TF binding sites upstream of certain metabolic operons. RS03310 has $R > 0.9$ with 24 different metabolic genes in *Salmonella*, suggesting that it may play a critical role in the transcriptional regulation of amino acid metabolism. Intriguingly, this gene in Paratyphi A (RS03250) also has strong correlation with amino acid metabolism genes, while the same gene in Typhimurium (RS11305 in ST4/74 & RS11495 in D23580) is not well-

correlated with any metabolic genes, suggesting that this transcription factor may play different roles in non-typhoidal and typhoidal *Salmonella*.

- C) RS10805 clusters with genes involved in iron acquisition in *S. Paratyphi A* (ie, *entE*, *entF*, etc...) This is an example that we go in a lot more detail in the paper (see Figure 4), but we've now specifically highlighted this uncharacterized gene in *Salmonella* as an example of an uncharacterized gene that clusters with known genes involved in siderophore synthesis.
- D) Many prophage-related genes in *S. Typhimurium* D23580 contained within the ST64B prophage (eg, RS10205, RS10230, RS10265, etc...., many of which are genes of unknown function) cluster together with each other and with genes involved in DNA replication & repair. These genes have very strong fitness effects in the presence of DNA- damaging stressors, including H₂O₂ and ciprofloxacin. This data suggests that specific prophages in *Salmonella* are being induced under infection-related stress, which we speculate may have intriguing effects on the physiology of this pathogen.

We have summarized some of these examples in the text as lines **279-289**.

6. Infection screens.

The authors go for an MOI of 10 - it would be good to access if this means that on average more than one cell is phagocytized by macrophages and if this translates to some extracellular phenotypes (e.g. that of effector mutants) may be trans-complemented this way.

As discussed in another Tn-seq paper involving *Salmonella* infection of macrophages (Canals et al., 2019 in Plos Pathogens), trans-complementation is indeed a possibility in these sorts of experiments, as different mutants will be competing for infection into the same pool of macrophages in a highly competitive environment. We have added this caveat into the paper at lines **425-430**.

The authors also focus exclusively on genes that are indispensable in the human-restricted serovars to make the point about pseudogenes. However, looking into Figs 5B-C, there are as many genes needed in Typhimurium specifically. What are those genes and why this happens warrants at least some discussion.

As the reviewer pointed out, there are some genes with decreased fitness in Typhimurium but not in Typhi/Paratyphi A. We note that many of these genes are involved in flagellar synthesis and regulation (ie, genes in the *fli* operon, see **Sup Fig. 16D, Sup. Table 21-23**). Previous studies have suggested that flagella may mediate uptake of *Salmonella* into macrophages (Olsen et al., 2013) and promote escape of *Salmonella* from macrophages (Sano et al., 2007); the Olsen study also noted differences in the role of flagella between *Salmonella* Typhimurium and Dublin. We have added this note in the text **as lines 500-503**. Other genes with strong intracellular fitness defects only in Typhimurium but not Typhi include *cgtA*, which has not been studied in *Salmonella* but likely contributes to persister formation in *E. coli* (Verstraeten et al., 2015.)

The authors almost ignore serovar-specific genes (i.e. genes present only in one/some serovars), which seems a pity. I would assume several effectors and infection related genes will be among them.

As the reviewer alludes to, there are several effectors that are encoded only in Typhimurium, but not in Typhi or Paratyphi A (*avrA*, *gogB*, *gtgA*, *gtgE*, *steB*, *steA*, *gogA*, *sfrJ*, *sseI*, *sseK1*, *sseK3*). Of these Typhimurium-unique genes, we note that Tn insertions in *sseK1* in both ST4/74 and D23580 lead to modest growth defects within macrophages (~2-3x), while Tn insertions in *sseK3* have strong fitness defects in both ST4/74 and D23580 (~4-50x). We have added this information to the text in **lines 455-460**. Both SseK1 and SseK3 likely inhibit NF-kB signaling and macrophage cell death. Typhi encodes one unique effector named *stoD*, and Tn insertions in *stoD* have a modest decrease in fitness (~1.5x). In addition, we find that the Tn insertions in the Typhi-unique gene *rbfE* also leads to decreased fitness within macrophages (see **Sup. Fig. 17A**), suggesting that the unique tyvelose sugar in Typhi plays an important role for the intracellular survival of this pathogen.

Minor points

1. Abstract – can benefit of being more specific on biological findings and processes investigated

We thank the reviewer for this point- unfortunately, the text limit for the abstract is fairly short (150 words), so there isn't too much room to add our three 3 biological processes into the abstract.

2. lines 147-150: rephrase to be more specific – myriads of genes is not that quantitative.

We have removed this phrase and stated in the next sentence that ~700 genes in each serovar had significant fitness effects in at least 1 condition.

2. lines 150-170: full of examples, which are impossible to see what they represent and if phenotypes are across strains/serovars. For example, *acrAB-tolC* the bile response is exemplified (is this only? as a major pump one would expect more) and responses is cited as universal, but unclear if one serovar or more plotted in scatter plot of Fig 1E.

We have completely rewritten this section based on the Reviewer's suggested heatmaps for reference. For each example we provide, we have indicated which serovar the example is derived from and which cluster # it corresponds to on the heatmap (see **lines 194-227**).

3. Fig 1B – hard to see much in this resolution. Some representation of gene conservation across serovars (genomes can be shown as lines and synteny between them) + insertion coverage would be more helpful.

We apologize for this, as the jpeg image we imported into the word document lost some resolution. For publication, we will provide high resolution .svg for this image.

4. Fig 1D – n is needed in these plots (for data above/below cutoff)

We have added information in the text at line 186, stating that between 678 to 781 genes passed this moderated t-stat based cut-off in each serovar.

5. Fig 1E – is this for all strains/one strain. If all for strains how calculated? N needs to be provided. Why are so few LPS-modification genes shown (in contrast numerous dots for AcrAB-TolC)?

For simplicity, we have re-made these graphs so that Fig. 1E is derived from one strain. Top (AcrAB/TolC) is now only from Typhi, so there are only 3 red dots highlighted. Bottom is now only from D23580, and all the highlighted dots are related to LPS-modification/synthesis.

6. Fig S1E-H – axes are not labelled, n is not provided. We have added this info to the figure now.

7. lines 212-215 & Fig. 2C/D: where are the unknown genes and how SAFE deals with those will be informative for reader.

We have added a section on unknown genes (see our response to Reviewer 2 point 5). We find that ~15-20% of each map is comprised of unknown genes, suggesting that our approach can be used to hypothesize functions for these uncharacterized genes.

8. lines 224-227: it's unclear how these criteria are implemented– what is the SAFE score used? what is the way infection-related genes are defined? which genes are deemed as not studies in typhoidal *Salmonella*.

We have clarified the criteria used for selecting our 3 clusters for more in-depth follow-up in lines **311-317**. These include: 1) gene cluster with genes uniquely encoded in typhoidal *Salmonella*, 2) gene cluster with genes that are shared among all 4 isolates but with serovar-specific changes in fitness, and 3) gene clusters that include uncharacterized genes in *Salmonella*.

9. Fig 2A /B– what do the colors in SAFE network in 2A represent – how is link from that to Fig 2B? Are all genes shown in Fig. 2B?

The colors in Fig. 2A correspond to different clusters of functionally enriched genes; the corresponding functional categories are shown in **Sup. Fig. 10A**. Note that Fig. 2A is the map from ST4/74, but the highlighted clusters in all 4 serovars are shown in **Sup. Fig. 10A-D**.

10. Fig 2C/D – using different colors for same processes is confusing.

Unfortunately, the SAFE plug-in cannot be adjusted to output specific colors on these correlation maps. We will write the original authors of SAFE to see if they can input this functionality onto Cytoscape.

11. Fig 2E – why is rbfE deemed unknown, whereas all others as known? It would be more informative to show all rfb genes log2 scores in heatmap.

We have clarified this in the text- RfbE is the only LPS modification gene that is uniquely encoded in Typhi. We have added a heatmap of all rfb genes (see Sup. Figure 11).

12. All heatmaps in Figs – please use the same scale.

Since the fitness of genes can vary substantially from figure to figure, we used different scales for easier visualization for each hit.

13. All OD plots in Figs – please plot OD logarithmically so reader gets a better overview of growth rates (now all focus is on yields).

We have now added in a new Supplemental figure (Sup. Fig. 20) that includes all growth curves in the main figures plotted on a natural log scale, so that the readers can get a better idea of growth rates.

14. *cysJ* – Typhi Ty2: why is the function impaired? Is expression impaired or do they have a specific mutations that affects protein activity? Latter can be easier deduced these days with structural models.

When looking at the crystal structure of CysJ, we note that there are a number of point mutations in the FNR/FAD-NADPH binding domain of this enzyme in the Typhi version of this protein (see Tavolieri et al, 2019, Journal of Structural Biology) that may interfere with its activity in Typhi. Determining what combination of these point mutations leads to decreased function of CysJ in *S. Typhi* will be an intriguing area of future work.

We have now also used qRT-PCR to measure the expression of *cysJ* between ST4/74 and Ty2. We find that there are no strong differences in *cysJ* expression between these strains in LB (which contains Cys/Met), or in InSPI2 minimal media lacking both Cys/Met, in which *cysJ* expression should be strongly induced. We include the qRT-PCR figure below here in the reviewer response. Note that all fold changes are normalized to the expression of *cysJ* in ST4/74 in LB, using *rpoD* as the reference gene. Cells were grown to an OD~1 in LB and then back-diluted 5x into either LB or InSPI2 for 30 minutes before RNA was extracted.

15. complementation experiments in Fig 3, 4 and 6 need better description –are they always done with plasmids? what expression/copy number?

These experiments are done with the pwsk29 plasmid, which is a low copy plasmid that has been used for complementation in *Salmonella*. We have added this detail to the methods at **line 884**.

16. YbdZ is not an Fe-S cluster protein, so the connection to the SUF complex remains enigmatic – can the authors provide more explanation on this?

We thank the reviewer for this comment. We note that previous studies done in *E. coli* (Outten et al., 2004 in Mol. Micro) and in *Erwinia chrysanthemi* (Nachin et al., 2003 in EMBO) have observed similar phenotypes in which SUF mutants exhibit increased sensitivity under iron restriction; in our case, the *ybdZ* mutant leads to a condition of iron restriction in Paratyphi A, likely due to defects in siderophore biosynthesis. These past studies in *E. coli* and *E. chrysanthemi* have suggested that SUF plays an essential role in Fe-S assembly specifically during iron starvation: under conditions of iron limitation, Fe-S cluster assembly is limiting and the SUF complex is specifically recruited to increase the production of these Fe-S clusters. If SUF is mutated or missing under conditions of iron limitation, Fe-S cluster assembly becomes further impaired and enzymes that require Fe-S co-factors (ie, phosphogluconate dehydratase in *E. coli*) can no longer function, eventually leading to growth arrest. Similar experiments have not yet been performed in *Salmonella*, but we believe this is an intriguing area of future research. We have added this information in the text at **lines 398-400**.

17. If the SUF complex is defective in Paratyphi, than the ISC system should become essential in these strains. Is this something the authors see?

Yes, we do see that genes involved in Fe-S cluster assembly, including ISC genes, are essential in Paratyphi A (see **Supplementary Table 5**).

18. line 315 – provide the MOI here. We have added this.

19. line 322 – how were significant phenotypes identified here?

We note that in the combined heatmap of macrophage fitness changes requested by Reviewer 2 point 2-iv, there were 1,521 genes with a significant fitness effect within macrophages in at least 1 tested isolate, which were defined as genes with fitness effects $p < 0.05$ by DEseq2 analysis, as previously described in Canals et al., 2019 in Plos Pathogens.

20. lines 324 – sitABCD importance in the the human-restricted serovars- is this linked to the *mntR* mutant followed up later (or everything explained by YiiP?)

We believe that the serovar-specific fitness defects of the *mntR* mutant can be entirely explained by the pseudogenization of *yiiP* in Typhi. First, we note that expressing a functional version of YiiP can fully rescue the growth defect of this *mntR* mutant under hydrogen peroxide stress (**Fig.6G**) and can also fully rescue the Mn-buildup phenotype measured by ICP-MS in this mutant (**Fig. 6F**). Second, we note that deleting *yiiP* in the *mntR* mutant in

Typhimurium ST4/74 greatly sensitizes this mutant to both Mn build-up and hydrogen peroxide exposure (**Sup. Fig. 18I-J**). Thus, we believe that YiiP is both necessary and sufficient to drive the fitness defects of an *mntR* mutant during macrophage-associated stress conditions.

21. lines 332-334 – don't the authors want to separate between the 2 scenarios? To me it sounds quite important for putting overall results in context

We have rephrased this section to make these two scenarios sound more separate from each other.

22. lines 359-367 – why YfeX plays no role in Typhimurium? What is the redundant peroxidase?

The reviewer is correct that this observation suggests that there may be a redundant peroxidase in Typhimurium that explains why *yfeX* appears to be less important in this isolate. We believe that the identification of this redundant peroxidase is an interesting avenue of future research.

23. line 426 – actually they are just a hundred fitness assays (4 x 25).

To clarify, we were including the fact that we performed biological duplicates for each experiment.

24. Fig 4D&E – black trace is not visible.

We have made the black trace more visible.

25. Fig 5B/C – scatter plots need n and r. Data on other Typhimurim strain are not shown. I would think all pairwise comparisons would be informative. Also if functional groups are shown beyond the only Typhi specific phenotypes (to all phenotypes), this would be more informative.

We have added R values to these panels (R=0.5 for Typhi, 0.4 for Paratyphi A), and specified in the caption that this is data from 2 biological replicates. We have also added the correlation between the two Typhimurium strains (ST4/74 and D23580) into **Sup. Fig. 18**; the R value here is higher (0.6), suggesting that the intracellular fitness changes between these non-typhoidal serovars are more similar to each other. This agrees with the clustering pattern we observe on the macrophage heatmap (**Sup. Fig. 15**).

26. Fig 5 E-J – stating the condition tested in figure panel would be easier for reader to navigate through these panels.

We have added these sub-headings.

Reviewer #3:

Remarks to the Author:

This remarkable piece of work represents an important step forward and is a tour de force. The authors have used an exciting new technology to ask fundamental questions about the host specificity of *Salmonella enterica*. Over the past 3 decades, *Salmonella* researchers have used a variety of approaches to address this key question that the results were largely disappointing.

This is the first study to investigate the function of ALL coding genes in 3 important *Salmonella* serovars (*S. Typhimurium*, *S. Typhi* and *S. Paratyphi*). Not only did the team investigate gene function in 24 in vitro conditions, they also identified the genes of each serovar involved in the intracellular infection of human macrophages.

As well as thoroughly describing the complex set of transposon mutagenesis experiments and the comprehensive analysis strategies, the authors built on the results of the global screening approach by adding experimental evidence for a number of profound discoveries. For example, mutation of the *yiiP* gene of *S. Typhi* causes serovar-specific effects related to manganese transport, and the functional role of the mutation of the *ybdZ* gene of *S. Paratyphi*. The discovery of the importance of the synthesis of sulphated amino acids in *S. Typhi* could have interesting implications.

We thank the reviewer for their very enthusiastic and positive feedback!

Specific comments:

1) This paper is it is likely to become extremely highly cited and be a valuable resource to the community... However, there is a trivial but important issue related to the supplementary data that needs to be corrected. In the supplementary tables, genes are identified using LocusID/gene identifiers such as STM474_RS01505 or STMMW_RS14660. The inclusion of the “_RS” identifier originates from the NCBI “Prokaryotic RefSeq Genome Re-annotation Project” of 2019, but has not been widely adopted by the *Salmonella* community – These “_RS” identifiers could change with future NCBI-generated genome annotations. Instead of using identifiers with the “_RS” suffix, the authors should add a column to all Supp tables with more widely accepted gene identifiers, such as those specified in the UNIPROT database. For examples, the identifier of the *sifA* gene is STM474_1221 (<https://www.uniprot.org/uniprotkb/E8XFF1/entry>)

The authors should also add a column to all Supp tables that lists the “common gene name” for each gene (for example, *sifA* for the STM474_1221 gene), to assist those readers interested in gene function.

We thank the reviewer for raising this important point. We have now updated our key supplemental tables containing all log₂ fitness changes and moderated t-stats to also include common gene names, as well as the more widely accepted gene identifiers (for example, see updated **Sup. Tables 8-9, 20**).

To further aid future researchers in using these data, we have also added an additional supplemental table (**Sup. Table 13**) listing ortholog information for all genes shared among these

4 serovars. This table also includes all common gene names, RS identifiers (as locusID 1), and more common identifiers (as locusID 2) for each gene, across each of the 4 serovars.

2) A particular highlight of the study of the excellent multi-panel figures which accurately summarise key findings. However, as many figures can have between 8 and 10 panels, they can be a challenge to understand because of the need to repeatedly consult the figure legend. This problem can be solved by adding strain names and gene names to specific panels. For example, in supplementary figure 1 panels ABCDEFGH, the name of each of the strains should be added. Perhaps to the top right of each panel? Other examples that need the strains to be specified on certain panels are Figure 3F, 3G, 5F, 5I & 5J

Similarly, the clarity of the extremely informative heat maps that show the fitness values of Tn insertions for particular genes in different serovars (such as supplementary figure 1, Panel I) should be improved. I suggest that the individual gene name should be added to the top-right of each of the heatmap panels in ALL Figures & Supplementary Figures, including Figures 3C, 4C, 5D, 6A & supplementary figure 3A.

We thank the reviewer for this suggestion and have added these labels with gene names, strains names, and/or conditions to the suggested panels.

3) A key strength of the study is the functional comparison of gene function between typhimurium and typhi. For example, Figures 3D and 3E identify an important physiological difference that was recognised from the transposon mutant data shown in panel 3C.

We thank the reviewer for this comment!

Line 30 (Abstract): The names of the 3 serovars (S. Typhimurium, S. Typhi and S. Paratyphi) should be incorporated into the abstract.

We have added this.

Line 112: the phrase “systemic tissues”. Little is known about the precise stresses experienced by Salmonella in human tissue.

We clarify that some of our stresses (e.g. bile) are found in systemic sites (e.g. the gallbladder).

Line 163: Explain more about the link between Molybdenum and replication in the gut. Ref 47 is a review article, the primary research papers should be cited. Ref 47 only describes a link between Molybdenum and virulence for S. Typhi, not for other serovars.

Contreras et al (1997, Microbiology) had shown that defects in molybdenum metabolism leads to decreased invasion of S. Typhi into epithelial cells and hinders replication within epithelial cells. In addition, several Mo-binding enzymes in S. Typhimurium have been shown to contribute to the virulence of this pathogen. For example, mutations in the Mo-binding enzyme BisC lead to growth defects *in vivo* during intraperitoneal infections in mice (Denkel et al., 2013). Similarly, a mutant in Mo-binding TtrA strain of S. Typhimurium is outcompeted by the WT strain in mouse and bovine gut colitis infection models (Winter et al., 2010). We have added some of this information to the

text at **lines 214-216**.

Line 171 onwards. Previously, Barquist and colleagues (PMID: 23470992) showed that Tn insertions in certain SPI2 genes caused reduced fitness for *S. Typhimurium* & *S. Typhi*. Add some sentences commenting on this finding , and state whether similar fitness effects were seen in the Rb-Tn-seq screens for all or certain serovars.

Barquist et al reported that several SPI-2 related genes (*spiC*, *sseA*, *ssaH*, *ssaI*, *ssaJ*, *ssaT*) were not hit by Tn insertions and fell into the “essential genes” category. The authors note that these genes are not truly essential, as they could generate KO mutants of each of these, but that the lack of Tn insertions in these genes was likely due to blocked access of these AT rich regions of the genome by H-NS. In our experiments, we also find that these genes have lower density of Tn insertions compared to the median (5.2 strains per gene vs 22 for ST4/74, 10.8 strains per gene vs. 27 for Typhi, 6.7 strains per gene vs. 44 for Para9150, and 2.2 strains per gene vs 12 for D23580). Thus, we observed a similar effect compared to Barquist et al., where these AT-rich genes exhibit decreased Tn insertion density. In turn, we were not able to detect fitness changes for some of these genes (ie, *ssaH* in all 4 serovars), due to too few reads/Tn insertions in this gene. We have added this caveat into the text at **lines 160-163**.

Line 190 onwards: The use of SAFE analysis is clearly justified because it underpinned a number of important mechanistic insights. However, reference 38 describes the use of SAFE analysis for a eukaryote (*Strep pneumoniae*). Perhaps this is the first use of SAFE analysis to investigate functional genomic data for a Gram-negative pathogen? Please add a couple of sentences to put the SAFE analytical approach into the context of other network analysis approaches that have been used for co-fitness analysis - and explain why SAFE analysis was the most appropriate choice here.

We thank the reviewer for this comment. While SAFE analysis has been applied in yeast (*S. cerevisiae*, in Baryshnikova et al., 2018) and a Gram-positive bacterium (*S. pneumoniae*, see Leshchiner et al., 2022), to our knowledge SAFE has not yet been used to study the fitness of Gram-negative pathogens or to compare functional differences among genetically distinct isolates of the same species. The original SAFE paper by Baryshnikova et al. (2016 in Cell Systems) highlights the advantages of SAFE over other methods. Specifically, despite the use of co-fitness network analysis for over a decade, the integration of functional data to identify 'functionally coherent regions' in network maps has been lacking. SAFE addresses this by enabling the comprehensive identification of functional clusters within a network by overlaying functional information on these correlation maps, facilitating the creation of a functional map that enhances the visualization of gene networks and clusters. We have incorporated these insights into the manuscript at **lines 255-260**.

Line 194 and elsewhere in the manuscript. Rephrase “based off the”. Perhaps change to “that reflects” ?

We have adjusted this as suggested.

Line 209: Please explain the fitness cutoff ($|t|>4$) in some part of the paper, with enough details to explain its basis, without needing the Wetmore et al 2015 paper to be consulted.

We have added more detail in (see Reviewer 2, points 2-i & ii).

Line 241:, To help the reader locate the *metI* gene in the ST4/74 genome, state that the gene is not currently annotated, and specify that it is STM474_0255, or yaeE.

It appears that *metI* is now annotated in ST4/74 genome, since the paper was out for review.

Line 282: change “RS10815” gene identifier to the UNIPROT equivalent.

The UNIPROT name for this gene in *ybdZ*, which we now mention in the sentence (see lines 373-375).

Line 453: add a citation for the genome decay in host-adapted Salmonella.

We have added this.

Line 708: Table 1 should be improved as it could be more informative. Rather than simply listing gene names, it should be explained whether these genes are important in all serovars or not.

Perhaps use colours to indicate whether the fitness changes were positive or negative?

Perhaps some type of heat map would be useful here?

As suggested by both reviewers 2 and 3, we have now replaced Table 1 with clustered heatmaps. This heatmap covers 24 *in vitro* conditions and includes genes showing significant fitness changes ($|t|>4$) in at least one condition. We generated separate heatmaps for each serovar, each containing between 678 to 781 genes, as shown in **Supplemental Figure 5-8**. We note a few important observations. First, intracellular stress conditions (e.g., bleach, InSPI2-Mg, H₂O₂, NO) tend to cluster together, as do extracellular stresses (e.g., bile, heat, anaerobic conditions, osmotic pressure) and antibiotics (e.g., ciprofloxacin, azithromycin, rifampicin, etc), respectively. This suggests that *Salmonella* employs overlapping gene sets to respond to similar stresses. Second, we observed many clusters of functionally related genes. Given the extensive number of genes on these heatmaps, we have extracted and displayed a few example clusters in **Supplemental Figures 9** for clarity. We also generated a new **Supplemental Table 10**, listing all gene clusters from the heatmaps across the four serovars. We have re-written the text around expected hits from our Tn-seq screen in lines **180-225**, using these heatmaps as our reference point instead of Table 1. For specific examples, please see our response to **Reviewer 2, point 2-iv**.

Line 724: Table 2 should be improved as it could be more informative. Make this into a larger table that describes the role of particular genes during macrophage infection by the 4 Salmonella strains. Perhaps use colours to indicate whether the fitness changes were positive or negative?

Reviewer 2 has also suggested that we replace Table 2 with a heatmap, which we have now constructed. This heatmap visually represents gene fitness within macrophages, integrating data from all four serovars. It includes genes that exhibit a statistically significant phenotype ($p < 0.05$) in at least one isolate, as illustrated in **Supplemental Figure 15**. Furthermore, we have compiled a comprehensive supplemental table (**Sup. Table 21**) detailing all gene clusters depicted on this heatmap. Selected examples of these gene clusters are showcased in **Supplemental Figures 16**. For additional details, please refer to the corresponding sections in the **text at lines 443-482** and our specific response to **Reviewer 2's comment 2-iv**.

Line 750. Change “Dh” to “DH” (i.e. the name of Douglas Hanahan).

We have adjusted this.

Lines 818-823. To improve the reproducibility of experiments in other laboratories, add a supplementary table that describes both the supplier and the product code of the individual chemicals used to prepare the GMM medium.

We have added in this information in **Supplemental Table 6**.

Line 970 & 273: change “THP-1s” to “THP-1 cells”.

We have adjusted this throughout the text

Figure 1B is useful – please add to this information by adding a Supplementary Figure that shows the Tn insertions in all Plasmids carried by the 4 strains.

We have now mapped all Tn insertions in plasmids- there are 3 plasmids in ST4/74 and 4 plasmids in D23580. These plasmid maps are shown in **Sup. Fig. 1**. In addition, we note that we were able to see Tn insertions in most plasmid-encoded genes (93% to 100%, depending on the plasmid), as summarized in the below table:

Plasmid	# of genes present	# of genes hit
ST4/74 #1	108	107
ST4/74 #2	98	97
ST4/74 #3	10	10
D23580 #1	84	78
D23580 #2	2	2
D23580 #3	3	3
D23580 #4	145	139

This paper focuses on fitness changes in chromosomally-encoded genes, but we note that fitness changes in plasmid-encoded genes may provide a foundation for very interesting future studies!

Figure 1 legend. Explain precisely what the “-bile” and “-Polymyxin B” conditions are.

In both cases, -bile and -polymyxin B means the experiment was done only in LB without any stressor. We have updated the figure legend.

Supplementary Figure 3 legend. Specify which serovar the strains used for Panels CDEFG belong to.

We have added all headings to these subpanels.

Supplementary Table 5 is an excellent idea. To improve the reproducibility of experiments in other laboratories, add columns to the table that describe both the supplier and the product code of the individual chemicals.

We have added in this information at **Supplemental Table 6**.

Decision Letter, first revision:

5th Feb 2024

Dear Dr. Monack,

Thank you for submitting your revised manuscript "High-throughput Fitness Experiments Reveal Specific Vulnerabilities of Human-Adapted Salmonella Isolates During Stress and Infection" (NG-A63455R). It has now been seen by the original referees and their comments are below. The reviewers find that the paper has improved in revision, and therefore we'll be happy in principle to publish it in Nature Genetics, pending minor revisions to satisfy the referees' final requests and to comply with our editorial and formatting guidelines.

Sincerely,

Michael Fletcher, PhD
Senior Editor, Nature Genetics

ORCID: 0000-0003-1589-7087

Reviewer #1 (Remarks to the Author):

The authors managed to convince me in part on the point I had raised, namely the generalizability and reliability of their results. I still think that part of the message is biased, but their approach will definitely indicate the most important genes involved in *Salmonella enterica* adaptive landscape. I also appreciated the author's efforts to respond to all the reviewers' comments. I therefore believe that this important work deserves to be published in *Nature Genetics*.

Reviewer #2 (Remarks to the Author):

The revised version of the manuscript is much improved. I would like to commend the authors for their large efforts to do so – and especially their ongoing effort to make the data available in an interactive interface! Without wanting to turn this to another round of revision (I am satisfied with current version), here are some small things I noticed when reading the revised version. Authors can decide whether they want to fix them (or not) in final version.

Essentiality: it would help if there is one table for all 4 serovars (Suppl. Table 5), so reader can easily access which genes are essential in all vs specific backgrounds. Also all 3 examples of conditional essentiality mentioned in the text lead to straightforward hypotheses, based on literature: a) for *igaA*, this means that the Rcs systems is likely mutated in Typhi/Paratyphi; b) for *rpoE*, deletions are not tolerated when periplasmic glycans or O-antigen is missing (PMID 30084765) – so addition *rfbE* or loss-of-functions mutations in *mdoG/H* in Typhi are likely behind this; c) essentiality of ISC in Paratyphi, means that the SUF system is not working, as authors show later (synthetic lethality between 2 systems across different bacteria is known). I would think leaving them as observations, if answer is rather obvious or easy to look for (based on presence missense mutations), is somewhat unsatisfactory.

Supp. Figures:

4: impossible to see violin plots shape in this graph. Advantage of violin plots is that one can see the distribution of the data, and the IQR. As plotted, one can only see outliers.

5-8, 15: there is no key to explain colors on the y-axis (gene side). It would be useful if cluster numbers mentioned in text (31, 33...) are also shown here.

20: why as separate fig (which just makes manuscript larger and shows data twice), rather than replacing the corresponding fig panels? Bacteria grow logarithmically

Final Decision Letter:

25th Apr 2024

Dear Denise,

I am delighted to say that your manuscript "High-throughput Fitness Experiments Reveal Specific

"Vulnerabilities of Human-Adapted Salmonella Isolates During Stress and Infection" has been accepted for publication in an upcoming issue of Nature Genetics.

Your paper will be published online after we receive your corrections and will appear in print in the next available issue. You can find out your date of online publication by contacting the Nature Press Office (press@nature.com) after sending your e-proof corrections.

Please note that *Nature Genetics* is a Transformative Journal (TJ). Authors may publish their research with us through the traditional subscription access route or make their paper immediately open access through payment of an article-processing charge (APC). Authors will not be required to make a final decision about access to their article until it has been accepted. Find out more about Transformative Journals

Authors may need to take specific actions to achieve compliance with funder and institutional open access mandates. If your research is supported by a funder that requires immediate open access (e.g. according to Plan S principles) then you should select the gold OA route, and we will direct you to the compliant route where possible. For authors selecting the subscription publication route, the journal's standard licensing terms will need to be accepted, including <https://www.nature.com/nature-portfolio/editorial-policies/self-archiving-and-license-to-publish>. Those licensing terms will supersede any other terms that the author or any third party may assert apply to any version of the manuscript.

If you have not already done so, we invite you to upload the step-by-step protocols used in this manuscript to the Protocols Exchange, part of our on-line web resource, natureprotocols.com. If you complete the upload by the time you receive your manuscript proofs, we can insert links in your article that lead directly to the protocol details. Your protocol will be made freely available upon publication of your paper. By participating in natureprotocols.com, you are enabling researchers to more readily reproduce or adapt the methodology you use. [Natureprotocols.com](http://natureprotocols.com) is fully searchable, providing your protocols and paper with increased utility and visibility. Please submit your protocol to <https://protocolexchange.researchsquare.com/>. After entering your nature.com username and password you will need to enter your manuscript number (NG-A63455R1). Further information can be found at <https://www.nature.com/nature-portfolio/editorial-policies/reporting-standards#protocols>

Sincerely,

Michael Fletcher, PhD
Senior Editor, Nature Genetics
ORCID: 0000-0003-1589-7087